# Effective and Efficient Time-Varying Counterfactual Prediction with State-Space Models

**Haotian Wang**[1]     **Haoxuan Li**[2]     **Hao Zou**[3]     **Haoang Chi**[4]     **Long Lan**[1]
**Wanrong Huang**[1]     **Wenjing Yang**[1*]

[1]College of Computer Science and Technology, National University of Defense Technology
[2]Center for Data Science, Peking University
[3]Tsinghua University
[4]Intelligent Game and Decision Lab
`wanghaotian13@nudt.edu.cn`   `hxli@stu.pku.edu.cn`

## ABSTRACT

Time-varying counterfactual prediction (TCP) from observational data supports the answer of when and how to assign multiple sequential treatments, yielding importance in various applications. Despite the progress achieved by recent advances, e.g., LSTM or Transformer based causal approaches, their capability of capturing interactions in long sequences remains to be improved in both prediction performance and running efficiency. In parallel with the development of TCP, the success of the state-space models (SSMs) has achieved remarkable progress toward long-sequence modeling with saved running time. Consequently, studying how Mamba simultaneously benefits the effectiveness and efficiency of TCP becomes a compelling research direction. In this paper, we propose to exploit advantages of the SSMs to tackle the TCP task, by introducing a counterfactual Mamba model with **C**ovariate-based **D**ecorrelation towards **S**elective **P**arameters (Mamba-CDSP). Motivated by the over-balancing problem in TCP of the direct covariate balancing methods, we propose to de-correlate between the current treatment and the representation of historical covariates, treatments, and outcomes, which can mitigate the confounding bias while preserve more covariate information. In addition, we show that the overall de-correlation in TCP is equivalent to regularizing the selective parameters of Mamba over each time step, which leads our approach to be effective and lightweight. We conducted extensive experiments on both synthetic and real-world datasets, demonstrating that Mamba-CDSP not only outperforms baselines by a large margin, but also exhibits prominent running efficiency.

## 1 INTRODUCTION

Inferring counterfactual outcomes in time series data is of critical importance across a broad range of domains (Van der Klaauw, 2002; Heidari & Krause, 2018; Li et al., 2021). In particular, time-varying counterfactual prediction (TCP) focuses on estimating counterfactual outcomes over various possible sequences of interventions (Melnychuk et al., 2022), supports the answer of when and how to assign multiple sequential treatments (Bica et al., 2020; Melnychuk et al., 2022; Huang et al., 2024).

Prior research has demonstrated that TCP under dynamic treatment regimes introduces significantly more challenges than in static settings, primarily due to increasing generalization errors (Alaa & Schaar, 2018) and potential biases arising from multiple prediction steps (Frauen et al., 2023). To be specific, the overall confounding bias could accumulate over time if the bias correction is not sufficient in previous time steps (Austin et al., 2006; Huang et al., 2024). To alleviate such issues, recent advances have focused on integrating sequential debiasing techniques into various models. Notable examples include recurrent marginal structural networks (RMSNs) (Lim, 2018), counterfactual recurrent networks (CRN) (Bica et al., 2020), G-net (Li et al., 2020), and the Causal Transformer (CT) (Melnychuk et al., 2022).

---

*Corresponding author.

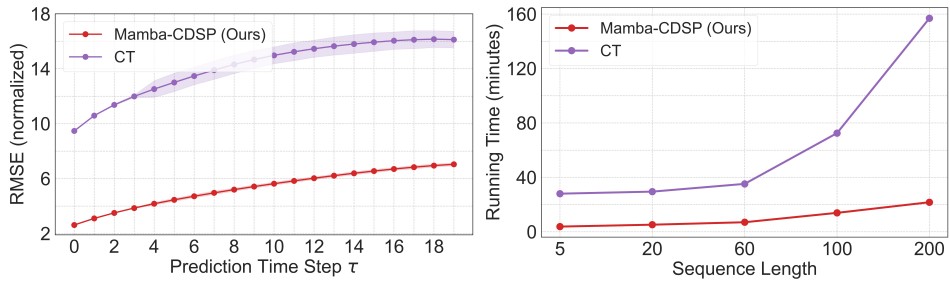

(a) RMSE with varying prediction time step     (b) Running time with varying sequence length

Figure 1: Effectiveness and efficiency comparison between Mamba-CDSP and Causal Transformer on the Tumor simulator dataset. The time complexity (right) and the estimation error (left, with sequence length as 60) of CT grows exponentially with sequence length and prediction time steps.

However, we observe that existing methods suffer from both the efficiency and the effectiveness in the TCP task. As shown in Fig. 1 (a), regarding the effectiveness, increasing prediction time step from 0 to 12 leads to dramatically increasing estimation error of the CT method (from less than 10 to over 15). On the other hand, when the sequence length increases, the running efficiency of CT scales with nearly quadratic complexity (from 26 min to over 120 min) when increasing the length of input sequence from 5 to 200. We further inform two critical reasons lying behind such phenomenon: (1) Impact of the backbone. Transformer, i.e., the backbone of CT, is not capable of modeling long sequences outside the attention window, while capturing all time stamps in the attention results in quadratic complexity w.r.t. the sequence length Gu & Dao (2023). (2) Drawback of concurrent balancing strategies. First, previous TCP methods Melnychuk et al. (2022); Bica et al. (2020) often control the **overall** confounding bias at the final time step. Following exactly the debiasing protocols in static setting, such operations ignore the accumulation of confounding bias over time series (Austin et al., 2006; Frauen et al., 2023). Second, directly balancing representations across treatment groups could raise the over-balancing problem, especially in the case of time-varying prediction (Huang et al., 2024), which will further corrupt the representations of covariates and degrade the overall estimation performance.

To tackle the backbone issue of TCP, we note that the success of Mamba (Gu & Dao, 2023), a data-selective state-space model (SSM) architecture (Gu et al., 2021a;b), offers new possibilities due to its capability of mapping across continuous sequences with nearly linear complexity Gu & Dao (2023). Meanwhile, to address problems of balancing in the TCP task, we propose an efficient bias correction method called **C**ovariate-based **D**ecorrelation towards **S**elective **P**arameters (CDSP) for building TCP-oriented Mamba. Specifically, our designed CDSP mechanism introduces a de-correlation strategy to address sequential confounding bias by removing the cross-covariance between current treatments and representation of historical covariates, treatments, and outcomes. On the one hand, our CDSP addresses the drawback of overall balancing in the TCP task with a **step-to-step** bias control manner. We show that our proposed CDSP can be decomposed into orthogonal regularization applied to selective parameters at each time step, concerning the linear properties of Mamba. On the other hand, empirical evidence show that our CDSP implements the bias correction in a covariate-preservation style, which also alleviates the over-balancing problem. Besides, to adapt the Mamba model for TCP, we modify its architecture by replacing the convolutional layer with a dropout layer (Gu & Dao, 2023) to mitigate overfitting. In summary, our main contributions are as follows:

(1) We develop an efficient and effective Mamba-based framework named **Mamba-CDSP** for the task of TCP. To the best of our knowledge, this is the pioneer Mamba model tailored to counterfactual prediction.

(2) We design a novel covariance de-correlation-based mechanism to migrate the sequential confounding bias. By accounting for the trade-off between prediction and bias correction, our CDSP mechanism overcomes the critical drawback of existing sequential debiasing methods.

(3) We validate the proposed Mamba-CDSP model across synthetic, semi-synthetic, and real-world data, demonstrating that our framework outperforms existing baselines by a large margin in performance with much more efficient training and inference phases.

## 2 RELATED WORK

**Balancing Covariates in Static Counterfactual Prediction.** Static methods focus on correcting the explicit confounding bias across different groups via diverse strategies, including reweighting (Kuang et al., 2020), matching (Stuart, 2010) or covariate balancing (Athey et al., 2018). Specifically, previous work have accounted for the finite-sample degrading effect of confounding bias raised by observed confounders (Shalit et al., 2017; Alaa & Schaar, 2018), even when the ignorability principle holds.

**Development on TCP with Statistical Approaches.** Earlier Studies on estimating treatment effects focuses in the area of epidemiology, where the g-estimation, Structural Nested Models (SNM) and Marginal Structural Models (MSM) are developed in the regime of statistical analysis (Robins, 1986; 1999; Robins & Hernan, 2008). Meanwhile, a bunch of methods has introduced non-parametric models with uncertainty estimation into treatment effect estimation over the whole population (Xu et al., 2016; Roy et al., 2017; Schulam & Saria, 2017). To be specific, a family of approaches built on Bayesian nonparametric models (primarily GP) have been proposed to better encode structure in temporal trends and treatment effects (Shi et al., 2012; Arbour et al., 2021; Xu et al., 2016). In recent, a multi-task Gaussian process model has been established by decoupling the response trend into individual-level and unit-level ones Chen et al. (2023).

**Development on TCP with Deep Models.** Recent advances construct the bias correction on top of a series of deep sequential models, and have achieved state-of-the-art (SOTA) estimation performance. For instance, RMSMs (Lim, 2018) incorporates the propensity score-based reweighting into recurrent neural networks (RNNs). Subsequently, counterfactual recurrent network (CRN) (Bica et al., 2020; Wang et al., 2024a; Berrevoets et al., 2021) combines the RNNs with adversarial covariate balancing techniques. Besides, the G-net (Li et al., 2020) performs deep G-computation with the simultaneous parameterization of the outcome and covariates. To further capture the long-range sequences, the Causal Transformer (Melnychuk et al., 2022) designed a transformer-tailored architecture with domain-confusion-based balancing to capture long-range and complex sequences.

**Distributional Counterfactual Estimation over Time.** Several recent studies have emerged to estimate the entire counterfactual distribution rather than the averaged outcome (Chernozhukov et al., 2013; Kennedy et al., 2023; Wang et al., 2018). To realize the generation of counterfactual density in high-dimensional over time series, Wu et al. (2024) approximates the counterfactual distribution with a generative model without explicitly estimating its density, which enables a wider range of application scenarios including continuous treatments.

**State-space models (SSMs) for long sequence modeling.** Stemming from the signal transformation theory, the state-space model aims to turn a continuous input signal into an output signal (Gu et al., 2021b; Gu & Dao, 2023; Gu et al., 2021a). In recent, (Gu et al., 2021a) has developed an efficient SSM named S4 by constructing the HiPPO operator with convolutional acceleration. The S4 model is further improved by incorporating the parallel scan into S4 layer (Smith et al., 2022). Regrading the expressivity of S4, Mehta et al. (2022) developed the Gated state-space layer by introducing more gating units. In recent, a data-dependent framework named Mamba builds a generic long-sequence backbone by replacing fixed parameters in S4 into selective parameters (Gu & Dao, 2023). As Mamba outperforms Transformers on a variety of realistic applications with linear scaling towards long-sequence, we aim to empower the capability of Mamba for estimating counterfactual outcomes over time series, i.e., constructing a new backbone for the area of causal inference.

## 3 PROBLEM DEFINITION

**Notations.** This paper uses upper-case letters (e.g., $X$) to denote random variables, with lower-case letters (e.g., $x$) denoting the corresponding realizations. Besides, the bold letters (e.g., $\mathbf{V}$) refer to vectors/matrices, and

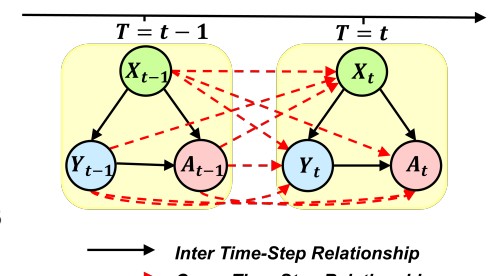

non-bold letters (e.g., $V$) refer to scalars. Following previous protocols (Melnychuk et al., 2022; Huang et al., 2024), we let $i$ refer to $i$-th individual (e.g., a patient) and with historical treatment trajectories over multiple time steps (e.g., a health plan with one month) from $a = 1, \ldots T^{(i)}$. At each time step $t$, one can observe $i$-th individual's features from four aspects: (1) Treatment Assignment $\mathbf{A}_t^{(i)} \in \{a_1, \ldots, a_{d_a}\}$ with $d_a$ categories (e.g., a health plan with $d_a$ kinds of drugs to be taken in turn); (2) The time-varying covariates $\boldsymbol{X}_t^{(i)} \in \mathcal{R}^{d_x}$ with $d_x$ dimensions (e. g., some disease-dependent factors of the patient); (3) The static features $\boldsymbol{V}^{(i)}$ that keeps invariant over time (e. g., gender and age of the patient); (4) The outcomes $\boldsymbol{Y}_t^{(i)} \in \mathcal{R}^{d_y}$ with dimensions $d_y$. Then the observed dataset can be defined as $\mathcal{D} = \{\{\boldsymbol{x}_t^{(i)}, \boldsymbol{v}^{(i)}, \boldsymbol{a}_t^{(i)}, \boldsymbol{y}_t^{(i)}\}_{a=1}^T\}_{i=1}^N$, where $N$ is the sample number. For the sake of clarity, we omit the superscript $(i)$ for each individual unless needed and use $\boldsymbol{X_t}$ to represent the overall features, i.e., both $\boldsymbol{V}^{(i)}$ and $\boldsymbol{X}_t^{(i)}$, at time step $t$.

**Problem Setup** Throughout this paper, we define historical information at time step $t$ as $\overline{\mathbf{H}}_t = \{\overline{\mathbf{X}}_t, \overline{\mathbf{A}}_{t-1}, \overline{\mathbf{Y}}_t, \mathbf{V}\}$, where $\overline{\mathbf{X}}_t = (\mathbf{X}_1, \ldots, \mathbf{X}_t)$, $\overline{\mathbf{Y}}_t = (\mathbf{Y}_1, \ldots, \mathbf{Y}_t)$, and $\overline{\mathbf{A}}_{t-1} = (\mathbf{A}_1, \ldots, \mathbf{A}_{t-1})$ (Melnychuk et al., 2022; Allam et al., 2021). As shown in Fig. 2, we follow standards in time-varying counterfatual prediction (Melnychuk et al., 2022; Huang et al., 2024; Li et al., 2021; Bica et al., 2020; Robins & Hernan, 2008), and model the causal relationship among variables as follows: (a) The treatment assignment $A_t$ is affected by the time-varying covariates $X_t$ as well as the history, including $\overline{\mathbf{Y}}_t$, $\overline{\mathbf{X}}_{t-1}$ and $\overline{\mathbf{A}}_{t-1}$, indicating the presence of time-varying confounders. (b) The outcome $Y_t$ is affected by the current covariates, and past treatment $\overline{\mathbf{A}}_{t-1}$, outcomes $\overline{\mathbf{Y}}_{t-1}$, and covariates $\overline{\mathbf{X}}_{t-1}$. (c) The time-varying covariates $X_t$ is affected by all past information, including $\overline{\mathbf{A}}_{t-1}$, $\overline{\mathbf{Y}}_{t-1}$, and $\overline{\mathbf{X}}_{t-1}$.

**Estimation Target.** We define the $\tau \geq 1$ as the projection horizon for a $\tau$-step-ahead prediction, with $\overline{a}_{t:t+\tau-1} = (a_t, a_{t+1}, \ldots, a_{t+\tau-1})$ is the sequence of the (imagined) assigned treatments in the future $\tau$ time steps. Following the potential-outcome framework (Splawa-Neyman et al., 1990; Rubin, 1978), we define the estimand of our problem as $\mathbb{E}\left(\boldsymbol{y}_{t+\tau}\left[\overline{\boldsymbol{a}}_{t:t+\tau-1}\right] \mid \overline{\boldsymbol{H}}_t\right)$, where $\overline{\boldsymbol{H}}_t = \left(\overline{\boldsymbol{X}}_t, \overline{\boldsymbol{A}}_{t-1}, \overline{\boldsymbol{Y}}_t, \boldsymbol{V}\right)$ is the historical observations.

**Basic Assumptions.** We present basic assumptions to support unbiased identification for time-varying counterfactual prediction using observational data (Robins & Hernan, 2008):

(1) **Consistency.** If $\mathbf{A}_t = \overline{\mathbf{a}}_t$ is a given sequence of treatments, then $\mathbf{Y}_{t+1}\left[\overline{\mathbf{a}}_t\right] = \mathbf{Y}_{t+1}$. This means that the potential outcome under treatment sequence $\overline{\mathbf{a}}_t$ coincides for the patient with the observed (factual) outcome, conditional on $\overline{\mathbf{A}}_t = \overline{\mathbf{a}}_t$.

(2) **Sequential Overlap.** There is always a non-zero probability of receiving/not receiving any treatment for all the history space over time: $0 < \mathbb{P}\left(\mathbf{A}_t = \mathbf{a}_t \mid \overline{\mathbf{H}}_t = \overline{\mathbf{h}}_t\right) < 1$, if $\mathbb{P}\left(\overline{\mathbf{H}}_t = \overline{\mathbf{h}}_t\right) > 0$, where $\overline{\mathbf{h}}_t$ is some realization of a patient history.

(3) **Sequential Ignorability.** The current treatment is independent of the potential outcome, conditioning on the observed history: $\mathbf{A}_t \mathbf{Y}_{t+1}\left[\mathbf{a}_t\right] \mid \overline{\mathbf{H}}_t, \ \forall \mathbf{a}_t$. This implies that there are no unobserved confounders that affect both treatment and outcome.

# 4 METHOD

## 4.1 PRELIMIARIES

Inspired by the continuous signal mapping (Gu et al., 2021b;a; Gu & Dao, 2023), the family of state-space models (SSM) is derived from solving the time-varying differential equations. To be specific, SSM aims to estimate a mapping from an **input signal** $x$ to a **output signal** $y$: $x(t) \in \mathcal{R} \mapsto T(a) \in \mathcal{R}$ through the transition of a hidden state $h(t) \in \mathcal{R}^N$. The continuous form of SSM models can be

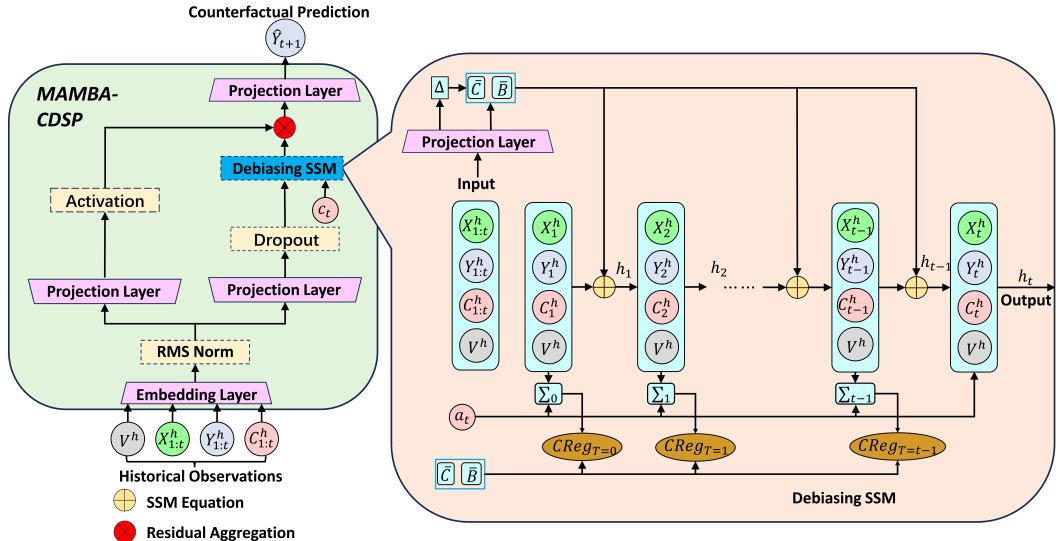

Figure 3: The Step-by-step Framework our proposed Mamba-CDSP.

written as follows:

$$h'(t) = \mathbf{C}h(t) + \mathbf{B}x(t), \ T(a) = \mathbf{R}h(t), \tag{1}$$

where $\mathbf{C} \in \mathcal{R}^{N \times N}$ is the evolution parameter, $\mathbf{B} \in \mathcal{R}^{\mathcal{N} \times 1}, \mathbf{R} \in \mathcal{R}^{1 \times N}$ are projection parameters. In recent, the S4 model (Gu et al., 2021a) provides a discrete form of the equation in equation 1 by discretizing $\mathbf{C}$ and $\mathbf{B}$ into $\overline{C}$ and $\overline{B}$ as follows:

$$h_t = \overline{\mathbf{C}}h_{t-1} + \overline{\mathbf{B}}x_t, \ y_t = \mathbf{R}h_t, \tag{2}$$

where $\overline{\mathbf{C}} = \exp(\mathbf{\Delta C}), \overline{\mathbf{B}} = (\mathbf{\Delta C})^{-1}(\exp(\mathbf{\Delta C}) - \mathbf{I}) \cdot \mathbf{\Delta B}$ and $\Delta$ represents the time step. Although S4 has the advantage in its fast computation with convolution operators, the lack of sequential-sensitive attention on different time steps becomes a performance bottleneck. To overcome this issue, the Mamba (Gu & Dao, 2023) model incorporates the selective mechanism into S4 by generating $\Delta, \overline{\mathbf{C}}$ and $\overline{\mathbf{B}}$ from the input $x$ with a linear-projection layer, i.e., the *selective parameters*.

### 4.2 MODELING TIME-VARYING TREATMENT EFFECTS USING MAMBA

**Input and Output.** Our Counterfactual Mamba (C-MAMBA) is designed as an integral block. During the inference phase, we treat the historical information $\overline{\mathbf{H}}_t = (\overline{\mathbf{X}}_t, \overline{\mathbf{A}}_{t-1}, \overline{\mathbf{Y}}_t, \mathbf{V})$ with $\boldsymbol{a}_t$ as the input signal $x(t)$ of Mamba, and $\mathbf{Y}_{t+1}$ as the output signal $T(a)$. As shown in Fig. 3, the input signals are embedded into the representation space with linear embedding. Following (Gu & Dao, 2023), a Root Mean Square Layer Normalization (RMS-Norm) is deployed on embedded $\tilde{X}$, and two projection layers split $\tilde{X}$ into two branches. Inside the SSM, the input at time step $t$, i.e., $(\mathbf{X}_t, \boldsymbol{a}_t, \overline{\mathbf{Y}}_t, \mathbf{V})$, are fused with historical information $h_{t-1}$ and produces the hidden state $h_t$.

**Adaptation of Mamba Structure.** On the one hand, the branch of the left side in Fig. 3 remains the same as in (Gu & Dao, 2023), where a *silu* activation layer is deployed to transform the projected branch. On the other hand, the right-side branch is first filted with a *silu* activation layer, and then fed into a *dropout* layer. **We note that in this main branch, the original convolutional layer (Gu & Dao, 2023) is replaced with a dropout layer.** The empirical motivation behind such operations is that we observe that the original Mamba model tends to overfit the training data, and the root-cause of such phenomenon is the 1-d convolution layer (Wang et al., 2024b). Originally, the convolutional layer is adapted to improve the mixture of neighboring tokens in language modeling (Gu & Dao, 2023), while such token-mixing mechanism might lead to the overfitting on the interaction among $\tilde{X}$. Hence, we remove the convolutional layer with a dropout layer to alleviate the overfitting phenomenon of Mamba structure on temporal data. Finally, the two branches are fused in a residual manner, i.e., an element-wise multiplication operation and a projection layer.

## 4.3 COVARIANCE-BASED DECORRELATION TOWARDS SELECTIVE PARAMETERS

Previous empirical studies (Huang et al., 2024) have pointed out that typical balancing methods for correcting confounding bias deteriorates the representations of the covariates itself. To overcome this issue, we start from the structure of Mamba models and design a novel **Covariate-based Decorrelation** towards **Selective Parameters** (CDSP) method to cut-off the correlation between current treatment $a_t$ and historical information $h_{t-1}$.

Recalling the transition equation in equation 2, we first expand the expression of $Cov(h_{t-1}, a_t)$ as follows:

$$Cov(h_{t-1}, a_t) = Cov\left(\sum_{i=1}^{t-1}\left(\boldsymbol{K}_i \tilde{X}_i^h\right), a_t\right) = \sum_{i=1}^{t-1} Cov\left(\boldsymbol{K}_i \tilde{X}_i^h, a_i\right) = \sum_{i=1}^{t-1} \boldsymbol{K}_i Cov\left(\tilde{X}_i^h, a_i\right),$$
(3)

where the second and the third equation are due to the property of cross-covariance. Meanwhile, due to the expression of the hidden transition equation in equation 2, the term $\boldsymbol{K}_i$ represents the time-accumulated multiplication of $\overline{C}_i$ and $\overline{B}_i{}^1$ as $\boldsymbol{K}_i = \overline{B}_i \Pi_{j=i}^{t-1} \overline{C}_j$. By denoting the term $Cov\left(\tilde{X}_i^h, a_t\right)$ as $\Sigma_{\tilde{X}_i^h, a_t}$, we then formally write the constraint, i.e., $\|Cov(h_{t-1}, a_t)\|_2^2$, as follows:

$$\|Cov(h_{t-1}, a_t)\|_2^2 \leq \sum_{i=1}^{t-1} \|\boldsymbol{K}_i \Sigma_{\tilde{X}_i^h, a_t}\|_2^2.$$
(4)

We further derive the following proposition:

**Proposition 1.** *Finding $\boldsymbol{K}_i$ to minimize equation 4 equals to minimizing $\boldsymbol{K}_i \Sigma_{\tilde{X}_i^h, a_t} \Sigma_{\tilde{X}_i^h, a_t}^T = \boldsymbol{0}$.*

Based on the above proposition, we design our proposed CSDP regularization term as follows:

$$\mathcal{L}_{\mathrm{CSDP}}(\overline{C}, \overline{B}) = \sum_{i=1}^{t-1} \left\| \overline{B}_i \Pi_{j=i}^{t-1} \overline{C}_j \Sigma_{\tilde{X}_i^h, a_t} \Sigma_{\tilde{X}_i^h, a_t}^T \right\|^2,$$
(5)

where the term $\Sigma_{\tilde{X}_i^h, a_t} \Sigma_{\tilde{X}_i^h, a_t}^T$ only concerns w.r.t. the representational versions of $\boldsymbol{Y}, \boldsymbol{A}, \boldsymbol{X}$ and $\boldsymbol{V}$, and can be pre-computed for each batch of samples. Besides, we fit the outcome prediction by minimizing the factual loss of the next outcome. Such operation can be done via the mean squared error (MSE):

$$\mathcal{L}_{\mathrm{MSE}} = \left\| \hat{\boldsymbol{Y}}_{t+1} - \Phi_Y\left(h_t\right) \right\|^2,$$
(6)

where $\Phi_Y$ represent the parameter of the projection layer after the residual fusion of two branches of our CDSP model (shown in Fig. 3).

*Remark.* We note that the objective $\mathcal{L}_{\mathrm{MSE}}$ updates **the whole CDSP**, while the objective $\mathcal{L}_{\mathrm{CSDP}}$ **only updates the selective parameters $\overline{C}$ and $\overline{B}$.** The overall objective function is $\mathcal{L}_{\mathrm{MSE}} + \alpha \mathcal{L}_{\mathrm{CSDP}}$, where $\alpha$ is the regularization parameter.

*Remark* (Computational Complexity Analysis). We perform computational complexity analysis and compare our proposed CDSP with previous adversarial balancing methods. Denote the overall length of the time horizon as $T$, and the representation dimension of $A$ and $X$ as $d_A^h$ and $d_X^h$, respectively. For adversarial balancing modules, previous practice shows that the discriminator has usually 2 layers with $d_X^h$ for each layer. Then the overall complexity of our proposed CDSP method is $\mathcal{O}(B((d_X^h)^3 + d_A^h)T)$, where as the overall complexity of ADB can be derived as $\mathcal{O}(B((d_X^h)^3 + (d_X^h)^2 d_Y^h + |A|)T)$ (see detailed derivation in Appendix A.4.3).

## 4.4 THEORETICAL ANALYSIS

To further show the superiority of our proposed method over existing balancing methods, we theoretically derive the upper bounds of counterfactual prediction risks, and find our CDSP outperforms existing adversarial balancing methods by preserving a tighter counterfactual prediction risk bound (see proofs in Appendix A.3). We assume the Gaussian covariates $\overline{X}_{|a} \sim \mathcal{N}(\mu_a, \sigma_a I)$ and linear outcome

---

[1]We note that $\overline{C}$ and $\overline{B}$ differs in different time steps.

structure $Y(a) = W^a \Phi \overline{X}_{|a} + \epsilon^a$, where $\epsilon^a \sim N(0, 1)$. Meanwhile, we use $N_a$ to denote sample number of each treatment arm, with $W$ and $\Phi$ denoted as model parameters of representation layers and prediction head, respectively. We omit the time-index (superscript) for convenience.

**Definition 1.** *The expected precision in Estimation of Counterfactual Prediction (ECP) is defined as:*

$$\epsilon_{\text{ECP}}(W, \Phi) = \int_{\mathcal{X}} (\hat{\tau}_f(x) - \tau(x))^2 p(x) dx. \tag{7}$$

**Theorem 1.** *(1) The following risk bound in the finite sample regime holds for the vanilla Empirical Risk Minimization (ERM) model* **without any balancing modules** *with probability $1 - 2\eta$:*

$$\epsilon_{\text{ECP}}^{vanilla}(W, \Phi) \leq 2 \left( vr_1^2 \|(\mu_1 - \mu_0)\|_2^2 + v(\sqrt{\sigma_1} - \sqrt{\sigma_0})r_2^2 + C - \left( \sqrt{\frac{1}{\eta}} (\sum_{j=1}^{2} \sqrt{\kappa_j}) + \overline{\sigma} \right) \right), \tag{8}$$

*where $\overline{\sigma} = \frac{1}{N_1} \sum_{i=1}^{N_1} \epsilon_i^{a2}$, and $\{\kappa_j\}_{j=1}^{2}$ denotes some constants w.r.t. the moments of $X$ and $Y$.*

*(2) The following risk bound in the finite sample regime holds for the vanilla prediction cases* **with adversarial balancing (ADB) modules** *with probability $1 - 2\eta$:*

$$\epsilon_{\text{ECP}}^{adb}(\tilde{W}, \tilde{\Phi}) \leq 2 \left( \frac{(2 + \sigma_0 + \sigma_1)(r_1 r_3)^2}{4} \|\mu_1 - \mu_2\|_2^2 + C - \left( \sqrt{\frac{1}{\eta}} (\sum_{j=1}^{2} \sqrt{\kappa_j}) + \overline{\sigma} \right) \right). \tag{9}$$

*(3) The following risk bound in the finite sample regime holds for the vanilla prediction cases* **with our proposed CDSP module** *with probability $1 - 2\eta$:*

$$\epsilon_{\text{ECP}}^{cdsp}(\tilde{W}, \tilde{\Phi}) \leq 2 \left( \frac{(r_1 r_3)^2}{2} \|\mu_1 - \mu_2\|_2^2 + C - \left( \sqrt{\frac{1}{\eta}} (\sum_{j=1}^{2} \sqrt{\kappa_j}) + \overline{\sigma} \right) \right). \tag{10}$$

Based on the above results, we demonstrate the superiority of our CDSP method as follows.

**Corollary 1** (Over-balancing of ADB method)**.** *When the following condition holds, the risk bound of vanilla ERM model is more tighter than that of adversarial balancing methods:*

$$\|\mu_1 - \mu_0\|_2^2 \leq \frac{v(\sqrt{\sigma_1} - \sqrt{\sigma_0})^2 r_2^2}{vr_1^2 - (2 + \sigma_1 + \sigma_0)r_1^2 r_3^2/4}. \tag{11}$$

This illustrates that as the gap between two groups decreases, the harm of over-balancing is much larger than that of observed confounding bias.

In addition, we conclude that our CDSP method has a tighter bound compared with the existing adversarial balancing methods, by noting the following inequality:

$$\frac{(r_1 r_3)^2}{2} \|\mu_1 - \mu_2\|_2^2 \leq \frac{(2 + \sigma_0 + \sigma_1)(r_1 r_3)^2}{4} \|\mu_1 - \mu_2\|_2^2. \tag{12}$$

**Corollary 2** (CDSP outperforms ADB with a tighter bound)**.** *Our CDSP method always outperforms ADB methods with a tighter bound, i.e., $\epsilon_{\text{ECP}}^{cdsp}(\tilde{W}, \tilde{\Phi}) \leq \epsilon_{\text{ECP}}^{adb}(\tilde{W}, \tilde{\Phi})$.*

## 5 EXPERIMENTS

**Benchmarks.** Following common practice in benchmarking for counterfactual inference, all the methods are validated on three datasets, including the synthetic tumor growth data (Geng et al., 2017), the MIMIC-III-based semi-synthetic data (Melnychuk et al., 2022; Schulam & Saria, 2017), the MIMIC-III real-world data (Johnson et al., 2016). To further validate our method with high-dimensional, long-range sequences, we follow (Melnychuk et al., 2022) by generating patient observations with outcomes under endogeneous and exogeneous dependencies while considering treatment effects. Details of MIMIC-III data are present in Appendix A.4.1.

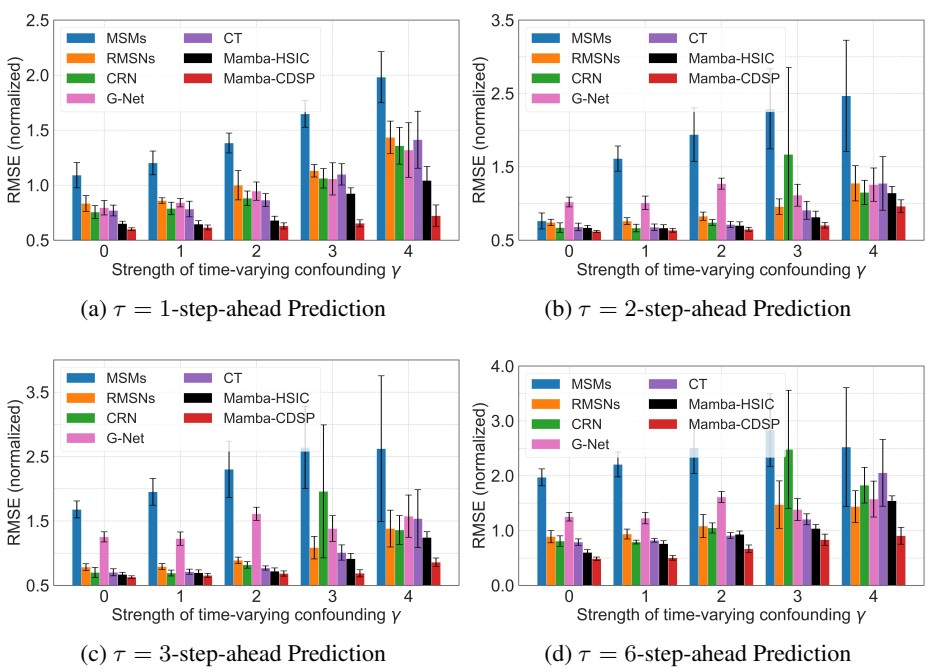

Figure 4: Results for fully synthetic data based on tumor growth simulator, which are reported as the mean performance averaged over five runs with different seeds.

**Baselines.** Although a bunch of temporal counterfactual estimation methods are present, this paper focuses on several state-of-the-art (SOTA) models for examination, including the Causal Transformer (Melnychuk et al., 2022), G-Net (Li et al., 2021), Recurrent Marginal Structural Networks (RMSNs) (Lim, 2018), Counterfactual Recurrent Network (CRN) (Bica et al., 2020) and Marginal Structural Models (MSMs) (Robins et al., 2000). We provide two realizations of our Mamba model, including the Mamba-CDSP method with covariance de-correlation and the Mamba-HSIC method by introducing the covariate balancing with Hilbert-Schmidt independence criterion (Gretton et al., 2005). (See details of baselines in Appendix A.4.2)

**Evaluation Protocols.** We evaluate the proposed Mamba-CDSP in a variety of scenarios, including:

*Supervised Prediction.* This scenario involves using historical data to predict future counterfactual outcomes over multiple time steps, following a standard train-test procedure. For the tumor-growth synthetic dataset, the evaluation protocols are different for one-step prediction and multiple-step prediction. For the semi-synthetic dataset, we account for all 8 possible combinations for three binary treatments during the one-step prediction task; and account for the random trajectories with $\tau_{\max} = 10$. Finally, for the real-world MIMIC-III dataset, we test all patient observations for one-step prediction, while selecting a subset of observations during multiple-step prediction (Melnychuk et al., 2022). (Details of evaluation protocols on three benchmarks are illustrated in Appendix A.4.1). All metrics are reported in the form of Root Mean Square Error (RMSE).

*Covariate Reconstruction.* This scenario involves using learned representations from baseline methods to reconstruct the original covariates. The evaluation metric is the reconstruction loss during the training process. All metrics are reported in the form of Root Mean Square Error (RMSE).

*Visualization of learned representations.* This scenario is to examine whether learned representations are contaminated by balancing strategies. The T-SNE technique is adopted for visualization (Van der Maaten & Hinton, 2008).

**Questions.** The empirical experiments are performed around the following three questions: (1) Can our proposed Mamba-CDSP outperform other baselines across various benchmarks? (2) Can our proposed Mamba-CDSP preserves the historical information of covariates when performing confounding bias correction?

Table 1: Results for semi-synthetic data for $\tau$-step-ahead prediction based on real-world MIMIC-III data, which are the average performance over five runs with different seeds, and * means statistically significant results (p-value $\leq 0.01$) using the paired-t-test compared with the best baseline.

| | $\tau=1$ | $\tau=2$ | $\tau=3$ | $\tau=4$ | $\tau=5$ | $\tau=6$ | $\tau=7$ | $\tau=8$ | $\tau=9$ | $\tau=10$ |
|---|---|---|---|---|---|---|---|---|---|---|
| MSMs | $0.37_{\pm0.01}$ | $0.57_{\pm0.03}$ | $0.74_{\pm0.06}$ | $0.88_{\pm0.03}$ | $1.14_{\pm0.10}$ | $1.95_{\pm1.48}$ | $3.44_{\pm4.57}$ | $>10.0$ | $>10.0$ | $>10.0$ |
| RMSNs | $0.24_{\pm0.01}$ | $0.47_{\pm0.01}$ | $0.60_{\pm0.01}$ | $0.70_{\pm0.02}$ | $0.78_{\pm0.04}$ | $0.84_{\pm0.05}$ | $0.89_{\pm0.06}$ | $0.94_{\pm0.08}$ | $0.97_{\pm0.09}$ | $1.00_{\pm0.11}$ |
| CRN | $0.30_{\pm0.01}$ | $0.48_{\pm0.02}$ | $0.59_{\pm0.02}$ | $0.65_{\pm0.02}$ | $0.68_{\pm0.02}$ | $0.71_{\pm0.01}$ | $0.72_{\pm0.01}$ | $0.74_{\pm0.01}$ | $0.76_{\pm0.01}$ | $0.78_{\pm0.02}$ |
| G-Net | $0.34_{\pm0.01}$ | $0.67_{\pm0.03}$ | $0.83_{\pm0.04}$ | $0.94_{\pm0.04}$ | $1.03_{\pm0.05}$ | $1.10_{\pm0.05}$ | $1.16_{\pm0.05}$ | $1.21_{\pm0.06}$ | $1.25_{\pm0.06}$ | $1.29_{\pm0.06}$ |
| EDCT | $0.29_{\pm0.01}$ | $0.46_{\pm0.01}$ | $0.56_{\pm0.01}$ | $0.62_{\pm0.01}$ | $0.67_{\pm0.01}$ | $0.70_{\pm0.01}$ | $0.72_{\pm0.01}$ | $0.74_{\pm0.01}$ | $0.76_{\pm0.01}$ | $0.78_{\pm0.01}$ |
| CT | $0.21_{\pm0.01}$ | $0.38_{\pm0.01}$ | $0.46_{\pm0.01}$ | $0.50_{\pm0.01}$ | $0.53_{\pm0.01}$ | $0.54_{\pm0.01}$ | $0.55_{\pm0.01}$ | $0.57_{\pm0.01}$ | $0.58_{\pm0.01}$ | $0.59_{\pm0.01}$ |
| Mamba-HSIC | $0.25_{\pm0.02}$ | $0.32_{\pm0.01}$ | $0.38_{\pm0.01}$ | $0.43_{\pm0.02}$ | $0.48_{\pm0.01}$ | $0.51_{\pm0.01}$ | $0.54_{\pm0.02}$ | $0.57_{\pm0.01}$ | $0.59_{\pm0.02}$ | $0.60_{\pm0.02}$ |
| Mamba-CDSP | $\mathbf{0.19}^*_{\pm0.01}$ | $\mathbf{0.25}^*_{\pm0.01}$ | $\mathbf{0.30}^*_{\pm0.01}$ | $\mathbf{0.34}^*_{\pm0.01}$ | $\mathbf{0.37}^*_{\pm0.01}$ | $\mathbf{0.42}^*_{\pm0.01}$ | $\mathbf{0.43}^*_{\pm0.01}$ | $\mathbf{0.44}^*_{\pm0.01}$ | $\mathbf{0.46}^*_{\pm0.01}$ | $\mathbf{0.47}^*_{\pm0.01}$ |

Table 2: Results for experiments with the real-world MIMIC-III data, which are the average performance over five runs with different seeds, and * means statistically significant results (p-value $\leq 0.01$) using the paired-t-test compared with the best baseline.

| | $\tau=1$ | $\tau=2$ | $\tau=3$ | $\tau=4$ | $\tau=5$ |
|---|---|---|---|---|---|
| MSMs | $6.37_{\pm0.26}$ | $9.06_{\pm0.41}$ | $11.89_{\pm1.28}$ | $13.12_{\pm1.25}$ | $14.44_{\pm1.12}$ |
| RMSNs | $5.20_{\pm0.15}$ | $9.79_{\pm0.31}$ | $10.52_{\pm0.39}$ | $11.09_{\pm0.49}$ | $11.64_{\pm0.62}$ |
| CRN | $4.84_{\pm0.08}$ | $9.15_{\pm0.16}$ | $9.81_{\pm0.17}$ | $10.15_{\pm0.19}$ | $10.40_{\pm0.21}$ |
| G-Net | $5.13_{\pm0.05}$ | $11.88_{\pm0.20}$ | $12.91_{\pm0.26}$ | $13.57_{\pm0.30}$ | $14.08_{\pm0.31}$ |
| CT | $4.60_{\pm0.08}$ | $9.01_{\pm0.21}$ | $9.58_{\pm0.19}$ | $9.89_{\pm0.21}$ | $10.12_{\pm0.22}$ |
| Mamba-HSIC | $4.72_{\pm0.05}$ | $5.19_{\pm0.19}$ | $7.24_{\pm0.25}$ | $8.30_{\pm0.28}$ | $9.25_{\pm0.30}$ |
| Mamba-CDSP | $\mathbf{4.41}^*_{\pm0.05}$ | $\mathbf{5.04}^*_{\pm0.13}$ | $\mathbf{5.14}^*_{\pm0.15}$ | $\mathbf{5.20}^*_{\pm0.18}$ | $\mathbf{5.25}^*_{\pm0.19}$ |

## 5.1 COUNTERFACTUAL PREDICTION: ANALYSIS AND EXAMINATION

**Performance Analysis on the Tumor-growth simulation.** We report the RMSE of the estimation performance (RMSE) across each baseline by controlling the confounding parameter $\gamma$. As shown in Fig. 4, the analysis is conducted w.r.t. different levels of $\gamma$ and $\tau$:

(1) With relatively small $\gamma$ and $\tau$, e.g., $\gamma=0$ in Fig. 4 (b), the counterfactual prediction results of each baseline, i.e., CT, Mamba-CDSP, CRN, are close to each other. However, with fixed $\gamma$ and increasing $\gamma$, e.g., $\gamma=0$ in Fig. 4 (e) and (f), the counterfactual estimation of Mamba-based methods, including Mamba-CDSP and Mamba-HSIC, outperforms other baselines.

(2) With increasing $\gamma$ for each $\tau$, e.g., $\gamma$ from 0 to 4 in Fig. 4 (e) and (f), our proposed Mamba-CDSP outperforms other baselines with an obvious margin. Such a phenomenon informs that our designed CDSP debiasing techniques achieve a better balance between prediction and preservation of the covariate information.

**Performance Analysis on the semi-synthetic and real-world MIMIC-III Dataset.** In similar to the Tumor-growth simulation, results on the semi-synthetic and real-world data further validate the superiority of our proposed Mamba-CDSP, especially on the real-world prediction (nearly reduces 50% of the RMSE compared to previous baselines). Especially, we note that due to the lack of realistic counterfatual outcomes in real-world data, the corresponding task becomes indeed a factual outcome prediction (Huang et al., 2024). Subsequently, ERM baselines without balancing modules should perform the best, and methods with over-balancing, by contrast, will worse the fitting of factual outcome. Hence, the superiority of our methods on real-world data reflects that our proposed CDSP regularization indeed preserves most of the covariate information such that the performance of factual prediction is not hurt. Furthermore, the stability of our Mamba-CDSP on the semi-synthetic data with long-sequence prediction results ($\tau=10$) also verifies the effectiveness of our method on long-sequence prediction. In addition, we obtain similar trends with empirical analysis on the M5 dataset (See Appendix 6).

## 5.2 FURTHER EMPIRICAL INSIGHT

**Reconstruction and Visualization: Better Trade-off Achieved by Our CDSP.** We follow Huang et al. (2024) to perform analysis on the Tumor-growth simulation by constructing and training LSTM-based decoder on learned representations from CT, Mamba-HSIC and our Mamba-CDSP. As shown in Fig. 5 (b), balancing the representations with HSIC leads to non-convergence with $\gamma=8$, while the vanilla mamba converges well. By construct, our proposed CDSP support obvious better

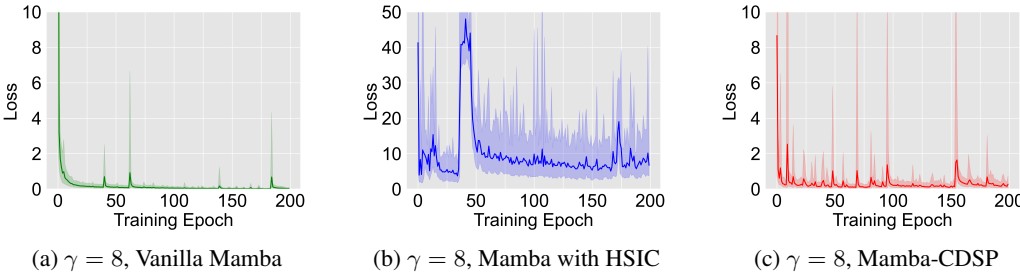

(a) $\gamma = 8$, Vanilla Mamba     (b) $\gamma = 8$, Mamba with HSIC     (c) $\gamma = 8$, Mamba-CDSP

Figure 5: Reconstruction loss curves for Mamba with different debiasing techniques.

convergence of reconstruction from learned representations. Besides, we further check the effect of each component of our Mamba-CDSP with an ablation study on the semi-synthetic MIMIC-III data in Table 5.

**Extreme Long Sequence, Sensitivity Analysis, and Failure Cases.** Besides, we also test our proposed Mamba-CDSP method by comparing it with CT on extreme sequences in Appendix (Table 13). We note that the contextual window of CT is set to 200 due to the limit of computational complexity. We also test the sensitivity of our proposed by tuning the de-correlation hyper-parameter $\alpha$ in the range of $[0, 0.0001, 0.001, 0.01, 0.1, 1.0, 10.0]$ in Appendix ( 12). We also note that our proposed Mamba-CDSP achieves near performance on the TG simulator and semi-synthetic MIMIC-III data when the predictio step $\tau$ is small, e.g., $\tau = 1$. The underlying reason contains two aspects: (a) confounding effect accumulates along with the time step; (2) the contextual window of Transformer baseline, i.e., CT, is set to the length of the whole sequence in most cases.

**Efficiency Analysis.** We analyze the running efficiency by reporting the parameter amount and running time per module of each baseline in Table 8 and Table 9 in the appendix for saving space. Results align with the original property of Mamba (Gu & Dao, 2023) that the parallel scanning mechanism guarantees that Mamba architecture enjoys superiority over the running efficiency.

## 6 CONCLUSION

This paper presents a pioneer work on studying the SSM-architecture-based estimator towards counterfactual prediction over time series. By introducing the covariance decorrelation into the Mamba model, our proposed Mamba-CDSP achieves superior empirical performance with a better balance between confounding bias correction and covariate information preservation.

**Limitations.** In view of empirical studies in Huang et al. (2024), it is necessary and important to consider the design of a paradigm that balances bias correction and covariate protection. The covariance decorrelation mechanism. However, our CDSP is designed specifically for linear SSMs, and more effective and general methods require further effort.

**Future Directions.** (1) Theoretical guidance on the design of the balancing technique for counterfactual prediction (not limited to time series prediction). As the covariate balancing and outcome prediction serves as two sub-tasks for counterfactual prediction, can we adapt learning theories from multi-task learning (Maurer et al., 2016; Royer et al., 2024) to inspire more reliable and theoretical-complete debiasing methods? (2) Practical guidance on debiasing for sequential counterfactual prediction. When transferring static prediction to sequential prediction, especially in multi-step prediction, how will the risk of confounding bias change?

## ACKNOWLEDGEMENT

This work was partially supported by the National Natural Science Foundation of China (No. 62372459, 62376282, 623B2002).

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

## A  APPENDIX

### A.1  NOTATION SUMMARY

For ease of reading, we summarize the correspondence between the abbreviation and full notation, as shown in Table 3. In addition, we named our proposed Covariate-based Decorrelation towards Selective Parameters method as CDSP.

Table 3: Notation table with detailed correspondence for concept, abbreviation, and full notation.

| Concept | Abbreviation | Full Notation |
| --- | --- | --- |
| Treatment | $A$ | $A_t$ |
| Outcome | $Y$ | $Y_t$ |
| Time-Varying Covariates | $X$ | $X_t$ |
| Time-invariant Covariates | $V$ | $V$ |
| Selective Parameters | $B, C$ | $B, C$ |
| Projection Parameters | $R$ | $R$ |
| Discrete Time Stamp | $\Delta$ | $\Delta$ |
| Covariance Matrix | $\Sigma$ | $\Sigma$ |
| Hidden State of Mamba | $h$ | $h_t$ |
| Time Horizon | $T$ | $T$ |

## A.2 DERIVATION OF LOSS FUNCTION

**Proposition 1.** *Finding $\boldsymbol{K}_i$ to minimize equation 4 is equivalent to minimizing $\boldsymbol{K}_i \Sigma_{\tilde{X}_i^h, a_t} \Sigma_{\tilde{X}_i^h, a_t}^T = \boldsymbol{0}$ for each $1 \leq i \leq t-1$.*

*Proof.* Recalling the expression $\|\boldsymbol{K}_i \Sigma_{\tilde{X}_t^h, a_t}\|_2^2$, we only need to prove that $\Sigma_{\tilde{X}_i^h, a_t} \Sigma_{\tilde{X}_i^h, a_t}^T \boldsymbol{K}_i = \boldsymbol{0}$ is the condition to achieve minimization of $\boldsymbol{K}_i \Sigma_{\tilde{X}_t^h, a_t}$. First, we show that, due to the fact that $\Sigma_{i,t,h} = \Sigma_{\tilde{X}_i^h, a_t} \Sigma_{\tilde{X}_i^h, a_t}^T$ is semi-positive, the term $\mathrm{Tr}\left(\boldsymbol{K}\Sigma_{i,t,h}\boldsymbol{K}\right)$ is a convex function such that deriving the derivative w.r.t. $\mathrm{Tr}\left(\boldsymbol{K}\Sigma_{i,t,h}\boldsymbol{K}^T\right)$ coincides with the minimization of $\mathrm{Tr}\left(\boldsymbol{K}\Sigma_{i,t,h}\boldsymbol{K}^T\right)$:

$$
\begin{aligned}
&\alpha\mathrm{Tr}\left(\boldsymbol{K_1}\Sigma_{i,t,h}\boldsymbol{K_1}^T\right) + \beta\mathrm{Tr}\left(\boldsymbol{K_2}\Sigma_{i,t,h}\boldsymbol{K_2}^T\right) - \mathrm{Tr}\left(\left(\alpha\boldsymbol{K_1} + \beta\boldsymbol{K_2}\right)\Sigma_{i,t,h}\left(\alpha\boldsymbol{K_1} + \beta\boldsymbol{K_2}\right)^T\right), \\
&= \alpha\beta\mathrm{Tr}\left(\left(\boldsymbol{K_1} - \boldsymbol{K_2}\right)\Sigma_{i,t,h}\left(\boldsymbol{K_1} - \boldsymbol{K_2}\right)^T\right), \\
&\geq 0,
\end{aligned} \tag{13}
$$

where final equation is due to the fact that $\Sigma_{i,t,h}$ is semi-positive. Since the term $\mathrm{Tr}\left(\boldsymbol{K}\Sigma_{i,t,h}\boldsymbol{K}\right)$ is convex w.r.t. $\boldsymbol{K}$, we then derive the derivative w.r.t. $\boldsymbol{K}$ as follows:

$$
\frac{\partial\mathrm{Tr}\left(\boldsymbol{K}\Sigma_{i,t,h}\boldsymbol{K}^T\right)}{\partial\boldsymbol{K}} = \boldsymbol{K}\Sigma_{i,t,h}. \tag{14}
$$

Then let $\boldsymbol{K} = \boldsymbol{K}_i$, and our claim follows. $\square$

## A.3 THEORETICAL ANALYSIS

We consider the historical covariates $\overline{X}_{t-1}$, treatment $A_{t-1}$, and outcome $Y_t$ in the last timestamp but omit the time index for clarity. Assuming the binary treatment with $A \in 0, 1$, we first introduces several basic definitions with lemmas as follows:

**Definition 2** (Expected Prediction Error). *Taking the common protocols Shalit et al. (2017); Bica et al. (2020); Melnychuk et al. (2022) that the counterfactual prediction models can be decomposed into representation layers $\Phi$ and prediction layers $W$, we introduce the definition of prediction error in each treatment arm as follows:*

$$
\begin{aligned}
\epsilon_F^{a=1}(W, \Phi) &= \int_{\mathcal{X}} \ell_{h,\Phi}(x, 1)p(X \mid A = 1)dx \\
\epsilon_F^{a=0}(W, \Phi) &= \int_{\mathcal{X}} \ell_{h,\Phi}(x, 0)p^{a=0}(x)dx,
\end{aligned} \tag{15}
$$

*where $\int_{\mathcal{Y}} L(y, W(\Phi(x), a)) \Pr(T(a) = y \mid X = x)dy$ refers to the individualized factual prediction error, and $L$ refers to the prediction loss per sample, e.g., MSE loss.*

Then we introduce an existing fundamental lemma as follows:

**Lemma 1** (Risk Bounds from Shalit et al. (2017)).

$$\epsilon_{ECP}(W, \Phi) \leq 2\left(\epsilon_F^{a=0}(W, \Phi) + \epsilon_F^{a=1}(W, \Phi) + v\textbf{\textit{IPM}}_{\text{G}}\left(p_\Phi^{a=1}, p_\Phi^{a=0}\right) + C\right), \tag{16}$$

*where $v$ and $C$ are constants w.r.t. model parameters, the integral probability metric (IPM), i.e., $\text{IPM}_G(u, v) \triangleq \sup_{m \in G} \int_{\mathcal{R}} m(r)[u(r) - v(r)]dr$ Shalit et al. (2017) defines the probability divergence between two probability measures $u, v$ with the aid of a family of functions $G$.*

Then we first illustrate an immediate lemma which is necessary for our following derivation:

**Lemma 2** (Empirical Prediction Error). *We first define empirical prediction error for predicting the factual outcomes in each treatment arm as follows:*

$$\epsilon_F^{a=1}(\hat{W}, \Phi) = \frac{1}{N_{a=1}} \sum_{i=1}^{N_1} L(y_i, W(\Phi(x_i), a = 1))$$
$$\epsilon_F^{a=0}(\hat{W}, \Phi) = \frac{1}{N_{a=0}} \sum_{i=1}^{N_0} L(y_i, W(\Phi(x_i), a = 0)). \tag{17}$$

*Assuming that the variance of the loss function is bounded, i.e., $\text{Var}(l_{a=0}) \leq \kappa_0$ and $\text{Var}(l_{a=1}) \leq \kappa_1$, we then have the following inequalities hold with probability at least $1 - \eta$, respectively:*

$$|\widehat{\epsilon_F^{a=1}(W, \Phi)} - \epsilon_F^{a=1}(W, \Phi)| \leq \sqrt{\frac{\kappa_1}{\eta}}$$
$$|\widehat{\epsilon_F^{a=0}(W, \Phi)} - \epsilon_F^{a=0}(W, \Phi)| \leq \sqrt{\frac{\kappa_0}{\eta}} \tag{18}$$

*Proof.* The resulting inequalities is immediately obtained by invoking the Chebyshev's Inequality w.r.t. each empirical prediction error term. □

**Lemma 3** (Equivalence between De-correlation and First-moment Matching). *We state that, our proposed CDSP method, i.e., $Cov(h, A) = 0$, is equivalent to matching first-order moment of historical covariates across the treatment arms.*

*Proof.* We first have:

$$\begin{aligned}
Cov(\Phi_x, A) &= \mathbb{E}[\Phi_x A] - P(A = 1)\mathbb{E}[\Phi_x] \\
&= \mathbb{E}[\Phi_x A \mid A = 1]P(A = 1) - P(A = 1)\mathbb{E}[\Phi_x] \\
&= \left(\mathbb{E}[\Phi_x \mid A = 1] - \mathbb{E}[\Phi_x]\right)P(A = 1) \\
&= \left(\mathbb{E}[\Phi_x \mid A = 1] - \left(\mathbb{E}[\Phi_x \mid A = 1]P(A = 1) + \mathbb{E}[\Phi_x \mid A = 0]P(A = 0)\right)\right)P(A = 1) \\
&= \left(\mathbb{E}[\Phi_x \mid A = 1] - \mathbb{E}[\Phi_x \mid A = 0]\right)P(A = 1)P(A = 0)
\end{aligned}, \tag{19}$$

Then our claim follows. □

**Lemma 4** (Construction of Global Minima of Adversarial Balancing). *Based on the sufficient and necessary condition in Theorem 4.1 in (Melnychuk et al., 2022), a treatment-invariant representation $\Phi_x$ that $\Phi_x A$ achieves the global minima of traditional adversarial balancing (ADB) modules.*

**Lemma 5** (Construction of Global Minima of Our CDSP). *By contrast, any representation $\Phi_x$ that matches the first-order of historical covariates of our proposed CDSP regularization term achieves the global minima.*

**Theorem 1.** *Let $W^a$ and $\Phi$ denotes model parameters of prediction models without any balancing modules, $\tilde{W}^a$ and $\tilde{\Phi}$ denotes model parameters of prediction models with adversarial balancing modules, and $\hat{W}^a$ and $\hat{\Phi}$ denotes model parameters of prediction models with our CDSP modules. We further assume that:*

*(a) Gaussian covariates per group such that $\overline{X}_{|a} \sim \mathcal{N}(\mu_a, \sigma_a I)$ for $a \in 0, 1$.*

(b) *Linear structure of $Y$, i.e., $Y(a) = W^a \Phi \overline{X}_{|a} + \epsilon^a$, where $\epsilon^a \sim N(0, 1)$, $\mathbb{E}[{\epsilon^0}^4] \leq \kappa_2$ and $\mathbb{E}[{\epsilon^1}^4] \leq \kappa_3$. Such an assumption is rationale, as introducing the model mis-specification error into prediction models with and without balancing modules equally will not affect our analysis.*

(c) *The representation layers to embed historical covariates are linear, e.g., linear transformer (Von Oswald et al., 2023) and linear state-space models (Zhang et al., 2023).*

(d) *The prediction layers of $Y$ are implemented with linear regression based on representation, i.e., the OLS regression.*

(e) *The scaled loss, i.e., $\frac{1}{v}L$, belongs to the set of all couplings between $p_\Phi^{a=1}$ and $p_\Phi^{a=0}$.*

(f) *The representation matrix $\Phi$ has bounded operator norm, i.e., $\|\Phi\| \leq r_1$ and bounded F-norm, i.e., $\|\Phi\|_F \leq r_2$.*

(g) *The prediction parameters $W^a, \hat{W}^a, \tilde{W}^a$ has bounded 2-norm as $r_3$.*

*Then: (1) the following risk bound in the finite sample regime holds for the vanilla prediction cases* **without any balancing modules** *with probability $1 - 2\eta$:*

$$\epsilon_{\mathrm{ECP}}^{vanilla}(W, \Phi) \leq 2 \left( v r_1^2 \|(\mu_1 - \mu_0)\|_2^2 + v(\sqrt{\sigma_1} - \sqrt{\sigma_0})r_2^2 + C - \left( \sqrt{\frac{1}{\eta}}(\sum_{j=1}^{2} \sqrt{\kappa_j}) + \overline{\sigma} \right) \right),$$
(20)

where $\overline{\sigma} = \frac{1}{N_1} \sum_{i=1}^{N_1} \epsilon_i^{a\,2}$.

*Proof.* As linear transformations preserves the Gaussian distributions, the representations of $X$ still follows the Gaussian distributions: $\overline{X}_{|a} \sim \mathcal{N}(\Phi\mu_a, \sigma_a \Phi^T \Phi)$. Then by invoking the Lemma 2 with the fact that the 1-Wasserstein distance is governed by the 2-Wasserstein distance (Jensen's Inequality), we have that:

$$\epsilon_{\mathrm{ECP}}(W, \Phi)$$
$$\leq 2 \left( \epsilon_F^{a=0}(W, \Phi) + \epsilon_F^{a=1}(W, \Phi) + v\mathrm{Wass}_2 \left( p_\Phi^{a=1}, p_\Phi^{a=0} \right) + C \right)$$
$$= 2 \left( \epsilon_F^{a=0}(W, \Phi) + \epsilon_F^{a=1}(W, \Phi) + v \left( \|\Phi(\mu_1 - \mu_0)\|_2^2 + \|(\sqrt{\sigma_1} - \sqrt{\sigma_0}) \left( \Phi^T \Phi \right)^{1/2} \|_F^2 \right) + C \right)$$
$$\leq 2 \left( \epsilon_F^{a=0}(W, \Phi) + \epsilon_F^{a=1}(W, \Phi) + v \left( r_1^2 \|(\mu_1 - \mu_0)\|_2^2 + (\sqrt{\sigma_1} - \sqrt{\sigma_0})\|\Phi\|_F^2 \right) + C \right)$$
$$\leq 2 \left( \epsilon_F^{a=0}(W, \Phi) + \epsilon_F^{a=1}(W, \Phi) + v \left( r_1^2 \|(\mu_1 - \mu_0)\|_2^2 + (\sqrt{\sigma_1} - \sqrt{\sigma_0})r_2^2 \right) + C \right),$$
(21)

where the first equality is due to the closed-form expression of 2-Wasserstein distance between Gaussian distributions, and the second inequality is due to the fact that $\Phi^T \Phi$ is symmetric and semi-positive definite. We further invoke Lemma 3 and substitute the expected prediction error terms with empirical prediction error terms, and the following inequality holds with probability at least $1 - 2\eta$:

$$\epsilon_{\mathrm{ECP}}(W, \Phi) \leq 2 \left( \epsilon_F^{a=0}(W, \Phi) + \epsilon_F^{a=1}(W, \Phi) + v \left( r_1^2 \|(\mu_1 - \mu_0)\|_2^2 + (\sqrt{\sigma_1} - \sqrt{\sigma_0})r_2^2 \right) + C \right)$$
$$\leq 2 \left( \widehat{\epsilon_F^{a=1}(W}, \Phi) + \widehat{\epsilon_F^{a=0}(W}, \Phi) + v \left( r_1^2 \|(\mu_1 - \mu_0)\|_2^2 + (\sqrt{\sigma_1} - \sqrt{\sigma_0})r_2^2 \right) + C - \left( \sqrt{\frac{\kappa_0}{\eta}} + \sqrt{\frac{\kappa_1}{\eta}} \right) \right).$$

$\square$

(2) the following risk bound in the finite sample regime holds for the vanilla prediction cases **with adversarial balancing modules** with probability $1 - 2\eta$:

$$\epsilon_{\mathrm{ECP}}^{adb}(\tilde{W}, \tilde{\Phi}) \leq 2 \left( \frac{(2 + \sigma_0 + \sigma_1)(r_1 r_3)^2}{4} \|\mu_1 - \mu_2\|_2^2 + C - \left( \sqrt{\frac{1}{\eta}}(\sum_{j=1}^{2} \sqrt{\kappa_j}) + \overline{\sigma} \right) \right), \quad (22)$$

where $\overline{\sigma} = \frac{1}{N_1} \sum_{i=1}^{N_1} \epsilon_i^{a\,2}$.

*Proof.* Based on Lemma 4, we further construct a representation mapping, i.e., $\Phi$, as a global minima of **adversarial balancing methods** including (Melnychuk et al., 2022; Bica et al., 2020; Wang et al., 2024a) as follows:

$$\tilde{\Phi}(\overline{x}_{|A=1}) = \sqrt{\sigma_0}\Phi(\overline{x}_{|1} - \frac{\mu_1 - \mu_0}{2})$$

$$\tilde{\Phi}(\overline{x}_{|A=0}) = \sqrt{\sigma_1}\Phi(\overline{x}_{|0} - \frac{\mu_0 - \mu_1}{2}) \tag{23}$$

such that the resulting distribution are treatment-invariant and the global minima is achieved. We then check the factual prediction term as follows:

$$\epsilon_F^{\widehat{a=1}}(\widehat{W}, \Phi) = \frac{1}{N_1}\sum_{i=1}^{N_1}\|W^1\Phi\overline{x}_{i|1} + \epsilon_i^{a=1} - \sqrt{\sigma_0}\hat{W}^1\Phi(\overline{x}_{|1} - \frac{\mu_1 - \mu_0}{2})\|_2^2$$

$$\leq \frac{1}{N_1}\sum_{i=1}^{N_1}\left((\epsilon_i^{a=1})^2 + \frac{1}{4}\|\sqrt{\sigma_0}\tilde{W}^1 - W^1\|_2^2\|\Phi\|^2\|\mu_1 - \mu_2\|_2^2\right). \tag{24}$$

$$\leq \frac{1}{N_1}\sum_{i=1}^{N_1}\left((\epsilon_i^{a=1})^2 + \frac{(1+\sigma_0)(r_1r_3)^2}{4}\|\mu_1 - \mu_2\|_2^2\right).$$

On the other hand, the divergence term equals to zero:

$$v\text{Wass}_2\left(p_{\tilde{\Phi}}^{a=1}, p_{\tilde{\Phi}}^{a=0}\right) = 0, \tag{25}$$

then our claim follows. □

(3) the following risk bound in the finite sample regime holds for the vanilla prediction cases **with our proposed CDSP module** with probability $1 - 2\eta$:

$$\epsilon_{\text{ECP}}^{cdsp}(\tilde{W}, \tilde{\Phi}) \leq 2\left(\frac{(r_1r_3)^2}{2}\|\mu_1 - \mu_2\|_2^2 + C - \left(\sqrt{\frac{1}{\eta}}(\sum_{j=1}^2\sqrt{\kappa_j}) + \overline{\sigma}\right)\right), \tag{26}$$

where $\overline{\sigma} = \frac{1}{N_1}\sum_{i=1}^{N_1}\epsilon_i^{a\,2}$.

*Proof.* Based on Lemma 5, we further construct a representation mapping, i.e., $\Phi$, as a global minima of our proposed CDSP method as follows:

$$\widehat{\Phi}(\overline{x}_{|A=1}) = \Phi(\overline{x}_{|1} - \frac{\mu_1 - \mu_0}{2})$$

$$\widehat{\Phi}(\overline{x}_{|A=0}) = \Phi(\overline{x}_{|0} - \frac{\mu_0 - \mu_1}{2}) \tag{27}$$

**We note that in order to achieve global minima, our CDSP only requires to align the mean across different treatment groups without aligning the variance terms.** We then derive the divergence term as follows:

$$v\text{Wass}_2\left(p_{\widehat{\Phi}}^{a=1}, p_{\widehat{\Phi}}^{a=0}\right) \leq v\left((\sqrt{\sigma_1} - \sqrt{\sigma_0})r_2^2\right), \tag{28}$$

and the prediction error terms as derived as follows:

$$\epsilon_F^{\widehat{a=1}}(\widehat{W}, \Phi) = \frac{1}{N_1}\sum_{i=1}^{N_1}\|W^1\Phi\overline{x}_{i|1} + \epsilon_i^{a=1} - \widehat{W^1}\Phi(\overline{x}_{|1} - \frac{\mu_1 - \mu_0}{2})\|_2^2$$

$$\leq \frac{1}{N_1}\sum_{i=1}^{N_1}\left((\epsilon_i^{a=1})^2 + \frac{1}{4}\|\widehat{W^1} - W^1\|_2^2\|\Phi\|^2\|\mu_1 - \mu_2\|_2^2\right). \tag{29}$$

$$\leq \frac{1}{N_1}\sum_{i=1}^{N_1}\left((\epsilon_i^{a=1})^2 + \frac{(r_1r_3)^2}{4}\|\mu_1 - \mu_2\|_2^2\right),$$

and our claim follows. □

**Proposition 2. [Over-balancing of Adversarial Learning]** *When the following condition holds, the risk bound of vanilla Empirical Risk Minimization (ERM) model is more tighter than that of adversarial balancing methods:*

$$\|\mu_1 - \mu_0\|_2^2 \leq \frac{v(\sqrt{\sigma_1} - \sqrt{\sigma_0})^2 r_2^2}{vr_1^2 - (2 + \sigma_1 + \sigma_0)r_1^2 r_3^2/4}. \tag{30}$$

*Remark.* The above conclusion is very intuitive, as the gap between two groups decreases, the harm of over-balancing is much larger than that of observed confounding bias.

**Proposition 3. [Why CDSP outperforms ADB]** *Our CDSP method always outperforms ABD methods with a tighter bound.*

*Proof.* The result is immediate from the fact that:

$$\frac{(r_1 r_3)^2}{2}\|\mu_1 - \mu_2\|_2^2 \leq \frac{(2 + \sigma_0 + \sigma_1)(r_1 r_3)^2}{4}\|\mu_1 - \mu_2\|_2^2 \tag{31}$$

always holds. □

*Remark.* The above conclusion is also very intuitive. Especially, on the factual outcome prediction tasks, i.e., real-world datasets, our CDSP outperforms CT with a large margin.

*Remark.* Our first conclusion is intuitive, as the gap between two groups decreases, the harm of over-balancing is much larger than that of observed confounding bias. The second conclusion reflects an important idea hidden behind our CDSP: compared with optimal representations achieved by adb methods, e.g., $\mathcal{N}(\mu_1, \sigma_1 I)$ and $\mathcal{N}(\mu_0, \sigma_0 I)$ are aligned via representation layers at both means and covariances, our proposed CDSP only aligns the means of two Gaussian representations with each other, i.e., $\mu_0$ and $\mu_1$ after some transformations. Subsequently, when performing prediction, the extent of the loss of predictive information embodied in covariates for our method will be smaller than that of adb methods. On the other hand, the effect of debiasing observed confounding remains nearly the same when comparing our methods with adb methods. Hence, our de-correltaion method achieves lower risk with tighter bound.

## A.4 EXTRA EXPERIMENTAL DETAILS

### A.4.1 BENCHMARK DETAILS

**Tumor Simulator.** As the state-of-the-art cancer effect simulator, the tumor growth (TG) model from (Geng et al., 2017) characterizes the volume of tumor $\mathbf{Y}_{t+1}$ for $t + 1$ days after cancer diagnosis. More specifically, the TG simulator contains the following components:

- Two binary treatments: (1) radiotherapy ($\mathbf{A}_t^r$) when assigned to a patient has an immediate effect $d(t)$ on the next outcome; (2) chemotherapy ($\mathbf{A}_t^c$), which affects future outcomes in an exponentially decaying approach scaled by $C(t)$;

- One-dimensional Outcome, i.e., the volume of tumor $\mathbf{Y}_{t+1}$. More specifically, $\mathbf{Y}_{t+1}$ is affected by $\mathbf{A}_t^c$ and $\mathbf{A}_t^r$ via the following equation:

$$\mathbf{Y}_{t+1} = \left(1 + \rho \log\left(\frac{K}{\mathbf{Y}_t}\right) - \beta_c C_t - (\alpha_r d_t + \beta_r d_t^2) + \varepsilon_t\right)\mathbf{Y}_t, \tag{32}$$

  where $\rho, K, \beta_c, \alpha_r, \beta_r, \varepsilon_t \sim N(0, 0.01^2)$ are simulation parameters, and $\beta_c, \alpha_r, \beta_r$ describe the individual response of each patient.

- Time-varying confounding effect. By treating the past outcomes as the confounders, current treatment will be assigned via a biased approcah:

$$\mathbf{A}_t^c, \mathbf{A}_t^r \sim \text{Bernoulli}\left(\sigma\left(\frac{\gamma}{D_{\max}}(\bar{D}_{15}(\bar{\mathbf{Y}}_{t-1}) - D_{\max}/2)\right)\right), \tag{33}$$

  where $\sigma(\cdot)$ is the sigmoid function, $D_{\max}$ is the maximum tumor diameter, $\bar{D}_{15}(\bar{\mathbf{Y}}_{t-1})$ and $\gamma$ are hyper-parameters.

Overall, the critical parameter in TG simulator, i.e., $\gamma$, controls the degree of confounding effect such that the confounding bias is controlled. With increasing values of $\gamma$, the the amount of time-varying confounding bias enlarges.

Finally, during the testing phase, we simulate counterfactual trajectories separately for one-step prediction and $\tau$-step prediction. For the former, we simulate all four possible combinations of $\mathbf{Y}_{t+1}$. Following protocols in (Melnychuk et al., 2022), we adopt two evaluation protocols to test $\tau$-step prediction, i.e., *Single sliding treatment* and *Random trajectories*. To be specific, single sliding treatment simulates a single treatment, where the treatments are iteratively moved over a window ranging from $t$ to $t + \tau_{\max} - 1$. Meanwhile, random trajectories are simulated as a fixed number, i.e., $2(\tau_{\max} - 1)$, of randomly sampled trajectories.

Besides, for each $\gamma$, we simulate 10,000 patients for training, 1,000 for validation, and 1,000 for testing. The length for sequence is limited to 60 time steps.

**Semi-synthetic Dataset.** Following protocols (Melnychuk et al., 2022), we construct the semi-synthetic dataset based on the real-world data extracted from the MIMIC-extract (Wang et al., 2020) with a standardized preprocessing pipeline in (Johnson et al., 2016). As done in (Melnychuk et al., 2022), we overcome missing values with forward and backward filling, and perform Standardization of all the continuous time-varying features. Following protocols, we extract 25 different vital signs, i.e., time-varying covariates with 3 static covariates (gender, ethnicity, and age), and encode all static covariates in the one-hot approach. Overall, the resulting dimension of features is 44.

By extending the idea in (Schulam & Saria, 2017), the semi-synthetic simulation protocols in (Melnychuk et al., 2022) first generate untreated trajectories of outcomes under endogeneous and exogeneous dependencies and, then, sequentially apply treatments to the trajectory[2]:

- 1,000 patients are randomly selected, with ICU stays lasting at least 20 hours. For these patients, their ICU stays are limited to a maximum of 100 hours such that the length of stay $(T^{(i)})$ is between 20 and 100 hours.

- $d_y$ untreated outcomes $\mathbf{Z}_t^{j,(i)}, j = 1 \dots, d_y$ is simulated for each patient $i$ from the cohort:

$$\mathbf{Z}_t^{j,(i)} = \underbrace{\alpha_S^j \, \text{B-spline}(t) + \alpha_g^j \, g^{j,(i)}(t)}_{\text{endogenous}} + \underbrace{\alpha_f^j \, f_Z^j(\mathbf{X}_t^{(i)})}_{\text{exogenous}} + \underbrace{\varepsilon_t}_{\text{noise}} \tag{34}$$

  with $\varepsilon_t \sim N(0, 0.005^2)$, and $\alpha_S^j, \alpha_g^j$, and $\alpha_f^j$ are weight parameters. Meanwhile, B-spline$(t)$ is sampled from a mixture of three cubic splines, $g^{j,(i)}(\cdot)$ is sampled independently for each patient from Gaussian process with Matérn kernel; and $f_Z^j(\cdot)$ is sampled from a random Fourier features (RFF) approximation of an Gaussian process.

- The $d_a$ binary treatments $\mathbf{A}_t^l, l = 1, \dots, d_a$, are simulated in a sequential approach:

$$p_{\mathbf{A}_t^l} = \sigma\left(\gamma_A^l \bar{A}_{T_l}(\bar{\mathbf{Y}}_{t-1}) + \gamma_X^l f_Y^l(\mathbf{X}_t) + b_l\right), \tag{35}$$

$$\mathbf{A}_t^l \sim \text{Bernoulli}\left(p_{\mathbf{A}_t^l}\right), \tag{36}$$

  where the random function $f_Y^l(\mathbf{X}_t)$[3] are treated as the confounding effect, and $\sigma(\cdot)$ refers to the sigmoid activation, $\gamma_A^l$ and $\gamma_X^l$ are confounding parameters, $b_l$ is a fixed bias.

- The outcome is generated via the addition of the untreated outcome $\mathbf{Z}_t^j$ and the simulated treatment effect $E^j(t)$:

$$\mathbf{Y}_t^j = \mathbf{Z}_t^j + E^j(t), \tag{37}$$

  where $\mathbf{Z}_t^j$ characterizes the effect of the treatment bias within a time window:

$$E^j(t) = \sum_{i=t-w^l}^{t} \frac{\min_{l=1,\dots,d_a} \mathbb{1}_{[\mathbf{A}_i^l=1]} p_{\mathbf{A}_i^l} \beta_{lj}}{(w^l - i)^2}, \tag{38}$$

  where $\beta_{lj}$ is the maximum effect size of treatment $l$.

---

[2]We refer detailed introduction of simulation protocols in (Melnychuk et al., 2022)

[3]$f_Y^l(\cdot)$ is sampled from an RFF approximation of a Gaussian process (similar to $f_Z^j(\cdot)$) (Melnychuk et al., 2022).

By setting $d_a = 3$ and $d_y = 2$, the cohort of 1,000 patients is split into train/validation/test subsets via a ratio of 60% / 20% / 20 %.

**MIMIC-III Real-world Dataset.** Following protocols in (Melnychuk et al., 2022; Huang et al., 2024), we also adopt the Standardized pre-processing pipeline (Johnson et al., 2016) with forward and backward filling for missing values. Keeping the same as in semi-synthetic data, the features, i.e., the potential confounders, contain 25 vital signs ($d_x = 25$) and the 3 static features. Two binary treatments ($d_a = 2$): vasopressors and mechanical ventilation are adopted. The diastolic blood pressure is the outcome to be estimated. A cohort of 5,000 patients is randomly selected from the patients with intensive care unit (ICU) stays of at least 30 hours.

The train/validation/test subsets are split with the ratio of 70%/15%/15%. For one-step prediction, all samples are adopted in the testing data. For multiple-step prediction with $\tau \geq 2$, we choose patients with observation lengths of at least 6 with a rolling origin.

**M5 Real-world Dataset.** The M5 Forecasting dataset, as cited in (Huang et al., 2024), comprises daily transaction data from Walmart stores across three U.S. states, along with comprehensive details on products, stores, pricing, and significant events. For our analysis, we repurpose this dataset to estimate treatment effects, designating product pricing as the treatment variable and product sales as the outcome variable. All other features are treated as covariates. Since this dataset is derived from real-world observations and lacks counterfactual data, we evaluate the performance of various models, including their Empirical Risk Minimization (ERM) variants, in predicting factual outcomes. Notably, GENT and MSM models are excluded from this analysis due to convergence issues with this dataset.

### A.4.2 BASELINE DETAILS

We categorize the previous baseline methods based on their architectures:

**Transformer:** The Causal Transformer (CT) method is proposed in (Melnychuk et al., 2022). Three sub-networks are constructed in CT, with cross-attention to capture interactions among predictors. For confounding bias, the domain confusion objective is proposed by adversarially training a domain predictor such that the learned representations of historical covariates are only predictive of the outcome.

**LSTM:**

- G-Net (Li et al., 2021) A deep network version of the G-computation, which estimates the following G-formula (Robins, 1986):

$$\mathbb{E}\left(\mathbf{Y}_{t+\tau}[\bar{\mathbf{a}}_{t:t+\tau-1}] \mid \bar{\mathbf{H}}_t\right)$$
$$= \int_{\mathbb{R}^{d_x} \times \cdots \times \mathbb{R}^{d_x}} \mathbb{E}\left(\mathbf{Y}_{t+\tau} \mid \bar{\mathbf{H}}_t, \bar{\mathbf{x}}_{t+1:t+\tau-1}, \bar{\mathbf{y}}_{t+1:t+\tau-1}, \bar{\mathbf{a}}_{t:t+\tau-1}\right) \times$$
$$\prod_{j=t+1}^{t+\tau-1} \mathbb{P}\left(\mathbf{x}_j\mathbf{y}_j \mid \bar{\mathbf{H}}_t, \bar{\mathbf{x}}_{t+1:j-1}, \bar{\mathbf{y}}_{t+1:j-1}, \bar{\mathbf{a}}_{t:j-1}\right) \, d\bar{\mathbf{x}}_{t+1:t+\tau-1} d\bar{\mathbf{y}}_{t+1:t+\tau-1}, \tag{39}$$

  where the final estimate is given by first sampling $\mathbb{P}(\mathbf{X}_j \mid \bar{\mathbf{H}}_t, \bar{\mathbf{x}}_{t+1:j-1}, \bar{\mathbf{a}}_{t:j-1})$ from Monte-Carlo, and then $\mathbf{Y}_{t+\tau}[\bar{\mathbf{a}}_{t,t+\tau-1}]$ is taken as the empirical mean (Melnychuk et al., 2022).

- Counterfactual Recurrent Network (CRN) deploys a single LSTM-layer-based decoder and an encoder. To remove the confounding bias, the gradient adversarial training is adopted to confuse the covariate classifier so that the resulting representations only inform the outcome.

- Recurrent Marginal Structural Networks (RMSNs) is also an LSTM-based deep estimation model by incorporating the inverse probability of treatment weights (IPTW) (Robins et al., 2000) re-weighting method for bias correction. To be specific, the stabilized weights, i.e., $f\left(\mathbf{A}_n \mid \bar{\mathbf{A}}_{n-1}\right)$ and $f\left(\mathbf{A}_n \mid \bar{\mathbf{H}}_n\right)$, are estimated via LSTM networks.

**Shallow Models:** Marginal Structural Models (MSMs) (Robins et al., 2000) is the linear realization of the IPTW. Similar to RMSNs, the stabilized weights are learned via the linear-logistic regression.

**Mamba-HSIC:** When projecting the three-dimensional tensors from Mamba, i.e., $h_{t-1}$ (sample index, time index, feature index), onto the two-dimensional tensor for final prediction, we extract the representations from the final fully-connected prediction layer, denoted as $h_{t-1}^{HSIC}$, to perform the HSIC regularization. To be specific, we perform HSIC regularization through the following equation:

$$\text{HSIC}(h_{t-1}^{HSIC}, a_t) = \text{Tr}\left(K_{h_{t-1}^{HSIC}} J K_{a_t} J\right), \tag{40}$$

where $J = I - 1/L$ ($L$ is the common dimension shared by $h_{t-1}^{HSIC}$ and $a_t$), $K$ is the kernel Gram Matrix (Gretton et al., 2007). We here deploy the standard Gaussian Kernel throughout our experiment.

### A.4.3 Computational Complexity Analysis

We perform theoretical analysis on computational complexity by comparing our proposed CDSP with previous adversarial balancing methods. Assuming the overall length of the time horizon is $T$, and the representation dimension of $Y$, $A$ and $X$ are $c$, $d_A^h$ and $d_X^h$, respectively. Moreover, for adversarial balancing modules, previous practice shows that the discriminator has usually 2 layers with $d_X^h$ for each layer.

For our proposed CDSP methods, the complexity of the debias process can be divided into four stages:

(1) Pre-computing the covariance terms, i.e., $\Sigma_{\tilde{X}_i^h, a_t} \Sigma_{\tilde{X}_i^h, a_t}^T$, before the beginning of the training process. Due to the complexity of matrices multiplication, we derive the complexity of this component as $\mathcal{O}(B((d_X^h)^2 + d_A^h)T)$;

(2) Computing the accumulated selective parameters $\boldsymbol{K}_i = \overline{B}_i \Pi_{j=i}^{t-1} \overline{A}_j$, which owns the complexity as $\mathcal{O}(B((d_X^h)^2)T)$;

(3) Computing the multiplications between $\Sigma_{\tilde{X}_i^h, a_t} \Sigma_{\tilde{X}_i^h, a_t}^T$ and $\boldsymbol{K}_i = \overline{B}_i \Pi_{j=i}^{t-1} \overline{A}_j$, which owns the complexity as $\mathcal{O}(B(d_X^h)^3 T)$;

(4) Computing the 2-norm of the matrix as $\mathcal{O}(B(d_X^h)^3 T)$.

Hence, the overall complexity of our method can be derived as $\mathcal{O}(B((d_X^h)^3 + d_A^h)T)$.

For traditional adversarial balancing methods, the complexity can be divided into three parts:

(1) Embedding representations of $X$ and $A$ with two layers of matrix multiplications, which owns the complexity of $\mathcal{O}(B((d_X^h)^3 + (d_X^h)^2 d_Y^h)T)$ in each round;

(2) Computing the softmax and the cross-entropy loss, which owns the complexity of $\mathcal{O}(B|A|T)$, where $|A|$ are the cardinal number of $A$.

Hence, the overall complexity of ADB can be derived as $\mathcal{O}(B((d_X^h)^3 + (d_X^h)^2 d_Y^h + |A|)T)$.

### A.5 Implementation Details

### A.5.1 Implementation on Model Parameters

**Details on our method.** We detail all the parameters, including the model layers, units of each layer, dropout rate, together with the batch size, EMA, and input-output size in Table 4.

**Details of other baselines.** We exactly follow the hyper-parameter tuning protocols in Table 6 & 7 in (Melnychuk et al., 2022) for each baseline we have compared with throughout our experiments.

### A.5.2 Sensitivity Analysis on Confounding Parameter

Furthermore, to inform how our proposed method performs towards the correction of confounding bias, we supplement detailed experiments on the TG simulator by tuning the confounding parameter, i.e., $\gamma$, and compared the SOTA method, i.e., CT, with our proposed Mamba-CDSP in Table 7. As $\gamma$ is usually set to $0 - 4$ in previous empirical studies Melnychuk et al. (2022); Bica et al. (2020), we

Table 4: Ranges for hyperparameter tuning across experiments. Here, we distinguish (1) data using the tumor growth (TG) simulator (=experiments with fully-synthetic data), (2) data from the semi-synthetic benchmark, and (3) real-world MIMIC-III data. EL refers to the embedding layer, and PL refers to the projection layer.

| Model | Hyperparameter | TG simulator | Semi-Synthetic Data | Real-world Data |
|-------|---------------|--------------|---------------------|-----------------|
| | Mamba blocks ($B$) | 1 | 1 | 2 |
| | Learning rate ($\eta$) | {0.0005, 0.001, 0.01} | {0.0005, 0.001, 0.01} | {0.0005, 0.001, 0.01} |
| | Minibatch size | 128 | 64 | 64 |
| | De-correlation Parameter | 1 | 1 | 1 |
| | EL hidden units ($d_E L$) | 32 | 32 | 64 |
| Mamaba-CDSP | PL hidden units ($d_{\text{PL}}$) | 32 | 32 | 64 |
| | Dropout rate ($p$) | 0.1 | 0.1 | 0.1 |
| | EMA of model weights | 0.99 | 0.99 | 0.99 |
| | Input size | $d_a + d_x + d_y + d_v$ | $d_a + d_x + d_y + d_v$ | $d_a + d_x + d_y + d_v$ |
| | Output size | $d_y$ | $d_y$ | $d_y$ |

increase the value of $\gamma$ to 8,10,16,20 in our analysis such that a much higher confounding effect exists in the TG simulation data. Throughout our comparison, we found that our proposed Mamba-CDSP substantially outperforms CT with a large margin.

### A.5.3 EXTRA EXPERIMENTAL RESULTS

We also provide extra experimental results including: (1) Extra visualization results at $a = 10$ in Fig. 6; (2) Performance under single treatment slide on the TG simulator in Table 10. Further visualization study on the real-world MIMIC-III dataset also validates the above justification, where the representation learned from Mamba-CDSP enjoys a similar distribution to that from the vanilla Mamba model in Fig. 7.

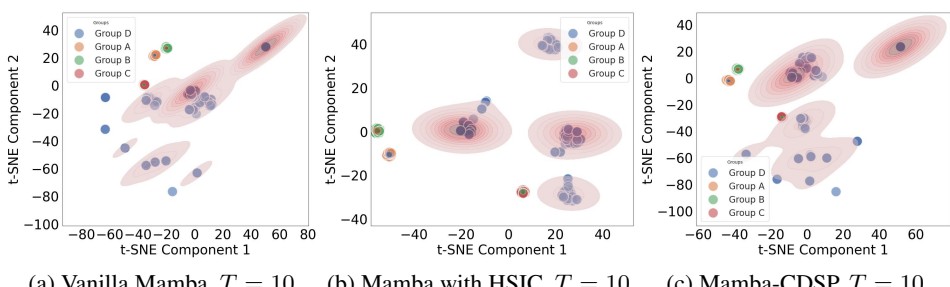

(a) Vanilla Mamba, $T = 10$    (b) Mamba with HSIC, $T = 10$    (c) Mamba-CDSP, $T = 10$

Figure 6: Representation visualizations of Mamba, Mamba with HSIC, and Mamba-CDSP.

### A.5.4 COMPARISON WITH ODE-BASED ESTIMATOR

Furthermore, we also note the existence of the application of Ordinary differential equations (ODE) on the problem of TCP, i.e., the Insite method in Holt et al. (2024). However, we also note that due to the limit the overall time-series process can be characterized using an ODE system Holt et al. (2024). Hence, we only compare it with our proposed Mamba-CDSP on the TG simulation benchmark (the only overlap between Insite Holt et al. (2024) and previous expreience Melnychuk et al. (2022); Bica et al. (2020); Li et al. (2020); Huang et al. (2024)). Results are present in Table 11.

### A.5.5 RE-EXAMINATION ON THE BIAS CORRECTION

In this section, we supplement extra experimental results to understand how our proposed CDSP correct the confounding bias, we compare the training loss curves of the HSIC criteria and our CDSP criteria on the real-world dataset. To be specific, as HSIC criteria measures how the covariate distributions shift across different treatment groups, we use the value of HSIC term to inform the confounding

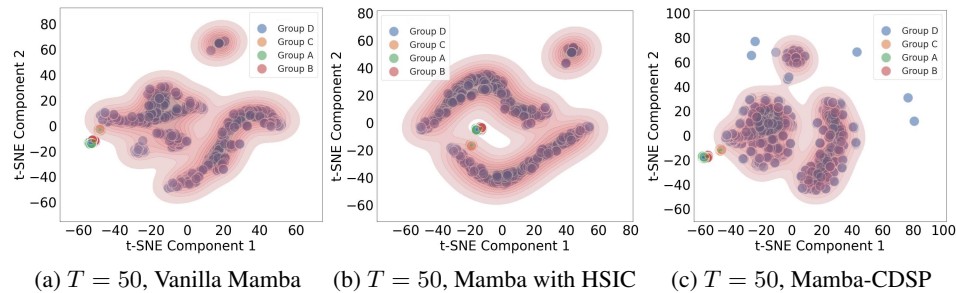

(a) $T = 50$, Vanilla Mamba    (b) $T = 50$, Mamba with HSIC    (c) $T = 50$, Mamba-CDSP

Figure 7: Representation visualizations of Mamba, Mamba with HSIC, and Mamba-CDSP.

Table 5: Ablation study for proposed Mamba-CDSP on semi-synthetic MIMIC-III Data. Reported: normalized RMSE with relative changes. w/ HSIC-Ref refers to the Mamba model debiased by only HSIC.

|  |  | $\tau = 1$ | $\tau = 3$ | $\tau = 5$ | $\tau = 7$ | $\tau = 9$ |
|---|---|---|---|---|---|---|
| Mamba-CDSP (Our proposed) |  | 0.19 | 0.30 | 0.37 | 0.43 | 0.48 |
| Ablation on Model | w/ convolution layer | $-0.01$ | $+0.01$ | $+0.01$ | $+0.04$ | $+0.11$ |
|  | w/o dropout | $+0.08$ | $+0.09$ | $+0.11$ | $+0.09$ | $+0.10$ |
|  | w/ selection mechanism | $\pm 0.05$ | $+0.07$ | $+0.12$ | $+0.16$ | $+0.19$ |
|  | w/o RMS-norm | $+2.13$ | $>10$ | $>10$ | $>10$ | $>10$ |
|  | w/o residual | $-0.03$ | $+0.06$ | $+0.06$ | $+0.06$ | $+0.07$ |
| Ablation on Loss | w/o Cov-Reg ($\alpha = 0$)* | $+0.16$ | $+0.17$ | $+0.16$ | $+0.22$ | $+0.25$ |
|  | w/ HSIC-Reg | $+0.06$ | $+0.08$ | $-0.10$ | $+0.07$ | $-0.07$ |

bias. As shown in Fig. 8, the HSIC term also decreases when our CDSP regularization term decreases. Such a phenomena informs that our method indeed effectively reduces the confounding bias through performing de-correlation.

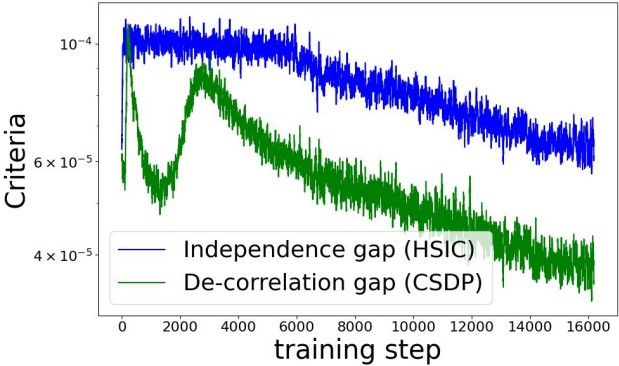

Figure 8: Examination of the Bias Correction on MIMIC-III Dataset.

Table 6: Results for experiments with the real-world M5 data, which are the average performance averaged over five runs with different seeds, and * means statistically significant results (p-value $\leq$ 0.01) using the paired-t-test compared with the best baseline.

| | $\tau = 1$ | $\tau = 2$ | $\tau = 3$ | $\tau = 4$ | $\tau = 5$ | $\tau = 6$ |
|---|---|---|---|---|---|---|
| RMSN | $5.19_{\pm 0.12}$ | $9.73_{\pm 0.26}$ | $10.48_{\pm 0.34}$ | $11.07_{\pm 0.43}$ | $11.63_{\pm 0.53}$ | $12.16_{\pm 0.65}$ |
| CRN | $4.98_{\pm 0.32}$ | $9.15_{\pm 0.17}$ | $9.79_{\pm 0.16}$ | $10.12_{\pm 0.17}$ | $10.38_{\pm 0.19}$ | $10.59_{\pm 0.22}$ |
| CRN w/o balancing | $4.68_{\pm 0.06}$ | $9.05_{\pm 0.16}$ | $9.68_{\pm 0.15}$ | $10.00_{\pm 0.16}$ | $10.26_{\pm 0.18}$ | $10.47_{\pm 0.20}$ |
| CT | $4.59_{\pm 0.08}$ | $8.99_{\pm 0.19}$ | $9.59_{\pm 0.19}$ | $9.91_{\pm 0.23}$ | $10.15_{\pm 0.25}$ | $10.35_{\pm 0.28}$ |
| CT w/o balancing | $4.59_{\pm 0.09}$ | $8.98_{\pm 0.18}$ | $9.58_{\pm 0.18}$ | $9.90_{\pm 0.20}$ | $10.13_{\pm 0.23}$ | $10.34_{\pm 0.26}$ |
| Mamba-CDSP | $\mathbf{4.08^*_{\pm 0.06}}$ | $\mathbf{4.05^*_{\pm 0.13}}$ | $\mathbf{4.59^*_{\pm 0.15}}$ | $\mathbf{5.87^*_{\pm 0.17}}$ | $\mathbf{6.47^*_{\pm 0.20}}$ | $\mathbf{8.39^*_{\pm 0.28}}$ |

Table 7: Sensitivity Analysis on Confounding Factors on the TG Simulator, where results are reported as the mean of 5 different seeds, and * means statistically significant results (p-value $\leq$ 0.01) using the paired-t-test compared with the best baseline.

| $\gamma$ | Model | $\tau = 1$ | $\tau = 2$ | $\tau = 3$ | $\tau = 4$ | $\tau = 5$ |
|---|---|---|---|---|---|---|
| 8 | CT | $2.98_{\pm 0.01}$ | $3.46_{\pm 0.01}$ | $3.56_{\pm 0.01}$ | $3.52_{\pm 0.01}$ | $3.33_{\pm 0.01}$ |
| | Mamba-CDSP | $\mathbf{2.97_{\pm 0.01}}$ | $\mathbf{3.36^*_{\pm 0.01}}$ | $\mathbf{3.29^*_{\pm 0.01}}$ | $\mathbf{3.02^*_{\pm 0.01}}$ | $\mathbf{2.69^*_{\pm 0.01}}$ |
| 10 | CT | $10.46_{\pm 0.01}$ | $9.38_{\pm 0.01}$ | $8.50_{\pm 0.01}$ | $7.63_{\pm 0.01}$ | $6.83_{\pm 0.01}$ |
| | Mamba-CDSP | $\mathbf{4.25^*_{\pm 0.01}}$ | $\mathbf{5.26^*_{\pm 0.01}}$ | $\mathbf{5.58^*_{\pm 0.01}}$ | $\mathbf{5.55^*_{\pm 0.01}}$ | $\mathbf{5.40^*_{\pm 0.01}}$ |
| 16 | CT | $15.93_{\pm 0.01}$ | $13.54_{\pm 0.01}$ | $15.54_{\pm 0.01}$ | $16.40_{\pm 0.01}$ | $16.58_{\pm 0.01}$ |
| | Mamba-CDSP | $\mathbf{8.99^*_{\pm 0.01}}$ | $\mathbf{9.06^*_{\pm 0.01}}$ | $\mathbf{8.09^*_{\pm 0.01}}$ | $\mathbf{6.93^*_{\pm 0.01}}$ | $\mathbf{5.95^*_{\pm 0.01}}$ |
| 20 | CT | $16.19_{\pm 0.01}$ | $15.84_{\pm 0.01}$ | $17.09_{\pm 0.01}$ | $16.55_{\pm 0.01}$ | $18.32_{\pm 0.01}$ |
| | Mamba-CDSP | $\mathbf{14.14^*_{\pm 0.01}}$ | $\mathbf{12.73^*_{\pm 0.01}}$ | $\mathbf{10.26^*_{\pm 0.01}}$ | $\mathbf{8.21^*_{\pm 0.01}}$ | $\mathbf{6.52^*_{\pm 0.01}}$ |

Table 8: Runtime of experiments (per training stage + per inference stage) on the real-world dataset. Experiments are carried out on $1\times$ NVIDIA GeForce RTX 3090 GPU, where ITPS refers to inference time per sample.

| | Stages of training & inference | Total runtime (in min) | training time | inference time | ITPS (in sec) |
|---|---|---|---|---|---|
| MSMs | 2 logistic regressions for IPTW & linear regression | $\mathbf{1.5_{\pm 0.4}}$ | $\mathbf{1.0_{\pm 0.1}}$ | $\mathbf{0.5_{\pm 0.2}}$ | $\mathbf{0.04}$ |
| RMSNs | 2 networks for IPTW & encoder & decoder | $38_{\pm 3.9}$ | $35_{\pm 0.1}$ | $3_{\pm 0.2}$ | $0.24$ |
| CRN | encoder & decoder | $109_{\pm 10.3}$ | $84_{\pm 0.1}$ | $25_{\pm 0.2}$ | $2$ |
| G-Net | single network & MC sampling for inference | $89_{\pm 5.2}$ | $86_{\pm 0.1}$ | $3_{\pm 0.2}$ | $0.24$ |
| CT | single multi-input network | $156$ | $98_{\pm 0.1}$ | $58_{\pm 0.1}$ | $4.64$ |
| Ours | Single Mamba Block | $12_{\pm 2.3}$ | $10_{\pm 0.1}$ | $2_{\pm 0.1}$ | $0.16$ |

Table 9: Total number of trainable parameters of models across synthetic data (Syn); semi-synthetic (SS) data, and (3) real-world (RW) MIMIC-III data.

| | Syn Data | | | | | SS data | RW data |
|---|---|---|---|---|---|---|---|
| | $\gamma = 0$ | $\gamma = 1$ | $\gamma = 2$ | $\gamma = 3$ | $\gamma = 4$ | | |
| MSMs | | | $<100$ | | | 3K | 1K |
| RMSNs | 20K | 4K | 23K | 21K | 22K | 477K | 947K |
| CRN | 4K | 6K | 8K | 7K | 8K | 165K | 219K |
| G-Net | 3K | 2K | 3K | 4K | 3K | 151K | 310K |
| CT | 11K | 11K | 10K | 10K | 10K | 45K | 69K |
| Mamba-CDSP | 12.8K | 12.7K | 12.8K | 12.8 | 12.9K | 26.8K | 27.3K |

Table 10: Results for experiments with the real-world MIMIC-III data, which are reported as the mean performance averaged over five runs with different seeds, and * means statistically significant results (p-value $\leq 0.01$) using the paired-t-test compared with the best baseline.

|  | Methods | $\gamma = 0$ | $\gamma = 1$ | $\gamma = 2$ | $\gamma = 3$ | $\gamma = 4$ |
|---|---|---|---|---|---|---|
| $\tau = 2$ | MSMs | $1.36_{\pm 0.11}$ | $1.61_{\pm 0.17}$ | $1.94_{\pm 0.37}$ | $2.29_{\pm 0.55}$ | $2.47_{\pm 1.06}$ |
|  | RMSNs | $0.74_{\pm 0.04}$ | $0.76_{\pm 0.05}$ | $0.83_{\pm 0.06}$ | $0.96_{\pm 0.11}$ | $1.28_{\pm 0.24}$ |
|  | CRN | $0.67_{\pm 0.07}$ | $\mathbf{0.67}_{\pm \mathbf{0.05}}$ | $0.74_{\pm 0.04}$ | $1.67_{\pm 1.18}$ | $1.15_{\pm 0.17}$ |
|  | G-Net | $1.02_{\pm 0.07}$ | $1.01_{\pm 0.09}$ | $1.27_{\pm 0.08}$ | $1.11_{\pm 0.15}$ | $1.26_{\pm 0.23}$ |
|  | CT | $0.68_{\pm 0.05}$ | $0.68_{\pm 0.04}$ | $\mathbf{0.71}_{\pm \mathbf{0.04}}$ | $0.91_{\pm 0.12}$ | $1.27_{\pm 0.37}$ |
|  | Mamba-HSIC | $0.73_{\pm 0.04}$ | $0.76_{\pm 0.04}$ | $0.79_{\pm 0.05}$ | $0.81_{\pm 0.59}$ | $0.84_{\pm 0.96}$ |
|  | Mamba-CDSP | $\mathbf{0.53}^{*}_{\pm \mathbf{0.04}}$ | $0.68_{\pm 0.03}$ | $0.72_{\pm 0.04}$ | $\mathbf{0.78}_{\pm \mathbf{0.50}}$ | $\mathbf{0.80}_{\pm \mathbf{0.70}}$ |
| $\tau = 3$ | MSMs | $1.68_{\pm 0.13}$ | $1.95_{\pm 0.21}$ | $2.30_{\pm 0.44}$ | $2.64_{\pm 0.64}$ | $2.62_{\pm 1.13}$ |
|  | RMSNs | $0.78_{\pm 0.05}$ | $0.79_{\pm 0.05}$ | $0.89_{\pm 0.05}$ | $1.09_{\pm 0.18}$ | $1.38_{\pm 0.29}$ |
|  | CRN | $0.70_{\pm 0.08}$ | $0.69_{\pm 0.05}$ | $0.82_{\pm 0.05}$ | $1.96_{\pm 1.03}$ | $1.36_{\pm 0.23}$ |
|  | G-Net | $1.25_{\pm 0.08}$ | $1.23_{\pm 0.10}$ | $1.61_{\pm 0.10}$ | $1.38_{\pm 0.20}$ | $1.57_{\pm 0.33}$ |
|  | CT | $0.70_{\pm 0.06}$ | $0.71_{\pm 0.04}$ | $0.77_{\pm 0.03}$ | $1.01_{\pm 0.12}$ | $1.54_{\pm 0.45}$ |
|  | Mamba-HSIC | $\mathbf{0.67}_{\pm \mathbf{0.05}}$ | $\mathbf{0.68}_{\pm \mathbf{0.04}}$ | $0.76_{\pm 0.05}$ | $0.91_{\pm 0.08}$ | $1.26_{\pm 0.25}$ |
|  | Mamba-CDSP | $0.68_{\pm 0.04}$ | $0.70_{\pm 0.05}$ | $\mathbf{0.73}^{*}_{\pm \mathbf{0.03}}$ | $\mathbf{0.78}^{*}_{\pm \mathbf{0.10}}$ | $\mathbf{0.89}^{*}_{\pm \mathbf{0.21}}$ |
| $\tau = 4$ | MSMs | $1.87_{\pm 0.15}$ | $2.15_{\pm 0.23}$ | $2.49_{\pm 0.47}$ | $2.79_{\pm 0.68}$ | $2.62_{\pm 1.14}$ |
|  | RMSNs | $0.82_{\pm 0.08}$ | $0.84_{\pm 0.06}$ | $0.96_{\pm 0.11}$ | $1.22_{\pm 0.24}$ | $1.42_{\pm 0.30}$ |
|  | CRN | $0.73_{\pm 0.09}$ | $0.72_{\pm 0.04}$ | $0.90_{\pm 0.07}$ | $2.20_{\pm 0.97}$ | $1.57_{\pm 0.26}$ |
|  | G-Net | $1.39_{\pm 0.09}$ | $1.35_{\pm 0.11}$ | $1.82_{\pm 0.13}$ | $1.54_{\pm 0.24}$ | $1.77_{\pm 0.41}$ |
|  | CT | $0.73_{\pm 0.06}$ | $0.75_{\pm 0.04}$ | $0.82_{\pm 0.04}$ | $1.09_{\pm 0.12}$ | $1.76_{\pm 0.52}$ |
|  | Mamba-HSIC | $\mathbf{0.69}_{\pm \mathbf{0.04}}$ | $0.76_{\pm 0.05}$ | $0.85_{\pm 0.04}$ | $1.22_{\pm 0.11}$ | $1.92_{\pm 0.35}$ |
|  | Mamba-CDSP | $0.70_{\pm 0.03}$ | $\mathbf{0.74}_{\pm \mathbf{0.04}}$ | $\mathbf{0.78}_{\pm \mathbf{0.05}}$ | $\mathbf{0.85}^{*}_{\pm \mathbf{0.09}}$ | $\mathbf{1.13}^{*}_{\pm \mathbf{0.28}}$ |
| $\tau = 5$ | MSMs | $1.96_{\pm 0.16}$ | $2.22_{\pm 0.23}$ | $2.55_{\pm 0.48}$ | $2.81_{\pm 0.68}$ | $2.54_{\pm 1.12}$ |
|  | RMSNs | $0.86_{\pm 0.10}$ | $0.89_{\pm 0.07}$ | $1.03_{\pm 0.17}$ | $1.35_{\pm 0.33}$ | $1.43_{\pm 0.30}$ |
|  | CRN | $0.77_{\pm 0.09}$ | $0.76_{\pm 0.04}$ | $0.98_{\pm 0.08}$ | $2.36_{\pm 1.00}$ | $1.73_{\pm 0.29}$ |
|  | G-Net | $1.48_{\pm 0.09}$ | $1.43_{\pm 0.12}$ | $1.96_{\pm 0.16}$ | $1.67_{\pm 0.28}$ | $1.91_{\pm 0.47}$ |
|  | CT | $0.76_{\pm 0.06}$ | $0.79_{\pm 0.04}$ | $0.87_{\pm 0.05}$ | $1.15_{\pm 0.11}$ | $1.92_{\pm 0.57}$ |
|  | Mamba-HSIC | $0.73_{\pm 0.04}$ | $0.76_{\pm 0.04}$ | $0.91_{\pm 0.05}$ | $1.22_{\pm 0.18}$ | $1.78_{\pm 0.31}$ |
|  | Mamba-CDSP | $\mathbf{0.72}_{\pm \mathbf{0.05}}$ | $\mathbf{0.75}_{\pm \mathbf{0.04}}$ | $\mathbf{0.84}_{\pm \mathbf{0.06}}$ | $\mathbf{0.91}^{*}_{\pm \mathbf{0.17}}$ | $\mathbf{1.13}^{*}_{\pm \mathbf{0.29}}$ |
| $\tau = 6$ | MSMs | $1.97_{\pm 0.16}$ | $2.21_{\pm 0.23}$ | $2.51_{\pm 0.47}$ | $2.73_{\pm 0.66}$ | $2.42_{\pm 1.08}$ |
|  | RMSNs | $0.89_{\pm 0.11}$ | $0.94_{\pm 0.09}$ | $1.08_{\pm 0.21}$ | $1.47_{\pm 0.43}$ | $1.44_{\pm 0.29}$ |
|  | CRN | $0.81_{\pm 0.10}$ | $\mathbf{0.79}_{\pm \mathbf{0.04}}$ | $1.05_{\pm 0.09}$ | $2.48_{\pm 1.08}$ | $1.83_{\pm 0.33}$ |
|  | G-Net | $1.54_{\pm 0.09}$ | $1.49_{\pm 0.12}$ | $2.06_{\pm 0.17}$ | $1.76_{\pm 0.29}$ | $1.99_{\pm 0.50}$ |
|  | CT | $0.79_{\pm 0.06}$ | $0.82_{\pm 0.03}$ | $0.91_{\pm 0.05}$ | $1.21_{\pm 0.10}$ | $2.05_{\pm 0.61}$ |
|  | Mamba-HSIC | $0.80_{\pm 0.05}$ | $0.85_{\pm 0.04}$ | $0.92_{\pm 0.03}$ | $1.02_{\pm 0.12}$ | $1.65_{\pm 0.31}$ |
|  | Mamba-CDSP | $\mathbf{0.77}_{\pm \mathbf{0.05}}$ | $0.81_{\pm 0.04}$ | $\mathbf{0.87}_{\pm \mathbf{0.05}}$ | $\mathbf{0.95}^{*}_{\pm \mathbf{0.18}}$ | $\mathbf{1.25}^{*}_{\pm \mathbf{0.26}}$ |

Table 11: Comparison with our Mamba-CDSP and Insite on synthetic TG simulator with the random-trajectory evaluation protocol, and * means statistically significant results (p-value $\leq 0.01$) using the paired-t-test compared with the best baseline.

|  | Methods | $\gamma = 0$ | $\gamma = 1$ | $\gamma = 2$ | $\gamma = 3$ | $\gamma = 4$ |
|---|---|---|---|---|---|---|
| $\tau = 2$ | Insite | $1.02_{\pm 0.03}$ | $1.01_{\pm 0.02}$ | $1.03_{\pm 0.07}$ | $1.15_{\pm 0.05}$ | $1.42_{\pm 0.02}$ |
|  | Mamba-CDSP | $\mathbf{0.60}^{*}_{\pm \mathbf{0.01}}$ | $\mathbf{0.62}^{*}_{\pm \mathbf{0.02}}$ | $\mathbf{0.63}^{*}_{\pm \mathbf{0.03}}$ | $\mathbf{0.65}^{*}_{\pm \mathbf{0.03}}$ | $\mathbf{0.72}^{*}_{\pm \mathbf{0.10}}$ |
| $\tau = 3$ | Insite | $1.00_{\pm 0.04}$ | $1.03_{\pm 0.04}$ | $1.03_{\pm 0.05}$ | $1.17_{\pm 0.59}$ | $1.44_{\pm 0.96}$ |
|  | Mamba-CDSP | $\mathbf{0.62}^{*}_{\pm \mathbf{0.01}}$ | $\mathbf{0.64}^{*}_{\pm \mathbf{0.03}}$ | $\mathbf{0.65}^{*}_{\pm \mathbf{0.03}}$ | $\mathbf{0.70}^{*}_{\pm \mathbf{0.04}}$ | $\mathbf{0.96}^{*}_{\pm \mathbf{0.09}}$ |
| $\tau = 4$ | Insite | $1.00_{\pm 0.04}$ | $1.04_{\pm 0.04}$ | $1.04_{\pm 0.05}$ | $1.18_{\pm 0.06}$ | $1.48_{\pm 0.04}$ |
|  | Mamba-CDSP | $\mathbf{0.64}^{*}_{\pm \mathbf{0.03}}$ | $\mathbf{0.68}^{*}_{\pm \mathbf{0.04}}$ | $\mathbf{0.73}^{*}_{\pm \mathbf{0.05}}$ | $\mathbf{0.79}^{*}_{\pm \mathbf{0.09}}$ | $\mathbf{1.12}^{*}_{\pm \mathbf{0.10}}$ |
| $\tau = 5$ | Insite | $0.99_{\pm 0.04}$ | $1.05_{\pm 0.04}$ | $1.05_{\pm 0.05}$ | $1.21_{\pm 0.59}$ | $1.51_{\pm 0.96}$ |
|  | Mamba-CDSP | $\mathbf{0.49}^{*}_{\pm \mathbf{0.03}}$ | $\mathbf{0.50}^{*}_{\pm \mathbf{0.04}}$ | $\mathbf{0.67}^{*}_{\pm \mathbf{0.07}}$ | $\mathbf{0.83}^{*}_{\pm \mathbf{0.10}}$ | $\mathbf{0.91}^{*}_{\pm \mathbf{0.15}}$ |
| $\tau = 6$ | Insite | $0.98_{\pm 0.04}$ | $1.05_{\pm 0.04}$ | $1.05_{\pm 0.05}$ | $1.21_{\pm 0.59}$ | $1.53_{\pm 0.96}$ |
|  | Mamba-CDSP | $\mathbf{0.49}^{*}_{\pm \mathbf{0.03}}$ | $\mathbf{0.50}^{*}_{\pm \mathbf{0.04}}$ | $\mathbf{0.67}^{*}_{\pm \mathbf{0.07}}$ | $\mathbf{0.83}^{*}_{\pm \mathbf{0.10}}$ | $\mathbf{0.91}^{*}_{\pm \mathbf{0.15}}$ |

Table 12: Sensitivity analysis on the semi-synthetic MIMIC-III dataset by tuning the de-correlation parameter $\alpha$, results are the average performance over five runs with different seeds.

| $\alpha$ | $\tau = 1$ | $\tau = 2$ | $\tau = 3$ | $\tau = 4$ | $\tau = 5$ | $\tau = 6$ | $\tau = 7$ | $\tau = 8$ | $\tau = 9$ | $\tau = 10$ |
|---|---|---|---|---|---|---|---|---|---|---|
| 0 | $0.32_{\pm 0.01}$ | $0.43_{\pm 0.01}$ | $0.54_{\pm 0.01}$ | $0.65_{\pm 0.01}$ | $0.75_{\pm 0.01}$ | $0.85_{\pm 0.01}$ | $0.93_{\pm 0.01}$ | $1.02_{\pm 0.01}$ | $1.09_{\pm 0.01}$ | $1.17_{\pm 0.01}$ |
| 0.0001 | $0.30_{\pm 0.01}$ | $0.37_{\pm 0.01}$ | $0.44_{\pm 0.01}$ | $0.49_{\pm 0.01}$ | $0.53_{\pm 0.01}$ | $0.57_{\pm 0.01}$ | $0.60_{\pm 0.01}$ | $0.62_{\pm 0.01}$ | $0.64_{\pm 0.01}$ | $0.65_{\pm 0.01}$ |
| 0.001 | $0.27_{\pm 0.01}$ | $0.33_{\pm 0.01}$ | $0.39_{\pm 0.01}$ | $0.43_{\pm 0.01}$ | $0.46_{\pm 0.01}$ | $0.49_{\pm 0.01}$ | $0.52_{\pm 0.01}$ | $0.54_{\pm 0.01}$ | $0.56_{\pm 0.01}$ | $0.58_{\pm 0.01}$ |
| 0.01 | $0.25_{\pm 0.01}$ | $0.32_{\pm 0.01}$ | $0.35_{\pm 0.01}$ | $0.44_{\pm 0.01}$ | $0.42_{\pm 0.01}$ | $0.46_{\pm 0.01}$ | $0.47_{\pm 0.01}$ | $0.48_{\pm 0.01}$ | $0.53_{\pm 0.01}$ | $0.53_{\pm 0.01}$ |
| 0.1 | $0.21_{\pm 0.01}$ | $0.28_{\pm 0.01}$ | $0.33_{\pm 0.01}$ | $0.41_{\pm 0.01}$ | $0.39_{\pm 0.01}$ | $0.45_{\pm 0.01}$ | $0.45_{\pm 0.01}$ | $0.45_{\pm 0.01}$ | $0.47_{\pm 0.01}$ | $0.49_{\pm 0.01}$ |
| 1.0 | $\mathbf{0.19_{\pm 0.01}}$ | $\mathbf{0.25_{\pm 0.01}}$ | $\mathbf{0.30_{\pm 0.01}}$ | $\mathbf{0.34_{\pm 0.01}}$ | $\mathbf{0.37_{\pm 0.01}}$ | $\mathbf{0.42_{\pm 0.01}}$ | $\mathbf{0.43_{\pm 0.01}}$ | $\mathbf{0.44_{\pm 0.01}}$ | $\mathbf{0.46_{\pm 0.01}}$ | $\mathbf{0.47_{\pm 0.01}}$ |
| 10.0 | $0.23_{\pm 0.01}$ | $0.27_{\pm 0.01}$ | $0.34_{\pm 0.01}$ | $0.36_{\pm 0.01}$ | $0.38_{\pm 0.01}$ | $0.44_{\pm 0.01}$ | $0.57_{\pm 0.01}$ | $0.59_{\pm 0.01}$ | $0.52_{\pm 0.01}$ | $0.50_{\pm 0.01}$ |

Table 13: Comparison with extreme long sequences with $a = 1000$ on TG simulator with $\gamma = 3$, and * means statistically significant results (p-value $\leq 0.01$) using the paired-t-test. Due to computational complexity of CT, we limit the range of the attention window CT in 200.

| | $\tau = 1$ | $\tau = 2$ | $\tau = 3$ | $\tau = 4$ | $\tau = 5$ | $\tau = 6$ |
|---|---|---|---|---|---|---|
| CT | $1.82_{\pm 0.25}$ | $2.51_{\pm 0.31}$ | $2.77_{\pm 0.33}$ | $2.89_{\pm 0.42}$ | $3.32_{\pm 0.47}$ | $3.85_{\pm 0.76}$ |
| Mamba-CDSP | $\mathbf{0.51^*_{\pm 0.02}}$ | $\mathbf{0.86^*_{\pm 0.06}}$ | $\mathbf{1.05^*_{\pm 0.09}}$ | $\mathbf{1.37^*_{\pm 0.13}}$ | $\mathbf{1.54^*_{\pm 0.18}}$ | $\mathbf{1.93^*_{\pm 0.21}}$ |

