# OpenReview forum: "Effective and Efficient Time-Varying Counterfactual Prediction with State-Space Models"
_ICLR.cc/2025/Conference — ICLR 2025 Poster_

### Official Review · Reviewer_PLDR · 2024-10-27

**Soundness:** 2
**Presentation:** 3
**Contribution:** 3
**Rating:** 6
**Confidence:** 4

**Summary:**

This paper proposes Mamba-CDSP, a novel approach for time-varying counterfactual prediction (TCP) based on state-space models (SSMs). The key contribution is adapting the Mamba architecture with a covariate-based decorrelation mechanism to handle sequential confounding bias while preserving covariate information. The authors demonstrate superior performance compared to existing methods on synthetic and real datasets.

**Strengths:**

1. Novel application of SSMs (specifically Mamba) to TCP, showing promising results in both effectiveness and efficiency. The paper leverages state-space models for counterfactual prediction, achieving significant improvements in both prediction accuracy and computational speed compared to existing methods.

2. Well-motivated decorrelation approach that addresses key limitations of existing balancing methods. The proposed CDSP mechanism offers a novel solution to the over-balancing problem in sequential settings, effectively balancing between confounding bias correction and preservation of important covariate information.

3. Comprehensive empirical evaluation across multiple datasets and settings.

**Weaknesses:**

1. Limited theoretical analysis of why covariance decorrelation works better than traditional balancing approaches.

2. While performance improvements are shown, deeper analysis of where/why the improvements come from would strengthen the paper. For instance, Table 2 shows substantial gains from CDSP on the MIMIC-III real-world dataset, but this is puzzling since we cannot observe counterfactuals in this data and thus confounding bias should have minimal impact on evaluation. The authors should explain why CDSP shows such dramatic improvements if the test metrics don't actually measure counterfactual prediction ability. This suggests the gains might come from other aspects of the method beyond bias correction, which deserves further investigation.

3. A more thorough literature review on temporal counterfactual estimation would enhance the paper by incorporating recent works like,
   - Chen et al, A Multi-Task Gaussian Process Model for Inferring Time-Varying Treatment Effects in Panel Data
   - Wu et al, Counterfactual Generative Models for Time-Varying Treatment
   - Wang et al, A Dual-module Framework for Counterfactual Estimation over Time
   - Berrevoets et al, Disentangled counterfactual recurrent networks for treatment effect inference over time

4. The paper lacks sufficient implementation details for reproducibility. While the model architecture is described, key details such as hyperparameters (hidden dimensions, number of layers), the decorrelation coefficient, and dropout rates are not specified. These details are crucial for reproducing the reported results.

The reference for domain adversarial learning on line 269 is incorrect, for example Lim 2018 did not use domain adversarial learning strategy

**Questions:**

1. Is CPSD on line 316 a typo?

2. Which dataset was used for Table 3?

3. How sensitive is the method to the choice of decorrelation hyperparameters?

---

> ### Author Response · Authors · 2024-11-26
>
> We sincerely appreciate the reviewer’s great efforts and insightful comments to improve our manuscript. We thank the reviewer for recognizing our novel application of SSMs, well-motivated decorrelation approach, as well as comprehensive empirical evaluation. In below, we address these concerns point by point and try our best to update the manuscript accordingly.
>
> > **W1: Limited theoretical analysis of why covariance decorrelation works better than traditional balancing approaches.**
>
> **Response:** We thank the reviewer for pointing out this issue, and we have added many theoretical analysis to show the superiority of the proposed CDSP method than traditional balancing approaches.
>
> ### **[Main Theoretical Result] CDSP has a tighter counterfactual prediction risk bound than traditional balancing approaches.**
> - To measure the counterfactual prediction accuracy, we **adopt expected precision in Estimation of Counterfactual Prediction (ECP) as evaluation metric** defined as
> $\epsilon_\mathrm{ECP}(W, \Phi)=\int_{\mathcal{X}}\left(\hat{\tau}_f(x)-\tau(x)\right)^2 p(x) dx$.
>
> - Consider the Gaussian covariates $X_{\mid a}   \sim N(\mu_{a}, \sigma_{a}I)$ and linear outcome structure $Y(a) = W^{a}\Phi X_{\mid a} + \epsilon^{a}$, where $\epsilon^{a} \sim N(0,1)$. Let $N_{a}$ be the sample number of each treatment arm, $W$ and $\Phi$ be the model parameters of representation layers and prediction head, respectively.
>
> - The **risk bounds** of **(1) vanilla Empirical Risk Minimization (ERM) method without any balancing modules**, (2) **adversarial balancing (ADB) methods,** and **(3) our proposed CDSP method** are derived as:
>   -  For **(1) vanilla ERM method without any balancing modules,** with probability at least $1-2\eta$, we have
> $$\epsilon_{ECP}^{vanilla}(W, \Phi) \leq 2(vr_1^2 |\mu_1 - \mu_0|^2_2 + v (\sqrt{\sigma_1} - \sqrt{\sigma_0})r_2^{2} + C - (\sqrt{\frac{1}{\eta}}(\sum_ {j=1}^{2} \sqrt{\kappa_j})  + \overline{\sigma})),$$
> where $\overline{\sigma} = \frac{1}{N_1} \sum_{i=1}^{N_1}{\epsilon^a_{i}}^2$.
>   - For **(2) adversarial balancing (ADB) methods, with probability at least $1-2\eta$,** we have
> $$\epsilon_{\mathrm{ECP}}^{adb}(\tilde{W}, \tilde{\Phi}) \leq 2(\frac{(2+ \sigma_0 + \sigma_1)(r_1r_3)^{2}}{4}  \|\mu_1-\mu_0\|^2_2 + C - (\sqrt{\frac{1}{\eta}}( \sum_{j=1}^{2}\sqrt{\kappa_j}) + \overline{\sigma} ) ).$$
>   - For **(3) our proposed CDSP method, with probability at least $1-2\eta$**, we have
> $$\epsilon_{\mathrm{ECP}}^{cdsp}(\tilde{W}, \tilde{\Phi}) \leq 2(\frac{(r_1r_3)^{2}}{2}  \|\mu_1-\mu_0\|^2_2 + C - (\sqrt{\frac{1}{\eta}}( \sum_{j=1}^{2}\sqrt{\kappa_j})+ \overline{\sigma} ) ).$$
> - With the above risk bounds, we then derive two main results:
>   - **(a) Over-balancing of ADB methods:** When the following condition holds, the risk bound of vanilla ERM model is more tighter than that of adversarial balancing methods:
> $$\|\mu_1 - \mu_0\|_{2}^{2} \leq \frac{v(\sqrt{\sigma_1} - \sqrt{\sigma_0})^2 r_2^2}{vr_1^2 - (2+\sigma_1 + \sigma_0)r_1^2r_3^2/4}. $$
> This illustrates that **as the gap between two groups decreases, the harm of over-balancing is much larger than that of observed confounding bias.**
>   - **(b) CDSP outperforms ADB with a tighter bound:** Our CDSP method has a tighter bound compared with the existing ADB methods, i.e., $\epsilon_{\mathrm{ECP}}^{cdsp}(\tilde{W}, \tilde{\Phi}) \leq \epsilon_{\mathrm{ECP}}^{adb}(\tilde{W}, \tilde{\Phi})$.
> - We have **added the above theoretical analysis in a new Section 4.4** with detailed proofs in Appendix A.3.
>
> ### **[Minor Theoretical Result] The computational complexity of our CDSP is smaller than traditional balancing approaches if the representation dimension of $A$ is small.**
>
> - We also **added computational complexity analysis** and **compare our proposed CDSP with previous adversarial balancing methods.**
> - Denote the overall length of the time horizon as $T$, and the representation dimension of $A$ and $X$ as $d^h_A$ and $d^h_X$, respectively. For adversarial balancing modules, previous practice shows that the discriminator has usually $2$ layers with $d^{h}_{X}$ for each layer.
>
> - Then the **overall complexity of our proposed CDSP method is $\mathcal{O}(B((d_{X}^{h})^3 + d_{A}^{h})T)$**, whereas **the overall complexity of ADB can be derived as $\mathcal{O}(B((d_{X}^{h})^3 + (d_{X}^{h})^2 d_{Y}^{h} + |A|)T)$** with $B$ as a constant related to parameter $\overline{B}$ (see detailed derivation in Appendix A.4.3).
>
> - From above, **the overall complexity of our proposed CDSP method will be smaller than the ADB methods if the representation dimension of $A$ denoted as $d_{A}^{h}$ is small,** i.e., $d_{A}^{h} \leq (d_{X}^{h})^2 d_{Y}^{h} + |A|$.

---

> ### Author Response · Authors · 2024-11-27
>
> **> W2: While performance improvements are shown, deeper analysis of where/why the improvements come from would strengthen the paper. For instance, Table 2 shows substantial gains from CDSP on the MIMIC-III real-world dataset, but this is puzzling since we cannot observe counterfactuals in this data and thus confounding bias should have minimal impact on evaluation. The authors should explain why CDSP shows such dramatic improvements if the test metrics don't actually measure counterfactual prediction ability. This suggests the gains might come from other aspects of the method beyond bias correction, which deserves further investigation.**
>
> **Response:** Thank you for the detailed and constructive comments. We would like to address your concerns by (1) recap the  synthetic and semi-synthetic experiments with simulated confounding biases, (2) add synthetic experiments with increasing confounding bias, (3) add real-world experiments on the M5 dataset, and (4) add a brief discussion of why our methods achieves significant improvement on real-world datasets.
>
> ### **Recap the synthetic and semi-synthetic experiments we have done with simulated confounding biases**
>
> - In our original manuscript, we have evaluated our methods on the synthetic dataset (Figure 4), semi-synthetic dataset (Table 1), and real-world dataset (Table 2).
>   -  Synthetic dataset and semi-synthetic MIMIC-III dataset: We follow M van der Schaar, et al. (ICLR 2020) to conduct our synthetic experiments, and follow Melnychuk, et al. (ICML 2022) to conduct our synthetic experiments on the MIMIC-III dataset. Both experiments are conducted with the **same data generation process (DGP)**, in which **the confounding bias indeed exists due to the simulated counterfactual outcomes;**
>   - Real-world MIMIC-III dataset: We follow van der Schaar, et al. (ICLR 2020) and Melnychuk, et al. (ICML 2022) to conduct our real-world experiments on the MIMIC-III dataset, in which the counterfactual outcomes are absent.
>
> ### **Add synthetic experiments with increasing confounding bias**
>
> - To further validate that our method can correct for confounding bias, we **added synthetic experiments with increasing confounding bias on the tumor growth (TG) simulator.**
> - We follow Melnychuk, et al. (ICML 2022) to **control the level of confounding via tuning $\gamma$** in Eq. (33) $\mathbf{A}^c_t, \mathbf{A}^r_t \sim \operatorname{Bernoulli}( \sigma ( \frac{\gamma}{D_{max}})(D_{15}(Y_{t-1}) - D_{max} / 2))$. For $\gamma=0$, the treatment assignment is fully randomized. **For increasing values, the amount of time-varying confounding becomes also larger.**
>
> | $\gamma$ |    Model   |     $\tau = 1$     |     $\tau = 2$     |     $\tau = 3$     |     $\tau = 4$     |     $\tau = 5$     |
> |----------|:----------:|:------------------:|:------------------:|:------------------:|:------------------:|:------------------:|
> | 8        |     Causal Transformer     |  $2.98_{\pm 0.01}$ |  $3.46_{\pm 0.01}$ |  $3.56_{\pm 0.01}$ |  $3.52_{\pm 0.01}$ |  $3.33_{\pm 0.01}$ |
> | 8        | Mamba-CDSP |  $2.97_{\pm 0.01}$ |  $\mathbf{3.36^*_{\pm 0.01}}$ |  $\mathbf{3.29^*_{\pm 0.01}}$ |  $\mathbf{3.02^*_{\pm 0.01}}$ |  $\mathbf{2.69^*_{\pm 0.01}}$ |
> | 10       |     Causal Transformer     | $10.46_{\pm 0.01}$ |  $9.38_{\pm 0.01}$ |  $8.50_{\pm 0.01}$ |  $7.63_{\pm 0.01}$ |  $6.83_{\pm 0.01}$ |
> | 10       | Mamba-CDSP |  $\mathbf{4.25^*_{\pm 0.01}}$ |  $\mathbf{5.26^*_{\pm 0.01}}$ |  $\mathbf{5.58^*_{\pm 0.01}}$ |  $\mathbf{5.55^*_{\pm 0.01}}$ |  $\mathbf{5.40^*_{\pm 0.01}}$ |
> | 16       |     Causal Transformer     | $15.93_{\pm 0.01}$ | $13.54_{\pm 0.01}$ | $15.54_{\pm 0.01}$ | $16.40_{\pm 0.01}$ | $16.58_{\pm 0.01}$ |
> | 16       | Mamba-CDSP |  $\mathbf{8.99^*_{\pm 0.01}}$ |  $\mathbf{9.06^*_{\pm 0.01}}$ |  $\mathbf{8.09^*_{\pm 0.01}}$ |  $\mathbf{6.93^*_{\pm 0.01}}$ |  $\mathbf{5.95^*_{\pm 0.01}}$ |
> | 20       |     Causal Transformer     | $16.19_{\pm 0.01}$ | $15.84_{\pm 0.01}$ | $17.09_{\pm 0.01}$ | $16.55_{\pm 0.01}$ | $18.32_{\pm 0.01}$ |
> | 20       | Mamba-CDSP | $\mathbf{14.14^*_{\pm 0.01}}$ | $\mathbf{12.73^*_{\pm 0.01}}$ | $\mathbf{10.26^*_{\pm 0.01}}$ |  $\mathbf{8.21^*_{\pm 0.01}}$ |  $\mathbf{6.52^*_{\pm 0.01}}$ |
>
> - From above results, we conclude that **our proposed method significantly outperform with $\text{p-value} \leq 0.01$ using the paired-t-test compared with Causal Transformer for correcting the confounding bias.** We have supplemented such results in our revised paper (Table 9, P24).
>
> ### **Add real-world experiments on the M5 dataset**
>
> - We also follow Huang et al. (ICML 2024) to **add experiments on a new real-world time-series prediction dataset,** i.e., the **M5 dataset,** which provided by Walmart, captures the unit sales data of a diverse range of products sold across the United States, structured as grouped time series.

---

> ### Author Response · Authors · 2024-11-27
>
> |                Models               |           $\tau = 1$          |        $\tau = 2$       |        $\tau = 3$       |        $\tau = 4$       |        $\tau = 5$       |        $\tau = 6$       |
> |:------------------------------:|:-----------------------------:|:-----------------------:|:-----------------------:|:-----------------------:|:-----------------------:|:-----------------------:|
> |              RMSN              |             $5.19_{\pm 0.12}$ |       $9.73_{\pm 0.26}$ |      $10.48_{\pm 0.34}$ |      $11.07_{\pm 0.43}$ |      $11.63_{\pm 0.53}$ |      $12.16_{\pm 0.65}$ |
> |               CRN              |             $4.98_{\pm 0.32}$ |       $9.15_{\pm 0.17}$ |       $9.79_{\pm 0.16}$ |      $10.12_{\pm 0.17}$ |      $10.38_{\pm 0.19}$ |      $10.59_{\pm 0.22}$ |
> |            CRN w/o Balancing           |             $4.68_{\pm 0.06}$ |       $9.05_{\pm 0.16}$ |       $9.68_{\pm 0.15}$ |      $10.00_{\pm 0.16}$ |      $10.26_{\pm 0.18}$ |      $10.47_{\pm 0.20}$ |
> |               Causal Transformer               |             $4.59_{\pm 0.08}$ |       $8.99_{\pm 0.19}$ |       $9.59_{\pm 0.19}$ |       $9.91_{\pm 0.23}$ |      $10.15_{\pm 0.25}$ |      $10.35_{\pm 0.28}$ |
> |            Causal Transformer w/o Balancing            |             $4.59_{\pm 0.09}$ |       $8.98_{\pm 0.18}$ |       $9.58_{\pm 0.18}$ |       $9.90_{\pm 0.20}$ |      $10.13_{\pm 0.23}$ |      $10.34_{\pm 0.26}$ |
> |           Mamba-CDSP           | $\mathbf{ 4.08^*_{\pm 0.06}}$ | $\mathbf{ 4.05^*_{\pm 0.13}}$ | $\mathbf{ 4.59^*_{\pm 0.15}}$ | $\mathbf{ 5.87^*_{\pm 0.17}}$ | $\mathbf{6.47^*_{\pm 0.20}}$ | $\mathbf{8.39^*_{\pm 0.28}}$ |
>
> - From the above results, we further validate the effectiveness of our Mamba-CDSP on the real-world dateset.
>
> ### **Add a brief discussion of why our methods achieves significant improvement on real-world datasets**
>
> - As refered by As refered by Huang et al. (ICML 2024), **prediction on the realistic data equals to factual outcome prediction due to the absence of realistic counterfactual outcomes.**
> - Hence, improvement on the MIMIC-III and M5 datasets verifies that **our method achieves successful preservation on the covariate information.**
> - We have **supplemented a more detailed discussion in our revised paper (P9-10, Lines 482-497).**
>
> > **W3: A more thorough literature review on temporal counterfactual estimation would enhance the paper by incorporating recent works**
>
> **Response:** Thank you for your kind advice! **We have re-write the related-work section (P3, Sec 2) with a more thorough and complete review by adding over 15 new recent works including but not limit to the 4 works reviewer mentioned.**
>
> - We have supplemented two extra aspects: **Development on TCP with Statistical Approaches** and **Distributional Counterfactual Estimation over Time** and grouped the 4 works reviewer mentioned as follows.
> - **Development on TCP with Statistical Approaches contains the statistical temporal causal methods:**
>   - Chen et al. proposes to disentangle the trends of unit and group when predicting the ATT.
>   - Wang et al. proposes to integrate historical information together based on adversarial balancing modules.
>   - Berrevoets et al. considers covariate disentangle for more precise separation between confounders, instruments and precision variables.
> - **Distributional Counterfactual Estimation aims to model the distribution of outcome rather than the mean:**
>   -  Wu et al. proposes a new counterfactual generation method with the aid of GANs.
>
> > **W4: The paper lacks sufficient implementation details for reproducibility. While the model architecture is described, key details such as hyperparameters (hidden dimensions, number of layers), the decorrelation coefficient, and dropout rates are not specified. These details are crucial for reproducing the reported results.**
>
> **Response:** To enhance reproducibility of our paper, we have **supplemented the following details parameter configuration in our revised PDF (P22, Table 6).**
>
> | Model | Hyperparameter | TG simulator | Semi-Synthetic Data | Real-world Data |
> |-|-|:-:|:-:|:-:|
> | Mamba-CDSP | Mamba blocks ($B$) | 1 | 1 | 2 |
> | | Learning rate ($\eta$) | \{0.0005, 0.001, 0.01\} | \{0.0005, 0.001, 0.01\} | \{0.0005, 0.001, 0.01\} |
> | | Minibatch size | 128 | 64 | 64 |
> | | De-correlation Parameter | 1 | 1 | 1 |
> | | EL hidden units ($d_EL$) | 32 | 32 | 64 |
> | | PL hidden units ($d_{\text{PL}}$) | 32 | 32 | 64 |
> | | Dropout rate ($p$)  | 0.1 | 0.1  | 0.1 |
> | | EMA of model weights | 0.99 | 0.99 | 0.99 |
> | | Input size  | $d_a + d_x + d_y + d_v$ | $d_a + d_x + d_y + d_v$ | $d_a + d_x + d_y + d_v$ |
> | | Output size | $d_y$ | $d_y$ | $d_y$ |

---

> > ### Author Response · Authors · 2024-11-27
> >
> > > **The reference for domain adversarial learning on line 269 is incorrect, for example Lim 2018 did not use domain adversarial learning strategy**
> >
> > **Response:** Thank you for pointing out this issue. We have corrected this incorrect reference in Section 4.3 in our revised paper.
> >
> > > **Q1: Is CPSD on line 316 a typo?**
> >
> > **Response:** Yes, CPSD on line 316 is a typo. We have carefully corrected the typos in the manuscript and summarized them below.
> >
> > |  Before |  After |  Location |
> > |---|---|---|
> > |  Missing : | Add :  | Line 266  |
> > |  CPSD  | CDSP  | Line 316  |
> > | Selective parameter ($A$) | $C$  | Lines 206, 208, 212, 306, 318  |
> > | Projection parameter ($C$) | $R$  | Lines 204, 206, 209  |
> >
> > > **Q2: Which dataset was used for Table 3?**
> >
> > **Response:** Table 3 corresponds to the semi-synthetic MIMIC-III dataset. We have re-clarified this point in Line 470, P9 in our revised paper.
> >
> > > **Q3: How sensitive is the method to the choice of decorrelation hyperparameters?**
> >
> > **Response:** As suggested by the reviewer, we **add sensitive analysis to study the choice of decorrelation hyperparameter $\alpha$ with varying time-steps $\tau \in \\{1, \ldots, 10\\}$ on the semi-synthetic MIMIC-III dataset.**
> >
> > - The higher decorrelation hyperparameter $\alpha$, the stronger de-correlation is performed to regularize the prediction process.
> >
> > | $\alpha$ |     $\tau = 1$    |     $\tau = 2$    |     $\tau = 3$    |     $\tau = 4$    |     $\tau = 5$    |     $\tau = 6$    |     $\tau = 7$    |     $\tau = 8$    |     $\tau = 9$    |    $\tau = 10$    |
> > |----------|:-----------------:|:-----------------:|:-----------------:|:-----------------:|:-----------------:|:-----------------:|:-----------------:|:-----------------:|:-----------------:|:-----------------:|
> > | 0        | $0.32_{\pm 0.01}$ | $0.43_{\pm 0.01}$ | $0.54_{\pm 0.01}$ | $0.65_{\pm 0.01}$ | $0.75_{\pm 0.01}$ | $0.85_{\pm 0.01}$ | $0.93_{\pm 0.01}$ | $1.02_{\pm 0.01}$ | $1.09_{\pm 0.01}$ | $1.17_{\pm 0.01}$ |
> > | 0.0001   | $0.30_{\pm 0.01}$ | $0.37_{\pm 0.01}$ | $0.44_{\pm 0.01}$ | $0.49_{\pm 0.01}$ | $0.53_{\pm 0.01}$ | $0.57_{\pm 0.01}$ | $0.60_{\pm 0.01}$ | $0.62_{\pm 0.01}$ | $0.64_{\pm 0.01}$ | $0.65_{\pm 0.01}$ |
> > | 0.001    | $0.27_{\pm 0.01}$ | $0.33_{\pm 0.01}$ | $0.39_{\pm 0.01}$ | $0.43_{\pm 0.01}$ | $0.46_{\pm 0.01}$ | $0.49_{\pm 0.01}$ | $0.52_{\pm 0.01}$ | $0.54_{\pm 0.01}$ | $0.56_{\pm 0.01}$ | $0.58_{\pm 0.01}$ |
> > | 0.01     | $0.25_{\pm 0.01}$ | $0.32_{\pm 0.01}$ | $0.35_{\pm 0.01}$ | $0.44_{\pm 0.01}$ | $0.42_{\pm 0.01}$ | $0.46_{\pm 0.01}$ | $0.47_{\pm 0.01}$ | $0.48_{\pm 0.01}$ | $0.53_{\pm 0.01}$ | $0.53_{\pm 0.01}$ |
> > | 0.1      | $0.21_{\pm 0.01}$ | $0.28_{\pm 0.01}$ | $0.33_{\pm 0.01}$ | $0.41_{\pm 0.01}$ | $0.39_{\pm 0.01}$ | $0.45_{\pm 0.01}$ | $0.45_{\pm 0.01}$ | $0.45_{\pm 0.01}$ | $0.47_{\pm 0.01}$ | $0.49_{\pm 0.01}$ |
> > | 1.0      |  $\mathbf{0.19_{\pm 0.01}}$ |  $\mathbf{0.25_{\pm 0.01}}$  |  $\mathbf{0.30_{\pm 0.01}}$  |  $\mathbf{0.34_{\pm 0.01}}$  |  $\mathbf{0.37_{\pm 0.01}}$  |  $\mathbf{0.42_{\pm 0.01}}$  |  $\mathbf{0.43_{\pm 0.01}}$  |  $\mathbf{0.44_{\pm 0.01}}$  |  $\mathbf{0.46_{\pm 0.01}}$  |  $\mathbf{0.47_{\pm 0.01}}$  |
> > | 10.0     | $0.23_{\pm 0.01}$ | $0.27_{\pm 0.01}$ | $0.34_{\pm 0.01}$ | $0.36_{\pm 0.01}$ | $0.38_{\pm 0.01}$ | $0.44_{\pm 0.01}$ | $0.57_{\pm 0.01}$ | $0.59_{\pm 0.01}$ | $0.52_{\pm 0.01}$ | $0.50_{\pm 0.01}$ |
> >
> > - From above results, we found that **our method is not sensitive to the choice of decorrelation hyperparameters when $\alpha$ is within a proper range [0.1, 10], and our method stably perform the best when choosing $\alpha=1.0$.** We have supplemented such results in our revised paper (Table 3, P9).
> >
> > ***
> > **We hope the above discussion will fully address your concerns about our work, and we would really appreciate it if you would like to kindly consider adjusting your score.** We look forward to your insightful and constructive responses to further help us improve the quality of our work. Thank you!
> > ***
> >
> > **References**
> >
> > [1] Bica et al, Estimating counterfactual treatment outcomes over time through adversarially balanced representations, ICLR 2020.
> >
> > [2] Melnychuk et al, Causal transformer for estimating counterfactual outcomes, ICML 2022.
> >
> > [3] Huang et al, An Empirical Examination of Balancing Strategy for Counterfactual Estimation on Time Series, ICML 2024.
> >
> > [4] Chen et al, A Multi-Task Gaussian Process Model for Inferring Time-Varying Treatment Effects in Panel Data, AISTATS 2023.
> >
> > [5] Wang et al, A Dual-module Framework for Counterfactual Estimation over Time, ICML 2024.
> >
> > [6] Berrevoets et al, Disentangled counterfactual recurrent networks for treatment effect inference over time, arXiv:2112.03811 2021.
> >
> > [7] Wu et al, Counterfactual Generative Models for Time-Varying Treatment, KDD 2024.

---

> > > ### Comment · Reviewer_PLDR · 2024-11-27
> > >
> > > Thank you very much for your response. Regarding the theoretical part, is the bound in case 2 tighter compared to case 1? If so, how can this theory explain why balancing strategies sometimes lead to performance degradation? I believe this line of thinking is worth exploring, but the author needs to design more appropriate experiments combined with the theory for verification. Also, the current version seems to be missing A.3. Regarding experiments on real datasets, is the loss of covariate information caused by balancing? However, looking at the results, for example in CT, there doesn't seem to be much difference.

---

> ### Author Response · Authors · 2024-11-27
> **Response to Reviewer PLDR**
>
> Thanks for your detailed and through response in time! We have uploaded the whole revised PDF, the revised component are marked in **red**. In below, we clarify your further questions point by point.
>
> > [Q1 Theoretical Insights]
> - The bound in case 2 is tighter compared to case 1 when **distributional shifts of covariates between different treatment groups decreases**:
> $$\|\mu_1 - \mu_0\|_{2}^{2} \leq \frac{v(\sqrt{\sigma_1} - \sqrt{\sigma_0})^2 r_2^2}{vr_1^2 - (2+\sigma_1 + \sigma_0)r_1^2r_3^2/4}. $$
> - The above inequality informs that the **when first-moment of covariates ($\mu_1, \mu_0$) across different treatment groups** are close to each other, the profit brought by balancing will be smaller than the harm caused by balancing.
> - **how can our theory explain why balancing strategies sometimes lead to performance degradation**:
>    1. The risk bound consists of **Wasserstein term** and supervised **loss term**, representing for **observed confounding bias** and **prediction loss** respectively;
>    2. The profit of balancing corresponds to the Wasserstein term;
>    3. In our theory, the process of balancing is formalized as the **variation** of both $E[X]$ and $Var(X)$ across $T=1$ and $T=0$ at the same time, i.e., distributional alignment at the representations;
>    4. At the same time, such **variation** causes the prediction error to be larger;
>    4. When the Wasserstein term is small, i.e., $\|\mu_1 - \mu_0\|_{2}^{2}$ is small, the profit of balancing will be small than the enlarged error of prediction.
>
>
> > [Q2 Adequate Empirical Verification]
> -  Regarding experiments on real datasets, the loss of covariate information is caused by over-balancing. A recent benchmarking study has conducted extensive empirical studies to inform the phenomena of over-balancing [1].
> - **In our paper**, wo would like to clarify that we have **already** conducted extensive experimental validations to verify such phenomena:
>    1. **Reconstruction Experiments**:
>         - We analyze if models, specifically the our proposed Mamba-CDSP, can accurately reconstruct original covariates with and without the balancing module.
>         - Results are shown in Fig.5 in our revised paper. (**CT w/o balancing** and **CT** are already shown in Fig.6 in [1]);
>         - For **CT**, without balancing modules, the re-constructed covariates curve converges; while the curve is difficult to converge with adversarial balancing (**An existing conclusion in [1]**);
>         - For **Mamba-CDSP**, our CDSP ensures a more converged training curve during re-construction.
>      1. **Visualization Experiments**:
>         - We perform representation visualization to further inform how distributions of $X$ are changes with and without balancing modules at various time steps in Fig. 6 and Fig.7;
>         - Compared to CT with adversarial balancing, our CDSP achieves better preservation of covariate information;
>
> > [Q3  There doesn't seem to be much difference.]
>
> - In the M5 real-world table, we rounded each value to two decimal places. We now list the original results to show the risk that **balancing may not lead to more superior prediction performance:**
>
> |                Models               |           $\tau = 1$          |        $\tau = 2$       |        $\tau = 3$       |        $\tau = 4$       |        $\tau = 5$       |        $\tau = 6$       |
> |:------------------------------:|:-----------------------------:|:-----------------------:|:-----------------------:|:-----------------------:|:-----------------------:|:-----------------------:|
> | CT | $4.5934 \pm 0.0823$ | $8.9863 \pm 0.1888$ | $9.5868 \pm 0.1934$ | $9.9135 \pm 0.2266$ | $10.1454 \pm 0.2535$ | $10.3476 \pm 0.2838$ |
> | CT w/o balancing | $4.5908 \pm 0.0855$ | $8.9802 \pm 0.1814$ | $9.5806 \pm 0.1789$ | $9.9029 \pm 0.2047$ | $10.1342 \pm 0.2273$ | $10.3354 \pm 0.2551$ |
>
> - Moreover, we have **supplemented** experiments on the real-world MIMIC-III dataset by comparing CT with and without balancing modules as follows:
>
> |                Models               |           $\tau = 1$          |        $\tau = 2$       |        $\tau = 3$       |        $\tau = 4$       |        $\tau = 5$
> |:------------------------------:|:-----------------------------:|:-----------------------:|:-----------------------:|:-----------------------:|:-----------------------:|
> | CT | $4.6091 \pm 0.0621$ |  $9.0188 \pm 0.1514$ | $9.5843 \pm 0.2109$ | $9.8965 \pm 0.2566$ | $10.1207 \pm 0.2934$ |
> | CT w/o balancing| $4.2305 \pm 0.0641$ | $8.4987 \pm 0.1373$ | $8.6014 \pm 0.1793$ | $9.3092 \pm 0.2104$ | $9.7446 \pm 0.2355$ |
>
> - Compared to the M5 real-world dataset, **the balancing on CT on the MIMIC-III dataset has a more significant performance damage compared to pure CT without balancing, due to overbalancing would cause loss of covariate information.**
> ---
> **Reference**
> [1] Huang et al, An Empirical Examination of Balancing Strategy for Counterfactual Estimation on Time Series, ICML 2024.

---

> ### Author Response · Authors · 2024-11-27
> **More Discussion on the Trade-Off Between Wasserstein Balancing Error Term and Supervised Loss Term**
>
> -  For **(1) vanilla ERM method without any balancing modules,** with probability at least $1-2\eta$, we have
> $$\epsilon_{ECP}^{vanilla}(W, \Phi) \leq 2(vr_1^2 |\mu_1 - \mu_0|^2_2 + v (\sqrt{\sigma_1} - \sqrt{\sigma_0})r_2^{2}+ C - (\sqrt{\frac{1}{\eta}}(\sum_ {j=1}^{2} \sqrt{\kappa_j})  + \overline{\sigma})),$$
> in which the **Wasserstein balancing error term** explains the risk due to **insufficient balancing in ERM**:
> $$2(vr_1^2 |\mu_1 - \mu_0|^2_2 + v (\sqrt{\sigma_1} - \sqrt{\sigma_0})r_2^{2}).$$
>   - For **(2) adversarial balancing (ADB) methods, with probability at least $1-2\eta$,** we have
> $$\epsilon_{\mathrm{ECP}}^{adb}(\tilde{W}, \tilde{\Phi}) \leq 2(\frac{(2+ \sigma_0 + \sigma_1)(r_1r_3)^{2}}{4}  \|\mu_1-\mu_0\|^2_2 + C - (\sqrt{\frac{1}{\eta}}( \sum_{j=1}^{2}\sqrt{\kappa_j}) + \overline{\sigma} ) ),$$
> in which the **supervised loss term** explains the risk due to the **loss of covariate information by overbalancing**:
> $$2\frac{(2+ \sigma_0 + \sigma_1)(r_1r_3)^{2}}{4}  \|\mu_1-\mu_0\|^2_2.$$
> - This shows a **trade-off between fitting the factual outcomes (ERM method) and covariate balancing (ADB method).**
>   - From our theory, the **advantage of balancing methods** is formulated as the **reduction of both the mean difference $|\mu_1 - \mu_0|^2_2$ and the variance difference $\sqrt{\sigma_1} - \sqrt{\sigma_0}$**, i.e., **distributional alignment** at the covariate representations;
>   - **When the Wasserstein balancing error term is small enough, i.e., $\|\mu_1 - \mu_0\|_{2}^{2}$ is small enough, the profit of balancing will be smaller than the extra supervised prediction error due to the loss of covariate information,** causing the performance drop of balancing methods compared to vanilla ERM methods.
> - As per your suggestion, **we are conducting experiments to validate this trade-off between the two terms, and we will follow up this message once we get the experimental results** : )

---

> > ### Comment · Reviewer_PLDR · 2024-11-27
> >
> > I am referring to experiments that can verify the current theoretical findings. For example: "Over-balancing of ADB methods: When the following condition holds, the risk bound of vanilla ERM model is more tighter than that of adversarial balancing methods: $$|\mu_1 - \mu_0|_{2}^{2} \leq \frac{v(\sqrt{\sigma_1} - \sqrt{\sigma_0})^2 r_2^2}{vr_1^2 - (2+\sigma_1 + \sigma_0)r_1^2r_3^2/4}. $$ This illustrates that as the gap between two groups decreases, the harm of over-balancing is much larger than that of observed confounding bias." I think this could be verified through designing corresponding simulation datasets. Regarding experiments on real datasets, thank you for the authors' supplement. I am curious whether removing CDSP would mean better preservation of covariate information? Besides, for CT experiments on MIMIC-III, does w/o balancing simply mean keeping only the experimental results with α=0?

---

> ### Author Response · Authors · 2024-11-29
>
> To verify the current theoretical findings regarding the over-balancing issue as well as the tighter risk bound of our CDSP method, we **add synthetic experiments with varying confounding strength $\gamma$.**
> - We adopt the following **three metrics**:
>   - **Wasserstein balancing error** (named **Wasserstein** in the Table) in our derived bound arises from **insufficient balancing** $2(vr_1^2 |\mu_1 - \mu_0|^2_2 + v (\sqrt{\sigma_1} - \sqrt{\sigma_0})r_2^{2})$;
>   - **Supervised error** (named **Supervised** in the Table) in our derived bound arises from the loss of covariate information by **over-balancing** $2\frac{(2+ \sigma_0 + \sigma_1)(r_1r_3)^{2}}{4}  \|\mu_1-\mu_0\|^2_2$;
>   - **Overall TCP performance** (named **Overall** in the Table) containing **both factual and counterfactual prediction error**.
> - We compare the following **four methods**:
>   - **CT**: adversarial balanced Causal Transformer, in which $\alpha$ are tuned with best hyper-parameter search as in the original CT paper (Melnychuk et al, ICML 2022);
>   - **CT w/o B**: CT without adversarial balancing by setting $\alpha_{CT}=0$;
>   - **Mamba-CDSP** (ours): Our proposed methods with de-correlation;
>   - **Mamba-CDSP w/o B**: Vanilla Mamba model without de-correltaion by setting $\alpha_{Mamba}=0$.
>
> | $\gamma$ | CT| CT | CT| CT w/o B | CT w/o B | CT w/o B | Mamba-CDSP | Mamba-CDSP | Mamba-CDSP | Mamba-CDSP w/o B | Mamba-CDSP w/o B | Mamba-CDSP w/o B |
> |--|--|--|--|--|--|--|--|--|--|--|--|--|
> | Metric   | Overall | Supervised | Wasserstein | Overall  | Supervised | Wasserstein  | Overall    | Supervised | Wasserstein | Overall     | Supervised | Wasserstein |
> |     3    |  $\mathbf{1.66_{\pm 0.07}}$   | $1.53_{\pm 0.06}$ | $1.46_{\pm 0.12}$ |   $1.98_{\pm 0.18}$   |   $1.41_{\pm 0.10}$   |   $1.63_{\pm 0.12}$   |    $\mathbf{0.97_{\pm 0.04}}$    |    $0.91_{\pm 0.02}$    |    $1.06_{\pm 0.05}$    |    $1.15_{\pm 0.06}$    |     $0.85_{\pm 0.03}$    |    $1.14_{\pm 0.09}$    |
> |     2    |  $1.56_{\pm 0.03}$    | $1.24_{\pm 0.05}$ | $1.23_{\pm 0.10}$ | $\mathbf{1.27_{\pm 0.11}}$ | $0.93_{\pm 0.07}$ | $1.32_{\pm 0.09}$ | $\mathbf{0.73_{\pm 0.04}}$ | $0.72_{\pm 0.03}$ | $0.88_{\pm 0.05}$ | $0.80_{\pm 0.04}$ | $0.67_{\pm 0.04}$ | $0.94_{\pm 0.04}$ |
> |     1    |   $0.96_{\pm 0.09}$  | $0.76_{\pm 0.06}$ | $0.95_{\pm 0.06}$ |   $\mathbf{0.88_{\pm 0.06}}$   |   $0.62_{\pm 0.03}$   |   $1.06_{\pm 0.06}$   |    $0.65_{\pm 0.03}$    |    $0.58_{\pm 0.03}$    |    $0.78_{\pm 0.04}$    |     $\mathbf{0.64_{\pm 0.03}}$    |     $0.51_{\pm 0.02}$    |    $0.81_{\pm 0.05}$    |
> |     0    |   $0.82_{\pm 0.04}$ | $0.68_{\pm 0.03}$ | $0.81_{\pm 0.05}$ |   $\mathbf{0.70_{\pm 0.05}}$   |   $0.52_{\pm 0.03}$   |   $0.84_{\pm 0.05}$   |    $0.44_{\pm 0.04}$    |    $0.40_{\pm 0.02}$    |    $0.68_{\pm 0.03}$    |     $\mathbf{0.42_{\pm 0.02}}$    |     $0.38_{\pm 0.02}$    |    $0.69_{\pm 0.04}$    |
>
> - From above results, our theoretical finding of the over-balancing issue can be verified by comparing performance between CT and CT w/o B:
>   - Traditional CT with balancing always has larger supervised error but smaller Wasserstein balancing error than CT w/o balancing;
>   - When confounding strength is large, i.e., $\gamma=3$, the reduction of Wasserstein balancing error in CT is larger than the increase of supervised error, making the balancing approach useful;
>   - **When confounding strength is small, i.e., $\gamma=0,1,2$, the reduction of Wasserstein balancing error in CT is smaller than the increase of supervised error, resulting in the over-balancing issue**.
>   - For example, when $\gamma=2$, the balancing approach in CT reduces the Wasserstein balancing error from 1.32 to 1.23 (decrease 0.09), but increases the supervised error from 0.93 to 1.24 (increase 0.31), i.e., the following equation holds:
> $$\|\mu_1 - \mu_0\|_{2}^{2} \leq \frac{v(\sqrt{\sigma_1} - \sqrt{\sigma_0})^2 r_2^2}{vr_1^2 - (2+\sigma_1 + \sigma_0)r_1^2r_3^2/4}. $$
>
> - Our theoretical finding of the superiority of the proposed Mamba-CDSP can be verified by comparing all 4 methods:
>   - Notably, Mamba-CDSP always has a summation of the Wasserstein balancing error and supervised error compared to CT, which leads to the optimal overall performance;
>   - Besides, the over-balancing phenomenon of Mamba-CDSP slightly exists in the case of $\gamma=0,1$.
>
> > **I am curious whether removing CDSP would mean better preservation?**
>
> Yes, but removing CDSP would also lead to larger error when correcting the confounding bias at the same time (see our detailed discussion above).
>
> > **Besides, for CT experiments on MIMIC-III, does w/o balancing simply mean keeping only the experimental results with α=0?**
>
> Yes, w/o balancing simply means setting $\alpha=0$ in the loss $(\hat \theta_Y, \hat \theta_R)=\arg \min_{\theta_Y, \theta_R} \mathcal L_{G_Y}(\theta_Y, \theta_R)+\alpha \mathcal L_\mathrm{conf}(\hat \theta_A, \theta_R)$ (see Eq. (18), Page 6, Section 4.3 in the original CT paper (Melnychuk et al, ICML 2022)).

---

> ### Author Response · Authors · 2024-12-02
> **We have added experiments that can verify our theoretical findings on the over-balancing issue and the tighter risk bound of our CDSP : )**
>
> Dear Reviewer PLDR,
>
> As the discussion deadline approaches, we are wondering whether our responses have properly addressed your concerns? Your feedback would be extremely helpful to us. If you have further comments or questions, we hope for the opportunity to respond to them.
>
> Many thanks,
>
> 7314 Authors

---

> > ### Author Response · Authors · 2024-12-03
> >
> > Dear reviewer PLDR,
> >
> > Since the discussion period will end in a few hours, we will be online waiting for your feedback on our rebuttal, which we believe has fully addressed your concerns.
> >
> > We would highly appreciate it if you could take into account our response when updating the rating and having discussions with AC and other reviewers.
> >
> > Thank you so much for your time and efforts. Sorry for our repetitive messages, but we're eager to ensure everything is addressed.
> >
> > Authors of # 7314

---

> > > ### Comment · Reviewer_PLDR · 2024-12-03
> > >
> > > Thank you for the author's response. I have already raised the score to 6.

---

> > > > ### Author Response · Authors · 2024-12-03
> > > > **Thank you for raising your score to 6!**
> > > >
> > > > Dear reviewer PLDR,
> > > >
> > > > Thank you for raising your score to 6! By now, we are appreciated that all reviewers are on the positive side of acceptance, except for one reviewer who uploaded a review of another ICLR submission : (
> > > >
> > > > Your multiple rounds of interaction have been invaluable to us in improving the quality of our manuscripts - thank you so much!
> > > >
> > > > Thanks for your time,
> > > >
> > > > Authors of # 7314

---

### Official Review · Reviewer_1NGR · 2024-11-01

**Soundness:** 3
**Presentation:** 3
**Contribution:** 2
**Rating:** 6
**Confidence:** 3

**Summary:**

The paper presents a novel approach to TCP using a Counterfactual Mamba model with Covariate-based Decorrelation towards Selective Parameters (Mamba-CDSP). It addresses the limitations of current methods, particularly those based on LSTM and Transformer architectures, in capturing long-sequence interactions effectively and efficiently. The proposed Mamba-CDSP aims to mitigate confounding bias by decorrelating current treatments from historical covariates and outcomes. The authors provide empirical experiment over both synthetic and real-world datasets, demonstrating that Mamba-CDSP significantly improves prediction performance and running efficiency compared to existing methods.

**Strengths:**

1. The motivation of this paper is well claimed.
2. The writing of this paper is good and easy to follow. The preliminaries part is particularly clear.
3. The experiment part is comprehensive, consisting of synthetic and real-world datasets.

**Weaknesses:**

1. The novelty and technical contribution of this paper are not enough. This paper claims to address the deconfounding and over-balance issues in the TCP task, but these issues are not first proposed by this paper. Below, I will further explain my opinion.
2. The authors claim that they are the first to consider the step-by-step deconfounding in the TCP task. However, to my knowledge, there are several existing works that achieve a similar goal, such as [1] and [2]. The authors should illustrate the difference between them.
3. Why would the linear correlation between $a_t$ and $h_{t-1}$ lead to the over-balancing issue while the non-linear correlation would not, as you remark in line 292? Can you provide a rational analysis regarding this method?
4. This method seems to simply replace the transformer with the Mamba model by slightly tailoring the backbone, which lacks technical contribution.
5. The sizes of the datasets used in the experiments are quite small; for example, there are only 5,000 patients in the MIMIC-III real-world dataset, which cannot adequately reveal the efficiency of replacing the Transformer with the Mamba model.



[1] Estimating Counterfactual Treatment Outcomes over Time through Adversarially Balanced Representations. ICLR 2020.

[2] Counterfactual Generative Models for Time-Varying Treatments. KDD 2024.

**Questions:**

See weaknesses.

---

> ### Author Response · Authors · 2024-11-29
>
> We sincerely appreciate the reviewer’s great efforts and insightful comments to improve our manuscript. We know that we are now approaching the end of the author-reviewer discussion and apologize for our late rebuttal. During the rebuttal period, we have been focusing on these beneficial suggestions from the reviewers and doing our best to add several theoretical analysis and experiments and revise our manuscript. We believe our current carefully revised manuscript can address all the reviewers’ concerns.
>
> > **W1: The novelty and technical contribution of this paper are not enough. This paper claims to address the deconfounding and over-balance issues in the TCP task, but these issues are not first proposed by this paper. Below, I will further explain my opinion.**
>
> **Response:** Thank you for the comment, we agree that these issues are not first proposed by this paper, but we cannot fully agree with the comment regarding novelty. To enhance technical contribution of our paper, we add extensive theoretical analysis with experimental validation as summarized below.
>
> ### **The novelty of this paper**
>
> - To the best of our knowledge, **this is the first work that adopts Mamba to address time-varying counterfactual prediction.** It should be noted that we are *not* simply replacing the transformer with the Mamba model.
> - From a deconfounding prospective, we propose **a novel de-correlation approach benefiting from Mamba for better trade-off between covariate balancing and covariate information preservation**, which distinguishes from the previous studies enforcing the independence operation in each time step.
> - Following the pioneer work focusing on the over-balancing issue in Huang et al. (ICML 2024), **we are the first work tackling the over-balancing issue in the task of time-varying counterfactual prediction.**
>
> ### **The added technical contribution during rebuttal**
>
> - For the **over-balancing issue in TCP** (W3), we **added theoretical analysis on the counterfactual prediction risk bounds**, revealing that **the intrinsic reason for the over-balancing issue is the profit of balancing in reducing the Wasserstein balancing error term will be smaller than the extra supervised prediction error due to the loss of covariate information** (see our response to W3).
>
> - For the **benefits of replace the transformer with the Mamba model by slightly tailoring the backbone** (W4), we **conduct more theoretical analysis showing that our CDSP method always has a tighter risk bound compared with the traditional adversarial balancing methods** (see our response to W4).
>
> - For the **efficiency of replacing the Transformer with the Mamba model** (W5), we **added theoretical computational complexity efficiency analysis on both CDSP and traditional adversarial balancing methods**, showing that **the overall complexity of our proposed CDSP method will be smaller than the previous methods**. We also **added experiments on the traning and inference efficiency to support our theoretical results** (see our response to W5).
>
> > **W2: The authors claim that they are the first to consider the step-by-step deconfounding in the TCP task. However, to my knowledge, there are several existing works that achieve a similar goal, such as [1] and [2]. The authors should illustrate the difference between them.**
>
> **Response:** Thank you for referring us two inspiring existing works and we found these works are beneficial to us! We would like to clarify that *we never claim that we are the first to consider the step-by-step deconfounding in the TCP task*, but have claimed that *this is the pioneer Mamba model tailored to counterfactual prediction* in Line 94 in our original draft.
>
> - For the referred two existing works, we first summarize the core idea of them as follows:
>   - [1] proposes to adapt the domain-invariant adversarial learning for TCP task, which enforces the adversarial loss to optimize each time-step.
>   - [2] is proposed for estimating the distribution of $Y$ rather than its mean, and mainly designed towards high-dimensional counterfactual generation.
> - As suggested by the reviewer, we illustrate the difference between our method and them mainly in two aspects:
>   - Our method indeed performs the **de-correlation operation in the last time step**, while both [1] and [2] enforce **the independence operation in each time step**. Considering the over-balancing phenomenons, both [1] and [2] will lead to **more serious deterioration of covariate information**.
>   - De-confounding contains many diverse techniques, e.g., matching, covariate balancing in [1], and re-weighting in [2], etc.. Different from [1] and [2], our paper proposes **a novel de-correlation approach benefiting from Mamba**, which leads to **efficient implementation with linear time complexity**.
> - Benefiting from the reviewer's suggestion, we **re-write the related-work section with a more thorough and complete review by adding over 15 new recent works in Section 2.**

---

> ### Author Response · Authors · 2024-11-29
>
> > **W3: Why would the linear correlation between $a_t$ and $h_{t-1}$ lead to the over-balancing issue while the non-linear correlation would not, as you remark in line 292? Can you provide a rational analysis regarding this method?**
>
> **Response:** Thank you for your detailed comments on the over-balancing issue. In below, we first clarify a misunderstanding that indeed **both the linear correlation and the non-linear correlation would lead to the over-balancing issue**. As suggested by the reviewer, we then **provide a comprehensive rational analysis regarding the over-balancing issue.** Finally, we **perform more simulations to support the current theoretical findings regarding the over-balancing issue.**
>
> ### **Clarification on a misunderstanding of the over-balancing issue in line 292**
>
> - First, we would like to clarify that our meaning is not *the linear correlation between $a_t$ and $h_{t-1}$ lead to the over-balancing issue while the non-linear correlation would not*. In fact, **both the linear correlation and the non-linear correlation would lead to the over-balancing issue**.
> - As mentioned in Huang et al. (ICML 2024), the over-balancing issue means that **enforcing treatment-independent representations of covariates lead to the loss of covariate information**, in which they conduct a critical empirical examination for the effectiveness of the balancing strategies within the realm of temporal counterfactual estimation in various settings on multiple datasets.
> - Formally, **treatment-independent balancing is equivalent to the alignment of from first-order moments to infinity-order moments of covariates (i.e., $E(X), E(X^2), \ldots$) simultaneously**.
> - In our paper, we propose to **align the first-moments of covariates across different treatment arms**, aiming to better trade-off the confounding bias elimination and covariate information preservation:
>   - On the one hand, the first-moments of covariates across different treatment arms **corrects the main component of the confounding bias**.
>   - On the other hand, it **preserves the high-order non-linear correlation between $A$ and $X$, which is beneficial for outcome prediction**.
>
> ### **Rational analysis regarding the over-balancing issue**
>
> - To measure the counterfactual prediction accuracy, we **adopt expected precision in Estimation of Counterfactual Prediction (ECP) as evaluation metric** defined as
> $\epsilon_\mathrm{ECP}(W, \Phi)=\int_{\mathcal{X}}\left(\hat{\tau}_f(x)-\tau(x)\right)^2 p(x) dx$.
>
> - Consider the Gaussian covariates $X_{\mid a}   \sim N(\mu_{a}, \sigma_{a}I)$ and linear outcome structure $Y(a) = W^{a}\Phi X_{\mid a} + \epsilon^{a}$, where $\epsilon^{a} \sim N(0,1)$. Let $N_{a}$ be the sample number of each treatment arm, $W$ and $\Phi$ be the model parameters of representation layers and prediction head, respectively.
>
> - The **risk bounds** of **(1) vanilla Empirical Risk Minimization (ERM) method without any balancing modules**, (2) **adversarial balancing (ADB) methods with the over-balancing issue** are derived as:
>   -  For **(1) vanilla ERM method without any balancing modules,** with probability at least $1-2\eta$, we have
> $$\epsilon_{ECP}^{vanilla}(W, \Phi) \leq 2(vr_1^2 |\mu_1 - \mu_0|^2_2 + v (\sqrt{\sigma_1} - \sqrt{\sigma_0})r_2^{2}+ C - (\sqrt{\frac{1}{\eta}}(\sum_ {j=1}^{2} \sqrt{\kappa_j})  + \overline{\sigma})),$$
> in which the **Wasserstein balancing error term** explains the risk due to **insufficient balancing in ERM**:
> $$2(vr_1^2 |\mu_1 - \mu_0|^2_2 + v (\sqrt{\sigma_1} - \sqrt{\sigma_0})r_2^{2}).$$
>   - For **(2) adversarial balancing (ADB) methods, with probability at least $1-2\eta$,** we have
> $$\epsilon_{\mathrm{ECP}}^{adb}(\tilde{W}, \tilde{\Phi}) \leq 2(\frac{(2+ \sigma_0 + \sigma_1)(r_1r_3)^{2}}{4}  \|\mu_1-\mu_0\|^2_2 + C - (\sqrt{\frac{1}{\eta}}( \sum_{j=1}^{2}\sqrt{\kappa_j}) + \overline{\sigma} ) ),$$
> in which the **supervised loss term** explains the risk due to the **loss of covariate information by overbalancing**:
> $$2\frac{(2+ \sigma_0 + \sigma_1)(r_1r_3)^{2}}{4}  \|\mu_1-\mu_0\|^2_2.$$
> - This shows a **trade-off between the factual outcomes fitting (ERM method) and covariate balancing (ADB method):**
>   - From our theory, the **advantage of balancing methods** is formulated as the **reduction of both the mean difference $|\mu_1 - \mu_0|^2_2$ and the variance difference $\sqrt{\sigma_1} - \sqrt{\sigma_0}$**, i.e., **distributional alignment** at the covariate representations;
>   - **When the Wasserstein balancing error term is small enough, i.e., $\|\mu_1 - \mu_0\|_{2}^{2}$ is small enough, the profit of balancing will be smaller than the extra supervised prediction error due to the loss of covariate information,** causing the performance drop of balancing methods compared to vanilla ERM methods.

---

> > ### Author Response · Authors · 2024-11-29
> >
> > ### **Experimental analysis regarding the over-balancing issue**
> >
> > To verify the current theoretical findings regarding the over-balancing issue as well as the tighter risk bound of our CDSP method, we **add synthetic experiments with varying confounding strength $\gamma$.**
> > - We adopt the following **three metrics**:
> >   - **Wasserstein balancing error** (named **Wasserstein** in the Table) in our derived bound arises from **insufficient balancing** $2(vr_1^2 |\mu_1 - \mu_0|^2_2 + v (\sqrt{\sigma_1} - \sqrt{\sigma_0})r_2^{2})$;
> >   - **Supervised error** (named **Supervised** in the Table) in our derived bound arises from the loss of covariate information by **over-balancing** $2\frac{(2+ \sigma_0 + \sigma_1)(r_1r_3)^{2}}{4}  \|\mu_1-\mu_0\|^2_2$;
> >   - **Overall TCP performance** (named **Overall** in the Table) containing **both factual and counterfactual prediction error**.
> > - We compare the following **four methods**:
> >   - **CT**: adversarial balanced Causal Transformer, in which $\alpha$ are tuned with best hyper-parameter search as in the original CT paper (Melnychuk et al, ICML 2022);
> >   - **CT w/o B**: CT without adversarial balancing by setting $\alpha_{CT}=0$;
> >   - **Mamba-CDSP** (ours): Our proposed methods with de-correlation;
> >   - **Mamba-CDSP w/o B**: Vanilla Mamba model without de-correltaion by setting $\alpha_{Mamba}=0$.
> >
> > | $\gamma$ | CT| CT | CT| CT w/o B | CT w/o B | CT w/o B | Mamba-CDSP | Mamba-CDSP | Mamba-CDSP | Mamba-CDSP w/o B | Mamba-CDSP w/o B | Mamba-CDSP w/o B |
> > |--|--|--|--|--|--|--|--|--|--|--|--|--|
> > | Metric   | Overall | Supervised | Wasserstein | Overall  | Supervised | Wasserstein  | Overall    | Supervised | Wasserstein | Overall     | Supervised | Wasserstein |
> > |     3    |  $\mathbf{1.66_{\pm 0.07}}$   | $1.53_{\pm 0.06}$ | $1.46_{\pm 0.12}$ |   $1.98_{\pm 0.18}$   |   $1.41_{\pm 0.10}$   |   $1.63_{\pm 0.12}$   |    $\mathbf{0.97_{\pm 0.04}}$    |    $0.91_{\pm 0.02}$    |    $1.06_{\pm 0.05}$    |    $1.15_{\pm 0.06}$    |     $0.85_{\pm 0.03}$    |    $1.14_{\pm 0.09}$    |
> > |     2    |  $1.56_{\pm 0.03}$    | $1.24_{\pm 0.05}$ | $1.23_{\pm 0.10}$ | $\mathbf{1.27_{\pm 0.11}}$ | $0.93_{\pm 0.07}$ | $1.32_{\pm 0.09}$ | $\mathbf{0.73_{\pm 0.04}}$ | $0.72_{\pm 0.03}$ | $0.88_{\pm 0.05}$ | $0.80_{\pm 0.04}$ | $0.67_{\pm 0.04}$ | $0.94_{\pm 0.04}$ |
> > |     1    |   $0.96_{\pm 0.09}$  | $0.76_{\pm 0.06}$ | $0.95_{\pm 0.06}$ |   $\mathbf{0.88_{\pm 0.06}}$   |   $0.62_{\pm 0.03}$   |   $1.06_{\pm 0.06}$   |    $0.65_{\pm 0.03}$    |    $0.58_{\pm 0.03}$    |    $0.78_{\pm 0.04}$    |     $\mathbf{0.64_{\pm 0.03}}$    |     $0.51_{\pm 0.02}$    |    $0.81_{\pm 0.05}$    |
> > |     0    |   $0.82_{\pm 0.04}$ | $0.68_{\pm 0.03}$ | $0.81_{\pm 0.05}$ |   $\mathbf{0.70_{\pm 0.05}}$   |   $0.52_{\pm 0.03}$   |   $0.84_{\pm 0.05}$   |    $0.44_{\pm 0.04}$    |    $0.40_{\pm 0.02}$    |    $0.68_{\pm 0.03}$    |     $\mathbf{0.42_{\pm 0.02}}$    |     $0.38_{\pm 0.02}$    |    $0.69_{\pm 0.04}$    |
> >
> > - From above results, our theoretical finding of the over-balancing issue can be verified by comparing performance between CT and CT w/o B:
> >   - Traditional CT with balancing always has larger supervised error but smaller Wasserstein balancing error than CT w/o balancing;
> >   - When confounding strength is large, i.e., $\gamma=3$, the reduction of Wasserstein balancing error in CT is larger than the increase of supervised error, making the balancing approach useful;
> >   - **When confounding strength is small, i.e., $\gamma=0,1,2$, the reduction of Wasserstein balancing error in CT is smaller than the increase of supervised error, resulting in the over-balancing issue**.
> >   - For example, when $\gamma=2$, the balancing approach in CT reduces the Wasserstein balancing error from 1.32 to 1.23 (decrease 0.09), but increases the supervised error from 0.93 to 1.24 (increase 0.31), i.e., the following equation holds:
> > $$\|\mu_1 - \mu_0\|_{2}^{2} \leq \frac{v(\sqrt{\sigma_1} - \sqrt{\sigma_0})^2 r_2^2}{vr_1^2 - (2+\sigma_1 + \sigma_0)r_1^2r_3^2/4}. $$
> >
> > - Our theoretical finding of the superiority of the proposed Mamba-CDSP can be verified by comparing all 4 methods:
> >   - Notably, Mamba-CDSP always has a summation of the Wasserstein balancing error and supervised error compared to CT, which leads to the optimal overall performance;
> >   - Besides, the over-balancing phenomenon of Mamba-CDSP slightly exists in the case of $\gamma=0,1$.

---

> ### Author Response · Authors · 2024-11-29
>
> > [W4. This method seems to simply replace the transformer with the Mamba model by slightly tailoring the backbone, which lacks technical contribution.]
>
> **Response:** We would like to emphasize that we are not simply replace the transformer with the Mamba model, **but also propose a novel de-correlation approach benefiting from Mamba for more effective and efficient trade-off between deconfounding and covariate information preservation,** which differentiates from the previous studies enforcing the independence operation in each time step.
>
> To enhance the technical contribution of our paper, we conduct more theoretical analysis showing the superiority of our propose Mamba-CDSP method.
>
> ### **[Main Technical Conclusion] CDSP has a tighter counterfactual prediction risk bound than traditional balancing approaches.**
> - Recap that for previous **adversarial balancing (ADB) methods, with probability at least $1-2\eta$,** we have
> $$\epsilon_{\mathrm{ECP}}^{adb}(\tilde{W}, \tilde{\Phi}) \leq 2(\frac{(2+ \sigma_0 + \sigma_1)(r_1r_3)^{2}}{4}  \|\mu_1-\mu_0\|^2_2 + C - (\sqrt{\frac{1}{\eta}}( \sum_{j=1}^{2}\sqrt{\kappa_j}) + \overline{\sigma} ) ).$$
> - For **our proposed CDSP method, with probability at least $1-2\eta$**, we have
> $$\epsilon_{\mathrm{ECP}}^{cdsp}(\tilde{W}, \tilde{\Phi}) \leq 2(\frac{(r_1r_3)^{2}}{2}  \|\mu_1-\mu_0\|^2_2 + C - (\sqrt{\frac{1}{\eta}}( \sum_{j=1}^{2}\sqrt{\kappa_j})+ \overline{\sigma} ) ).$$
> - With the above risk bounds, by noting
> $$
> \frac{(r_1 r_3)^2}{2}\|\mu_1-\mu_0\|^2_2 \leq \frac{(2+\sigma_0+\sigma_1)(r_1 r_3)^2}{4}\|\mu_1-\mu_0\|^2_2,
> $$
> we conclude that **our CDSP method has a tighter bound compared with the existing ADB methods,** i.e., $\epsilon_{\mathrm{ECP}}^{cdsp}(\tilde{W}, \tilde{\Phi}) \leq \epsilon_{\mathrm{ECP}}^{adb}(\tilde{W}, \tilde{\Phi})$.
>
> - We have **added the above theoretical analysis in a new Section 4.4** with detailed proofs in Appendix A.3.
>
> > [W5. The sizes of the datasets used in the experiments are quite small; for example, there are only 5,000 patients in the MIMIC-III real-world dataset, which cannot adequately reveal the efficiency of replacing the Transformer with the Mamba model.]
>
> **Response:** Thank you for the constructive comments. In below, we first **clarify a misunderstanding that the efficiency of Mamba compared to Transformer should be sequence length, instead of the sample size.** Next, we **analyze the computational complexity efficiency of our proposed CDSP method and traditional adversarial balancing methods from both theoretical and empirical prospectives,** showing the superiority of the CDSP method. Finally, we **add experiments on a new real-world time-series prediction dataset,** i.e., the M5 dataset, to **further validate the effectiveness of our method on the real-world datesets.**
>
> ### **Efficiency of replacing the Transformer with the Mamba model on the extremely long sequences**
>
> - We would like to clarify that the original Mamba paper as well as its follow up works demonstrate **the efficiency advantage of Mamba compared to Transformer is not sample size, but sequence length**.
> - In counterfactual prediction tasks, Mihaela van der Schaar et al. (ICML 2018) revealed that **the confounding effect will vanish with increasing sample size in the absence of unmeasured confounders** (see formal theoretical results in Section 3.2 and Figure 1 in Mihaela van der Schaar et al. (ICML 2018)). That is, **in the case of large sample regime, the counterfactual prediction will reduce to the trivial temporal forecasting problem.**
> - We highlight that **Mamba-based methods perform better than Transformer-based methods on extremely long sequences,** because the computational complexity of Transformer-based methods is squared-level $\mathcal{O}(L^2)$, which prevents setting the contextual window as the whole sequence length (1000 timesteps), but only as a subsequence length (200 timesteps) in practice due to the high computational cost.
> - To illustrate this point, we **have added experiments on extremely long sequences** using the TG simulator data with $L = 1000$ in Table 4, Page 10 in our revised paper.
>
> |    Method       | $\tau = 1$          |           $\tau = 2$          |        $\tau = 3$       |        $\tau = 4$       |        $\tau = 5$       |        $\tau = 6$        |
> |:--:|:--:|:--:|:--:|:--:|:--:|:--:|
> |  Causal Transformer     |             $1.82_{\pm 0.25}$ |             $2.51_{\pm 0.31}$ |       $2.77_{\pm 0.33}$ |       $2.89_{\pm 0.42}$ |       $3.32_{\pm 0.47}$ |        $3.85_{\pm 0.76}$ |
> | Mamba-CDSP | $\mathbf{0.51^*_{\pm \mathbf{0.02}}}$ | $\mathbf{0.86^*_{\pm \mathbf{0.06}}}$ | $\mathbf{1.05^*_{\pm 0.09}}$ | $\mathbf{1.37^*_{\pm 0.13}}$ | $\mathbf{1.54^*_{\pm 0.18}}$ | $\mathbf{1.93^*_{\pm 0.21}}$ |
>
> - From above results, **our proposed methods stably outperforms Causal Transformer at the significance level $\text{p-value}\leq 0.01$ at varying time-steps**.

---

> > ### Author Response · Authors · 2024-11-29
> >
> > ### **Theoretical computational complexity efficiency**
> >
> > - Denote the overall length of the time horizon as $T$, and the representation dimension of $A$ and $X$ as $d^h_A$ and $d^h_X$, respectively. For adversarial balancing modules, previous practice shows that the discriminator has usually $2$ layers with $d^{h}_{X}$ for each layer.
> > - For **our proposed CDSP method**, its complexity of the debias process can be divided into four stages:
> >   - Pre-computing the covariance terms, i.e., $\Sigma_{\tilde X^h_i,a_t}\Sigma_{\tilde X^h_i,a_t}^T$, before the beginning of the training process. Due to the complexity of matrices multiplication, we derive the complexity of this component as $\mathcal{O}(B((d^h_X)^2+d^h_A)T)$;
> >   - Computing the accumulated selective parameters $\mathbf K_i =\overline B_i \Pi_{j=i}^{t-1}\overline A_j$, which owns the complexity as $\mathcal{O}(B((d^h_X)^2)T)$;
> >   - Computing the multiplications between $\Sigma_{\tilde X^h_i,a_t}\Sigma_{\tilde X^h_i,a_t}^T$ and $\mathbf K_i = \overline B_i\Pi_{j=i}^{t-1}\overline A_j$, which owns the complexity as $\mathcal{O}(B(d^h_X)^3T)$;
> >   - Computing the 2-norm of the matrix as $\mathcal{O}(B(d_X^h)^3 T)$.
> > - Hence, **the overall complexity of our proposed CDSP methods is $\mathcal{O}(B((d_{X}^{h})^3 + d_{A}^{h})T)$,** with $B$ as a constant related to parameter $\overline{B}$ (see detailed derivation in Appendix A.4.3).
> > - For **traditional adversarial balancing methods**, the complexity can be divided into two parts:
> >   - Embedding representations of $X$ and $A$ with two layers of matrix multiplications, which owns the complexity of $\mathcal{O}(B((d^h_X)^3 + (d^h_X)^2 d^h_Y)T)$ in each round;
> >   - Computing the softmax and cross-entropy loss, which owns the complexity of $\mathcal{O}(B|A|T)$, where $|A|$ are the cardinal number of $A$.
> > - Hence, **the overall complexity of traditional adversarial balancing methods is $\mathcal{O}(B((d_X^h)^3 + (d_X^h)^2 d_Y^h + |A|)T)$.**
> > - From above, **the overall complexity of our proposed CDSP method will be smaller than the ADB methods if the representation dimension of $A$ denoted as $d_A^h$ is small, i.e., $d_A^h \leq (d_X^h)^2 d_Y^h + |A|$, which holds for the most cases in practice.**
> >
> > ### **Empirical computational complexity efficiency**
> >
> > - We also **added empirical computation complexity analysis**, and found **our method outperform traditional balancing methods in terms of both training time and inference time per sample in the context of deep models as backbone** (exclude MSMs, which uses Logistic regression as its backbone).
> >
> > | | Stages of training \& inference|Total runtime (in min)|Training time (in min) |Inference time (in min)  | Inference time per sample (in sec) |
> > |-|--|:--:|:--:|:--:|:--:|
> > | MSMs | 2 Logistic regressions for IPTW & linear regression | $1.5_{\pm 0.4}$ | $1.0_{\pm 0.1}$ | $0.5_{\pm 0.2}$ | $0.04$ |
> > | RMSNs | 2 networks for IPTW & encoder & decoder | $38_{\pm 3.9}$ | $35_{\pm 0.1}$ | $3_{\pm 0.2}$ | $0.24$ |
> > | CRN | encoder & decoder | $109_{\pm 10.3}$ | $84_{\pm 0.1}$ | $25_{\pm 0.2}$ | $2$ |
> > | G-Net | single network & MC sampling for inference | $89_{\pm 5.2}$ | $86_{\pm 0.1}$ | $3_{\pm 0.2}$ | $0.24$ |
> > | CT | single multi-input network | $156$ | $98_{\pm 0.1}$ | $58_{\pm 0.1}$ | $4.64$ |
> > | Ours | Single Mamba Block | $12_{\pm 2.3}$ | $10_{\pm 0.1}$ | $2_{\pm 0.1}$ | $0.16$ |
> >
> > ### **Add real-world experiments on the M5 dataset**
> >
> > - We also follow Huang et al. (ICML 2024) to **add experiments on a new real-world time-series prediction dataset,** i.e., the **M5 dataset,** which provided by Walmart, captures the unit sales data of a diverse range of products sold across the United States, structured as grouped time series.
> >
> > |Models|$\tau = 1$|$\tau = 2$|$\tau = 3$|$\tau = 4$|$\tau = 5$|$\tau = 6$|
> > |:--:|:--:|:---:|:--:|:--:|:--:|:--:|
> > |RMSN|$5.19_{\pm 0.12}$ |       $9.73_{\pm 0.26}$ |      $10.48_{\pm 0.34}$ |      $11.07_{\pm 0.43}$ |      $11.63_{\pm 0.53}$ |      $12.16_{\pm 0.65}$ |
> > |CRN |$4.98_{\pm 0.32}$ |       $9.15_{\pm 0.17}$ |$9.79_{\pm 0.16}$ |      $10.12_{\pm 0.17}$ |      $10.38_{\pm 0.19}$ |      $10.59_{\pm 0.22}$ |
> > |CRN w/o Balancing           |             $4.68_{\pm 0.06}$ |       $9.05_{\pm 0.16}$ |       $9.68_{\pm 0.15}$ |      $10.00_{\pm 0.16}$ |$10.26_{\pm 0.18}$ |      $10.47_{\pm 0.20}$ |
> > |Causal Transformer |$4.59_{\pm 0.08}$ |       $8.99_{\pm 0.19}$ |       $9.59_{\pm 0.19}$ |       $9.91_{\pm 0.23}$ |      $10.15_{\pm 0.25}$ |      $10.35_{\pm 0.28}$ |
> > |Causal Transformer w/o Balancing |$4.59_{\pm 0.09}$ |       $8.98_{\pm 0.18}$ |       $9.58_{\pm 0.18}$ |       $9.90_{\pm 0.20}$ |$10.13_{\pm 0.23}$ |      $10.34_{\pm 0.26}$ |
> > |Mamba-CDSP           | $\mathbf{ 4.08^*_{\pm 0.06}}$ | $\mathbf{ 4.05^*_{\pm 0.13}}$ | $\mathbf{ 4.59^*_{\pm 0.15}}$ | $\mathbf{ 5.87^*_{\pm 0.17}}$ | $\mathbf{6.47^*_{\pm 0.20}}$ | $\mathbf{8.39^*_{\pm 0.28}}$ |
> >
> > - From the above results, we further validate the effectiveness of our Mamba-CDSP on the real-world datesets.

---

> ### Author Response · Authors · 2024-11-29
>
> **Please understand and forgive our delayed response to the updated review, as we have been working to supplement the extensive technical contributions and experiments to fully address your concerns. We would really appreciate it if you would like to kindly consider adjusting your score.** We look forward to your insightful and constructive responses to further help us improve the quality of our work. Thank you!
> ***
>
> **References**
>
> [1] Bica et al, Estimating counterfactual treatment outcomes over time through adversarially balanced representations, ICLR 2020.
>
> [2] Wu et al, Counterfactual Generative Models for Time-Varying Treatments. KDD 2024.
>
> [3] Huang et al, An Empirical Examination of Balancing Strategy for Counterfactual Estimation on Time Series, ICML 2024.
>
> [4] Melnychuk et al, Causal transformer for estimating counterfactual outcomes, ICML 2022.
>
> [5] Ahmed M. Alaa and Mihaela van der Schaar, Limits of Estimating Heterogeneous Treatment Effects, ICML 2018.

---

> > ### Author Response · Authors · 2024-12-02
> >
> > Dear Reviewer 1NGR,
> >
> > Once again, we are grateful for your time and effort for reviewing our paper. Since the discussion period will end in around a day, we are very eager to get your feedback on our response. We understand that you are very busy, but we would highly appreciate it if you could take into account our response when updating the rating and having a discussion with AC and other reviewers.
> >
> > Thanks for your time,
> >
> > Authors of # 7314

---

> > > ### Author Response · Authors · 2024-12-03
> > >
> > > Dear reviewer 1NGR,
> > >
> > > Since the discussion period will end in a few hours, we will be online waiting for your feedback on our rebuttal, which we believe has fully addressed your concerns.
> > >
> > > We would highly appreciate it if you could take into account our response when updating the rating and having discussions with AC and other reviewers.
> > >
> > > Thank you so much for your time and efforts. Sorry for our repetitive messages, but we're eager to ensure everything is addressed.
> > >
> > > Authors of # 7314

---

> ### Comment · Reviewer_1NGR · 2024-12-03
>
> Thanks for your detailed reply, some of my concerns have been solved, but there are also some concerns still exist. Due to the detailed response, I have raised my score to 6.

---

> > ### Author Response · Authors · 2024-12-03
> > **Thank you for recognizing our detailed reply and raising your score to the positive side!**
> >
> > Dear reviewer 1NGR,
> >
> > We are encouraged by your recognition of our detailed responses and your raised score to the positive side! We would be grateful if you could continue to support us during the upcoming AC-reviewer discussion period -- thank you so much!
> >
> > Authors of # 7314

---

### Official Review · Reviewer_FCGz · 2024-11-04

**Soundness:** 3
**Presentation:** 3
**Contribution:** 3
**Rating:** 6
**Confidence:** 3

**Summary:**

This paper introduces Mamba-CDSP, a novel approach for time-varying counterfactual prediction (TCP) using state-space models. The key innovation lies in combining the Mamba architecture (a recent advance in state-space modeling) with a new Covariate-based Decorrelation towards Selective Parameters (CDSP) mechanism. The method addresses two major challenges in TCP: computational efficiency and the over-balancing problem. The authors demonstrate superior performance over existing methods like Causal Transformer and G-Net on both synthetic and real-world datasets.

**Strengths:**

1. Technical Innovation:

-Novel combination of Mamba architecture with TCP

-Well-designed CDSP mechanism that addresses known limitations

-Efficient implementation with linear time complexity

2. Practical Value:

-Better handling of long sequences

-Improved computational efficiency

-Real-world applicability demonstrated on MIMIC-III dataset


3.Experimental Design:

-Comprehensive ablation studies

-Multiple evaluation scenarios

-Reasonable baseline comparisons

**Weaknesses:**

1. Theoretical Analysis:

-Limited theoretical justification for why CDSP works better than traditional balancing

-Could benefit from more formal analysis of the bias-variance trade-off

2.Empirical Validation:

-Could benefit from more diverse real-world datasets

-Limited discussion of failure cases

-More detailed hyperparameter sensitivity analysis needed

**Questions:**

1. How sensitive is the CDSP mechanism to the choice of decorrelation threshold?
2. Could the authors provide more insight into the computational complexity trade-offs between CDSP and traditional balancing methods?
3. How does the method perform on extremely long sequences (e.g., >1000 timesteps)?

---

> ### Author Response · Authors · 2024-11-27
>
> We sincerely appreciate the reviewer’s great efforts and insightful comments to improve our manuscript. We thank the reviewer for recognizing our technical innovation, practical value, as well as experimental design. In below, we address these concerns point by point and try our best to update the manuscript accordingly.
>
> > **W1: Limited theoretical analysis.**
>
> **Response:** We thank the reviewer for pointing out this issue, and we have added the following theoretical analysis to show the superiority of the proposed CDSP method than traditional balancing approaches.
>
> ### **Main Theoretical Result: CDSP has a tighter counterfactual prediction risk bound than traditional balancing approaches.**
> - To measure the counterfactual prediction accuracy, we **adopt expected precision in Estimation of Counterfactual Prediction (ECP) as evaluation metric** defined as
> $\epsilon_\mathrm{ECP}(W, \Phi)=\int_{\mathcal{X}}\left(\hat{\tau}_f(x)-\tau(x)\right)^2 p(x) dx$.
>
> - Consider the Gaussian covariates $X_{\mid a}   \sim N(\mu_{a}, \sigma_{a}I)$ and linear outcome structure $Y(a) = W^{a}\Phi X_{\mid a} + \epsilon^{a}$, where $\epsilon^{a} \sim N(0,1)$. Let $N_{a}$ be the sample number of each treatment arm, $W$ and $\Phi$ be the model parameters of representation layers and prediction head, respectively.
>
> - The **risk bounds** of **(1) vanilla Empirical Risk Minimization (ERM) method without any balancing modules**, (2) **adversarial balancing (ADB) methods,** and **(3) our proposed CDSP method** are derived as:
>   -  For **(1) vanilla ERM method without any balancing modules,** with probability at least $1-2\eta$, we have
> $$\epsilon_{ECP}^{vanilla}(W, \Phi) \leq 2(vr_1^2 |(\mu_1 - \mu_0)|^2_2 + v (\sqrt{\sigma_1} - \sqrt{\sigma_0})r_2^{2} + C - (\sqrt{\frac{1}{\eta}}(\sum_ {j=1}^{2} \sqrt{\kappa_j})  + \overline{\sigma})),$$
> where $\overline{\sigma} = \frac{1}{N_1} \sum_{i=1}^{N_1}{\epsilon^a_{i}}^2$.
>   - For **(2) adversarial balancing (ADB) methods, with probability at least $1-2\eta$,** we have
> $$\epsilon_{\mathrm{ECP}}^{adb}(\tilde{W}, \tilde{\Phi}) \leq 2(\frac{(2+ \sigma_0 + \sigma_1)(r_1r_3)^{2}}{4}  \|\mu_1-\mu_0\|^2_2 + C - (\sqrt{\frac{1}{\eta}}( \sum_{j=1}^{2}\sqrt{\kappa_j}) + \overline{\sigma} ) ).$$
>   - For **(3) our proposed CDSP method, with probability at least $1-2\eta$**, we have
> $$\epsilon_{\mathrm{ECP}}^{cdsp}(\tilde{W}, \tilde{\Phi}) \leq 2(\frac{(r_1r_3)^{2}}{2}  \|\mu_1-\mu_0\|^2_2 + C - (\sqrt{\frac{1}{\eta}}( \sum_{j=1}^{2}\sqrt{\kappa_j})+ \overline{\sigma} ) ).$$
> - With the above risk bounds, we then derive two main results:
>   - **(a) Over-balancing of ADB methods:** When the following condition holds, the risk bound of vanilla ERM model is more tighter than that of adversarial balancing methods:
> $$\|\mu_1 - \mu_0\|_{2}^{2} \leq \frac{v(\sqrt{\sigma_1} - \sqrt{\sigma_0})^2 r_2^2}{vr_1^2 - (2+\sigma_1 + \sigma_0)r_1^2r_3^2/4}. $$
> This illustrates that **as the gap between two groups decreases, the harm of over-balancing is much larger than that of observed confounding bias.**
>   - **(b) CDSP outperforms ADB with a tighter bound:** Our CDSP method has a tighter bound compared with the existing ADB methods, i.e., $\epsilon_{\mathrm{ECP}}^{cdsp}(\tilde{W}, \tilde{\Phi}) \leq \epsilon_{\mathrm{ECP}}^{adb}(\tilde{W}, \tilde{\Phi})$:
> - We have **added the above theoretical analysis in a new Section 4.4** with detailed proofs in Appendix A.3.
>
> > **W2.1: Empirical Validation - Could benefit from more diverse real-world datasets**
>
> **Response:** As suggested by the reviewer, we follow Huang et al. (ICML 2024) to **add experiments on a new real-world time-series prediction dataset,** i.e., the **M5 dataset,** which provided by Walmart, captures the unit sales data of a diverse range of products sold across the United States, structured as grouped time series.
>
> |Models|$\tau = 1$ |$\tau = 2$|$\tau = 3$ | $\tau = 4$| $\tau = 5$|$\tau = 6$ |
> |:--:|:--:|:--:|:--:|:--:|:--:|:--:|
> |RMSN |$5.19_{\pm 0.12}$|$9.73_{\pm 0.26}$|$10.48_{\pm 0.34}$|$11.07_{\pm 0.43}$ |      $11.63_{\pm 0.53}$ |      $12.16_{\pm 0.65}$ |
> |  CRN|$4.98_{\pm 0.32}$ |       $9.15_{\pm 0.17}$|$9.79_{\pm 0.16}$ | $10.12_{\pm 0.17}$ |      $10.38_{\pm 0.19}$ |      $10.59_{\pm 0.22}$ |
> | CRN w/o Balancing|$4.68_{\pm 0.06}$|$9.05_{\pm 0.16}$|$9.68_{\pm 0.15}$ |$10.00_{\pm 0.16}$|$10.26_{\pm 0.18}$ |      $10.47_{\pm 0.20}$ |
> |Causal Transformer|$4.59_{\pm 0.08}$|$8.99_{\pm 0.19}$|$9.59_{\pm 0.19}$|$9.91_{\pm 0.23}$|$10.15_{\pm 0.25}$ |      $10.35_{\pm 0.28}$ |
> | Causal Transformer w/o Balancing|$4.59_{\pm 0.09}$|$8.98_{\pm 0.18}$|$9.58_{\pm 0.18}$|$9.90_{\pm 0.20}$ |      $10.13_{\pm 0.23}$ |      $10.34_{\pm 0.26}$ |
> |Mamba-CDSP |$\mathbf{ 4.08^*_{\pm 0.06}}$ | $\mathbf{ 4.05^*_{\pm 0.13}}$ | $\mathbf{ 4.59^*_{\pm 0.15}}$ | $\mathbf{ 5.87^*_{\pm 0.17}}$ | $\mathbf{6.47^*_{\pm 0.20}}$ | $\mathbf{8.39^*_{\pm 0.28}}$ |
>
> - From the above results, we further validate the effectiveness of our Mamba-CDSP on the real-world datesets.

---

> ### Author Response · Authors · 2024-11-28
>
> > **W2.2: Empirical Validation - Limited discussion of failure cases**
>
> **Response:** Thanks for your beneficial advice! We **have added more discussion of failure cases in Section 5.3, P10, in which our method does not have significant improvement over Causal Transformer.**
>  - A critical observation is that **our proposed Mamba-CDSP achieves near performance on the TG simulator (synthetic dataset) and the semi-synthetic MIMIC-III dataset when the prediction step $\tau$ is small,** e.g., $\tau=1$.
> - The first reason is **confounding effect will accumulate along with the time step, and the bias in initial steps are not significant enough.**
> - The second reason is the **contextual window of the Causal Transformer (CT)** is normally set to the length of the whole sequence when the sequence length is short, but **cannot set to the length of the whole sequence when the sequence length is long (e.g., $L\geq 1000$), due to the computational complexity is squared-level $\mathcal{O}(L^2)$.** This limits the prediction performance of CT in the long sequence scenarios.
>
> > **W2.3: Empirical Validation - More detailed hyperparameter sensitivity analysis needed**
>
> **Response:** As suggested by the reviewer, we conduct more experiments including **(i) sensitivity analysis on the choice of decorrelation threshold $\alpha$ on the semi-synthetic MIMIC-III dataset** (see our response to Q1) and **(ii) sensitivity analysis on the confounding strength $\gamma$ on the synthetic dataset.**
>
> - To validate that our method can correct for confounding bias, we **added synthetic experiments with increasing confounding bias on the tumor growth (TG) simulator.**
> - We follow Melnychuk, et al. (ICML 2022) to **control the level of confounding via tuning $\gamma$** in Eq. (33) $\mathbf{A}^c_t, \mathbf{A}^r_t \sim \operatorname{Bernoulli}( \sigma ( \frac{\gamma}{D_{max}})(D_{15}(Y_{t-1}) - D_{max} / 2))$. For $\gamma=0$, the treatment assignment is fully randomized. **For increasing values, the amount of time-varying confounding becomes also larger.**
>
> | $\gamma$ |    Model   |     $\tau = 1$     |     $\tau = 2$     |     $\tau = 3$     |     $\tau = 4$     |     $\tau = 5$     |
> |----------|:----------:|:------------------:|:------------------:|:------------------:|:------------------:|:------------------:|
> | 8        |     Causal Transformer     |  $2.98_{\pm 0.01}$ |  $3.46_{\pm 0.01}$ |  $3.56_{\pm 0.01}$ |  $3.52_{\pm 0.01}$ |  $3.33_{\pm 0.01}$ |
> | 8        | Mamba-CDSP |  $\mathbf{2.97_{\pm 0.01}}$ |  $\mathbf{3.36^*_{\pm 0.01}}$ |  $\mathbf{3.29^*_{\pm 0.01}}$ |  $\mathbf{3.02^*_{\pm 0.01}}$ |  $\mathbf{2.69^*_{\pm 0.01}}$ |
> | 10       |     Causal Transformer     | $10.46_{\pm 0.01}$ |  $9.38_{\pm 0.01}$ |  $8.50_{\pm 0.01}$ |  $7.63_{\pm 0.01}$ |  $6.83_{\pm 0.01}$ |
> | 10       | Mamba-CDSP |  $\mathbf{4.25^*_{\pm 0.01}}$ |  $\mathbf{5.26^*_{\pm 0.01}}$ |  $\mathbf{5.58^*_{\pm 0.01}}$ |  $\mathbf{5.55^*_{\pm 0.01}}$ |  $\mathbf{5.40^*_{\pm 0.01}}$ |
> | 16       |     Causal Transformer     | $15.93_{\pm 0.01}$ | $13.54_{\pm 0.01}$ | $15.54_{\pm 0.01}$ | $16.40_{\pm 0.01}$ | $16.58_{\pm 0.01}$ |
> | 16       | Mamba-CDSP |  $\mathbf{8.99^*_{\pm 0.01}}$ |  $\mathbf{9.06^*_{\pm 0.01}}$ |  $\mathbf{8.09^*_{\pm 0.01}}$ |  $\mathbf{6.93^*_{\pm 0.01}}$ |  $\mathbf{5.95^*_{\pm 0.01}}$ |
> | 20       |     Causal Transformer     | $16.19_{\pm 0.01}$ | $15.84_{\pm 0.01}$ | $17.09_{\pm 0.01}$ | $16.55_{\pm 0.01}$ | $18.32_{\pm 0.01}$ |
> | 20       | Mamba-CDSP | $\mathbf{14.14^*_{\pm 0.01}}$ | $\mathbf{12.73^*_{\pm 0.01}}$ | $\mathbf{10.26^*_{\pm 0.01}}$ |  $\mathbf{8.21^*_{\pm 0.01}}$ |  $\mathbf{6.52^*_{\pm 0.01}}$ |
>
> - From above results, we conclude that **our proposed method significantly outperform with $\text{p-value} \leq 0.01$ using the paired-t-test compared with Causal Transformer for correcting the confounding bias.** We have supplemented such results in our revised paper (Table 9, P24).

---

> > ### Author Response · Authors · 2024-11-28
> >
> > > **Q1: How sensitive is the CDSP mechanism to the choice of decorrelation threshold?**
> >
> > **Response:** As suggested by the reviewer, we **add sensitive analysis to study the choice of decorrelation hyperparameter $\alpha$ with varying time-steps $\tau \in \\{1, \ldots, 10\\}$ on the semi-synthetic MIMIC-III dataset.**
> >
> > - The higher decorrelation hyperparameter $\alpha$, the stronger de-correlation is performed to regularize the prediction process.
> >
> > | $\alpha$|$\tau = 1$|$\tau = 2$|$\tau = 3$|$\tau = 4$|$\tau = 5$|$\tau = 6$|$\tau = 7$|$\tau = 8$|$\tau = 9$|$\tau = 10$|
> > |--|:--:|:--:|:--:|:--:|:--:|:--:|:---:|:--:|:--:|:--:|
> > |0|$0.32_{\pm 0.01}$ | $0.43_{\pm 0.01}$ | $0.54_{\pm 0.01}$ | $0.65_{\pm 0.01}$ | $0.75_{\pm 0.01}$ | $0.85_{\pm 0.01}$ | $0.93_{\pm 0.01}$ | $1.02_{\pm 0.01}$ | $1.09_{\pm 0.01}$ | $1.17_{\pm 0.01}$ |
> > | 0.0001|$0.30_{\pm 0.01}$ | $0.37_{\pm 0.01}$ | $0.44_{\pm 0.01}$ | $0.49_{\pm 0.01}$ | $0.53_{\pm 0.01}$ | $0.57_{\pm 0.01}$ | $0.60_{\pm 0.01}$ | $0.62_{\pm 0.01}$ | $0.64_{\pm 0.01}$ | $0.65_{\pm 0.01}$ |
> > | 0.001|$0.27_{\pm 0.01}$ | $0.33_{\pm 0.01}$ | $0.39_{\pm 0.01}$ | $0.43_{\pm 0.01}$ | $0.46_{\pm 0.01}$ | $0.49_{\pm 0.01}$ | $0.52_{\pm 0.01}$ | $0.54_{\pm 0.01}$ | $0.56_{\pm 0.01}$ | $0.58_{\pm 0.01}$ |
> > | 0.01|$0.25_{\pm 0.01}$ | $0.32_{\pm 0.01}$ | $0.35_{\pm 0.01}$ | $0.44_{\pm 0.01}$ | $0.42_{\pm 0.01}$ | $0.46_{\pm 0.01}$ | $0.47_{\pm 0.01}$ | $0.48_{\pm 0.01}$ | $0.53_{\pm 0.01}$ | $0.53_{\pm 0.01}$ |
> > | 0.1|$0.21_{\pm 0.01}$ | $0.28_{\pm 0.01}$ | $0.33_{\pm 0.01}$ | $0.41_{\pm 0.01}$ | $0.39_{\pm 0.01}$ | $0.45_{\pm 0.01}$ | $0.45_{\pm 0.01}$ | $0.45_{\pm 0.01}$ | $0.47_{\pm 0.01}$ | $0.49_{\pm 0.01}$ |
> > | 1.0|$\mathbf{0.19_{\pm 0.01}}$ |  $\mathbf{0.25_{\pm 0.01}}$  |  $\mathbf{0.30_{\pm 0.01}}$|$\mathbf{0.34_{\pm 0.01}}$  |  $\mathbf{0.37_{\pm 0.01}}$  |  $\mathbf{0.42_{\pm 0.01}}$|$\mathbf{0.43_{\pm 0.01}}$|$\mathbf{0.44_{\pm 0.01}}$  |  $\mathbf{0.46_{\pm 0.01}}$  |  $\mathbf{0.47_{\pm 0.01}}$|
> > | 10.0|$0.23_{\pm 0.01}$ | $0.27_{\pm 0.01}$ | $0.34_{\pm 0.01}$|$0.36_{\pm 0.01}$|$0.38_{\pm 0.01}$ | $0.44_{\pm 0.01}$ | $0.57_{\pm 0.01}$ | $0.59_{\pm 0.01}$ | $0.52_{\pm 0.01}$ | $0.50_{\pm 0.01}$ |
> >
> > - From above results, we found that **our method is not sensitive to the choice of decorrelation hyperparameters when $\alpha$ is within a proper range [0.1, 10], and our method stably perform the best when choosing $\alpha=1.0$.** We have supplemented such results in our revised paper (Table 3, P9).
> >
> > > **Q2: Could the authors provide more insight into the computational complexity trade-offs between CDSP and traditional balancing methods?**
> >
> > **Response:** We **added both theoretical analysis and empirical validations on computation complexity trade-offs between CDSP and traditional balancing methods**.
> >
> > ### **Theoretical Computational Complexity Analysis**
> >
> > - Denote the overall length of the time horizon as $T$, and the representation dimension of $A$ and $X$ as $d^h_A$ and $d^h_X$, respectively. For adversarial balancing modules, previous practice shows that the discriminator has usually $2$ layers with $d^{h}_{X}$ for each layer.
> > - For **our proposed CDSP methods**, its complexity of the debias process can be divided into four stages:
> >   - Pre-computing the covariance terms, i.e., $\Sigma_{\tilde X^h_i,a_t}\Sigma_{\tilde X^h_i,a_t}^T$, before the beginning of the training process. Due to the complexity of matrices multiplication, we derive the complexity of this component as $\mathcal{O}(B((d^h_X)^2+d^h_A)T)$;
> >   - Computing the accumulated selective parameters $\mathbf K_i =\overline B_i \Pi_{j=i}^{t-1}\overline A_j$, which owns the complexity as $\mathcal{O}(B((d^h_X)^2)T)$;
> >   - Computing the multiplications between $\Sigma_{\tilde X^h_i,a_t}\Sigma_{\tilde X^h_i,a_t}^T$ and $\mathbf K_i = \overline B_i\Pi_{j=i}^{t-1}\overline A_j$, which owns the complexity as $\mathcal{O}(B(d^h_X)^3T)$;
> >   - Computing the 2-norm of the matrix as $\mathcal{O}(B(d_X^h)^3 T)$.
> > - Hence, **the overall complexity of our proposed CDSP methods is $\mathcal{O}(B((d_{X}^{h})^3 + d_{A}^{h})T)$,** with $B$ as a constant related to parameter $\overline{B}$ (see detailed derivation in Appendix A.4.3).
> > - For **traditional adversarial balancing methods**, the complexity can be divided into two parts:
> >   - Embedding representations of $X$ and $A$ with two layers of matrix multiplications, which owns the complexity of $\mathcal{O}(B((d^h_X)^3 + (d^h_X)^2 d^h_Y)T)$ in each round;
> >   - Computing the softmax and cross-entropy loss, which owns the complexity of $\mathcal{O}(B|A|T)$, where $|A|$ are the cardinal number of $A$.
> > - Hence, **the overall complexity of traditional adversarial balancing methods is $\mathcal{O}(B((d_X^h)^3 + (d_X^h)^2 d_Y^h + |A|)T)$.**
> > - From above, **the overall complexity of our proposed CDSP method will be smaller than the ADB methods if the representation dimension of $A$ denoted as $d_A^h$ is small, i.e., $d_A^h \leq (d_X^h)^2 d_Y^h + |A|$, which holds for the most cases in practice.**

---

> > > ### Author Response · Authors · 2024-11-28
> > >
> > > ### **Empirical Computational Complexity Analysis**
> > >
> > > - We also **added empirical computation complexity analysis**, and found **our method outperform traditional balancing methods in terms of both training time and inference time per sample in the context of deep models as backbone** (exclude MSMs, which uses Logistic regression as its backbone).
> > >
> > > |       | Stages of training \& inference                      |      Total runtime (in min)      |  Training time (in min)   |  Inference time (in min)  | Inference time per sample (in sec) |
> > > |-------|------------------------------------------------------|:--------------------------------:|:---------------:|:---------------:|:-------------:|
> > > | MSMs | 2 Logistic regressions for IPTW & linear regression | $1.5_{\pm 0.4}$ | $1.0_{\pm 0.1}$ | $0.5_{\pm 0.2}$ | $0.04$ |
> > > | RMSNs | 2 networks for IPTW & encoder & decoder | $38_{\pm 3.9}$ | $35_{\pm 0.1}$ | $3_{\pm 0.2}$ | $0.24$ |
> > > | CRN | encoder & decoder | $109_{\pm 10.3}$ | $84_{\pm 0.1}$ | $25_{\pm 0.2}$ | $2$ |
> > > | G-Net | single network & MC sampling for inference | $89_{\pm 5.2}$ | $86_{\pm 0.1}$ | $3_{\pm 0.2}$ | $0.24$ |
> > > | CT | single multi-input network | $156$ | $98_{\pm 0.1}$ | $58_{\pm 0.1}$ | $4.64$ |
> > > | Ours | Single Mamba Block | $12_{\pm 2.3}$ | $10_{\pm 0.1}$ | $2_{\pm 0.1}$ | $0.16$ |
> > >
> > > > **Q3: How does the method perform on extremely long sequences (e.g., >1000 timesteps)?**
> > >
> > > **Response:** Thanks for your inspiring comments! We **have added experiments on the TG simulator data with $L = 1000$ in Table 4, Page 10 in our revised paper.**
> > >
> > > |    Method       |           $\tau = 1$          |           $\tau = 2$          |        $\tau = 3$       |        $\tau = 4$       |        $\tau = 5$       |        $\tau = 6$        |
> > > |:----------:|:-----------------------------:|:-----------------------------:|:-----------------------:|:-----------------------:|:-----------------------:|:------------------------:|
> > > |  Causal Transformer     |             $1.82_{\pm 0.25}$ |             $2.51_{\pm 0.31}$ |       $2.77_{\pm 0.33}$ |       $2.89_{\pm 0.42}$ |       $3.32_{\pm 0.47}$ |        $3.85_{\pm 0.76}$ |
> > > | Mamba-CDSP | $\mathbf{0.51^*_{\pm \mathbf{0.02}}}$ | $\mathbf{0.86^*_{\pm \mathbf{0.06}}}$ | $\mathbf{1.05^*_{\pm 0.09}}$ | $\mathbf{1.37^*_{\pm 0.13}}$ | $\mathbf{1.54^*_{\pm 0.18}}$ | $\mathbf{1.93^*_{\pm 0.21}}$ |
> > >
> > > - From above results, our proposed methods stably outperforms Causal Transformer at the significance level $\text{p-value}\leq 0.01$ at varying time-steps.
> > >
> > > - We highlight that **Mamba-based methods perform better than Transformer-based methods on extremely long sequences,** because the computational complexity of Transformer-based methods is squared-level $\mathcal{O}(L^2)$, which prevents setting the contextual window as the whole sequence length (1000 timesteps), but only as a subsequence length (200 timesteps) in practice due to the high computational cost.
> > >
> > > ***
> > > **We hope the above discussion will fully address your concerns about our work, and we would really appreciate it if you would like to kindly consider adjusting your score.** We look forward to your insightful and constructive responses to further help us improve the quality of our work. Thank you!
> > > ***
> > >
> > > **References**
> > >
> > > [1] Huang et al, An Empirical Examination of Balancing Strategy for Counterfactual Estimation on Time Series, ICML 2024.
> > >
> > > [2] Melnychuk et al, Causal transformer for estimating counterfactual outcomes, ICML 2022.

---

> ### Author Response · Authors · 2024-12-02
>
> Dear Reviewer FCGz,
>
> Once again, we are grateful for your time and effort for reviewing our paper. Since the discussion period will end in around a day, we are wondering whether our responses have properly addressed your concerns? Your feedback would be extremely helpful to us. If you have further comments or questions, we hope for the opportunity to respond to them.
>
> Many thanks,
>
> 7314 Authors

---

> > ### Author Response · Authors · 2024-12-03
> >
> > Dear reviewer FCGz,
> >
> > Since the discussion period will end in a few hours, we will be online waiting for your feedback on our rebuttal, which we believe has fully addressed your concerns.
> >
> > We would highly appreciate it if you could take into account our response when updating the rating and having discussions with AC and other reviewers.
> >
> > Thank you so much for your time and efforts. Sorry for our repetitive messages, but we're eager to ensure everything is addressed.
> >
> > Authors of # 7314

---

### Official Review · Reviewer_TFBs · 2024-11-04

**Soundness:** 2
**Presentation:** 2
**Contribution:** 3
**Rating:** 6
**Confidence:** 2

**Summary:**

The paper works with a time varying counterfactual prediction method using STATE-SPACE model.  It introduces methods that de-correlate between current treatment and historical covariates. They claimed that their model is effective and lightweight. Finally, they performed experiments on several datasets to highlight the efficacy of their method.

**Strengths:**

(1) New problem on TCP on state-space model

(2) Design of novel de-correlation mechanism to reduce confounding bias.

**Weaknesses:**

(1) My major concern is notation and presentation of the paper: The paper has too many overloading of notations-- for example, "a" or the actions are giving variable A_t but the system parameter is also A. This has been quite confusing to me for sometimes.

(2) Re. experiments: I am not sure, results of Table 2 are statistically significant: I was looking for paired t test to see how well their method is effective with respect to baselines.

**Questions:**

See above.

---

> ### Author Response · Authors · 2024-11-26
>
> We sincerely appreciate the reviewer’s great efforts and insightful comments to improve our manuscript. We thank the reviewer for recognizing our new problem on TCP on state-space model as well as novelty of our de-correlation mechanism design. In below, we address these concerns point by point and try our best to update the manuscript accordingly.
>
> > **W1: My major concern is notation and presentation of the paper: The paper has too many overloading of notations-- for example, "a" or the actions are giving variable A_t but the system parameter is also A. This has been quite confusing to me for sometimes.**
>
> **Response:** Thank you for your careful review of our manuscript.
> - For **notation**, we have fully polished our manuscript to avoid any overloading of notations. We also **added the following notation table in Appendix A.1.**
> |  Concept |  Abbreviation |  Full Notation |
> |---|---|---|
> |  Treatment | $A$  | $A_{t}$  |
> |  Outcome | $Y$  | $Y_{t}$  |
> | Time-Varying Covariates | $X$  | $X_{t}$  |
> |  Time-invariant Covariates | $V$  | $V$  |
> |  Selective Parameters | $B,C$  |  $B,C$ |
> |  Projection Parameters | $R$  |  $R$ |
> |  Discrete Time Stamp |  $\Delta$  |   $\Delta$ |
> |  Covariance Matrix |  $\Sigma$ | $\Sigma$  |
> |  Hidden State of Mamba |  $h$ | $h_{t}$  |
> |  Time Horizon | $T$  | $T$  |
> |  Our method name | CDSP | Covariate-based Decorrelation towards Selective Parameters  |
>
> - For **presentation**, we have carefully corrected the typos in the manuscript and summarized them below.
>
> |  Before |  After |  Location |
> |---|---|---|
> |  Missing : | Add :  | Line 266  |
> |  CPSD  | CDSP  | Line 316  |
> | Selective parameter ($A$) | $C$  | Lines 206, 208, 212, 306, 318  |
> | Projection parameter ($C$) | $R$  | Lines 204, 206, 209  |
>
> - For the example mentioned, **A_t now only refers to "a" or the actions, not the system parameter.** We have replaced the system parameter A to avoid reuse in Eq. (1) from $h^{\prime}(t)=\mathbf{A} h(t)+\mathbf{B} x(t)$ to $h^{\prime}(t)  =\mathbf{C} h(t)+\mathbf{B} x(t)$.
>
> > **W2: Re. experiments: I am not sure, results of Table 2 are statistically significant: I was looking for paired t test to see how well their method is effective with respect to baselines.**
>
> **Response:** We thank the reviewer for pointing out this issue.
>
> - As suggested by the reviewer, **we have added paired t test comparing our method to the most competitive baseline to all tables including Table 2 in our revised version.**
>
> - **For Table 2, we found that our method stably outperform the best baseline at the significance level $\text{p-value} \leq 0.01$ (marked with \*).**
>
> | |$\tau = 1$|$\tau = 2$|$\tau = 3$|$\tau = 4$|$\tau = 5$|
> |:-:|:-:|:-:|:-:|:-:|:-:|
> |MSMs|6.37$_{\pm 0.26}$|9.06$_{\pm 0.41}$|11.89$_{\pm 1.28}$|13.12$_{\pm 1.25}$|14.44$_{\pm 1.12}$|
> |RMSNs|5.20$_{\pm 0.15}$|9.79$_{\pm 0.31}$|10.52$_{\pm 0.39}$|11.09$_{\pm 0.49}$|11.64$_{\pm 0.62}$|
> |CRN|4.84$_{\pm 0.08}$|9.15$_{\pm 0.16}$|9.81$_{\pm 0.17}$|10.15$_{\pm 0.19}$|10.40$_{\pm 0.21}$|
> |G-Net|5.13$_{\pm 0.05}$|11.88$_{\pm 0.20}$|12.91$_{\pm 0.26}$|13.57$_{\pm 0.30}$|14.08$_{\pm 0.31}$|
> |CT|4.60$_{\pm 0.08}$|9.01$_{\pm 0.21}$|9.58$_{\pm 0.19}$|9.89$_{\pm 0.21}$|10.12$_{\pm 0.22}$|
> |Mamba-HSIC|4.72$_{\pm 0.05}$|5.19$_{\pm 0.19}$|7.24$_{\pm 0.25}$|8.30$_{\pm 0.28}$|9.25$_{\pm 0.30}$|
> |Mamba-CDSP|**4.41**$^{*}_{\pm {\bf 0.05}}$ | **5.04**$^{*}_{\pm 0.13}$ | **5.14**$^{*}_{\pm 0.15}$ | **5.20**$^{*}_{\pm 0.18}$ | **5.25**$^{*}_{\pm 0.19}$ |
>
> ***
>
> We hope the above discussion will fully address your concerns about our work. We look forward to your insightful and constructive responses to further help us improve the quality of our work. Thank you!

---

> > ### Author Response · Authors · 2024-12-02
> >
> > Once again, we are grateful for your time and effort for reviewing our paper. Since the discussion period will end in around a day, we are very eager to get your feedback on our response. We understand that you are very busy, but we would highly appreciate it if you could take into account our response when updating the rating and having a discussion with AC and other reviewers.
> >
> > Thanks for your time,
> >
> > Authors of # 7314

---

> > > ### Author Response · Authors · 2024-12-03
> > >
> > > Dear reviewer TFBs,
> > >
> > > Since the discussion period will end in a few hours, we will be online waiting for your feedback on our rebuttal, which we believe has fully addressed your concerns.
> > >
> > > We would highly appreciate it if you could take into account our response when updating the rating and having discussions with AC and other reviewers.
> > >
> > > Thank you so much for your time and efforts. Sorry for our repetitive messages, but we're eager to ensure everything is addressed.
> > >
> > > Authors of # 7314

---

> > > > ### Comment · Reviewer_TFBs · 2024-12-03
> > > > **Thanks!**
> > > >
> > > > Thanks for your response, most of concerns are addressed. However, I will keep my score same, since I don't think the paper crosses the bar to get an '8'. Also, I would request that notations should be in the main paper rather than Appendix.

---

> ### Author Response · Authors · 2024-12-03
> **We're glad that most of your concerns are addressed! Thank you for maintaining on the positive side!**
>
> Dear reviewer TFBs,
>
> We're glad that most of your concerns are addressed! By now, we are appreciated that all reviewers are on the positive side of our paper acceptance, except for one reviewer who uploaded a review of another ICLR submission : (
>
> We will definitely put the notations table in the main paper rather than Appendix - thank you for your kind reminder!
>
> Many thanks,
>
> Authors of # 7314

---

### Official Review · Reviewer_WoTq · 2024-11-04

**Soundness:** 2
**Presentation:** 3
**Contribution:** 3
**Rating:** 5
**Confidence:** 3

**Summary:**

This paper presents a Time-shared Heterogeneity Learning from Time Series (THLTS) method which infers the shared part of latent factor across time steps with a variational auto-encoders (VAE), the method could capture the hidden heterogeneity by recovering the hidden factors and incorporate it into the outcome prediction. This method can be a flexible component to be easily inserted into arbitrary counterfactual outcome forecast models. The authors demonstrate the effectiveness of THLTS on (semi-)synthetic data in capturing shared patterns by combining several existing counterfactual outcome forecast methods to improve their performance.

**Strengths:**

A substantive assessment of the strengths of the paper, touching on each of the following dimensions: originality, quality, clarity, and significance. We encourage reviewers to be broad in their definitions of originality and significance. For example, originality may arise from a new definition or problem formulation, creative combinations of existing ideas, application to a new domain, or removing limitations from prior results. You can incorporate Markdown and Latex into your review. See https://openreview.net/faq.

Originality
The paper proposes a novel approach to capturing hidden heterogeneity in time series based counterfactual prediction, which is a significant domain problem in causal learning. The proposed Time-shared Heterogeneity Learning from Time Series method is a novel method that addresses this specific challenge by encoding the shared time-aware latent confounder and then utilizing them for counterfactual outcome forecasting.

Quality
The paper provides a clear and well-structured presentation of the proposed method, including a detailed explanation of the shared latent confounder variable encoding process via VAE and how to adapt to time series data.
The experimental results basically demonstrate the effectiveness of the proposed method in improving the performance of mainstream models.

Clarity
The paper is well-written and easy to follow, with clear explanations of technical concepts and methods. The authors provide an informative context in each section that effectively organizes the story and summarizes the paper contributions.

Significance
The proposed THLTS method has the potential to improve the accuracy of counterfactual outcome in time-series data scenarios. The capture of hidden heterogeneity across time domains is a common challenge in many fields. The proposed method is flexible and can be easily inserted with arbitrary causal modeling framework, making it a valuable contribution to the field.

**Weaknesses:**

Lack of Novelty in Methodology
The proposed THLTS method is based on the use of variational encoders (VAE) to improve counterfactual prediction in time-series data, which has been explored in other works such as
1. https://doi.org/10.1145/3637528.3671950,
2. https://doi.org/10.48550/arXiv.2310.18615 .
While the shared latent factor encoding part is new, the underlying methodology is not entirely novel. To strengthen the contribution, the authors could provide a more detailed comparison with existing methods and highlight the specific advantages of their approach.


Limited Experimental Evaluation
The paper only presents experimental results on (semi) synthetic datasets, which may not accurately reflect real-world scenarios. To demonstrate the practical applicability of the proposed method, it would be beneficial to include experiments on (or the connection to) real-world datasets .

**Questions:**

Please list up and carefully describe any questions and suggestions for the authors. Think of the things where a response from the author can change your opinion, clarify a confusion or address a limitation. This is important for a productive rebuttal and discussion phase with the authors.



Besides RMSE, it would be good to add other ablation study such as distribution analysis of the counterfactual prediction from utilizing the proposed method vs. baselines, which would provide more evidence to validate the the effectiveness of introducing the shared latent factor as illustrated in Figure 1
In table1 and table2, could the author elaborate on more details of the baseline THLTS(v)? Why author think this would be fair baseline to justify the rationality of learning shared part of latent factors compared to the proposed method
In Algorithm 1, what is the difference between forecast model pρ() and gρ()?

---

### Author Response · Authors · 2024-11-28
**General responses and manuscript revision summary**

Dear reviewers and AC,

We sincerely thank all reviewers and AC for their great effort and constructive comments on our manuscript. We know that we are now approaching the end of the author-reviewer discussion and apologize for our late rebuttal. During the rebuttal period, we have been focusing on these beneficial suggestions from the reviewers and doing our best to add several experiments and revise our manuscript. We believe our current carefully revised manuscript can address all the reviewers’ concerns.

As reviewers highlighted, we believe our paper **tackles a new problem on time-varying counterfactual prediction with high practical value** (Reviewer TFBs, Reviewer FCGz, Reviewer 1NGR). We also appreciate that the reviewers found the **proposed methods well-motivated** (Reviewer PLDR and Reviewer 1NGR), **well-designed with efficient implementation** (Reviewer FCGz, Reviewer TFBs), offers **a novel solution to the over-balancing problem in sequential settings** (Reviewer PLDR, Reviewer TFBs), as well as **comprehensive experiments across multiple datasets and settings** (Reviewer FCGz, Reviewer 1NGR and Reviewer PLDR).

Moreover, we thank the reviewers for pointing out the limitations regarding **more through theoretical motivation and analysis** (Reviewer PLDR, Reviewer 1NGR and Reviewer FCGz), as well as for the suggestions for investigating **extreme-long sequences** (Reviewer FCGz), **hyper-parameter sensitivity analysis** (Reviewer PLDR and Reviewer FCGz), **more real-world experiments** (Reviewer FCGz), **more through related work review** (Reviewer PLDR), **more detailed clarification of our method on correction of confounding bias** (Reviewer PLDR), **clarification on difference between existing balancing work** (Reviewer 1NGR), **paired-t test** (Reviewer TFBs), and **typo corrections** (Reviewer TFBs). In response to these comments, we have carefully revised and enhanced our manuscript with the following important changes with the added experiments:

- [Reviewer PLDR, Reviewer 1NGR and Reviewer FCGz] We **add theoretical analysis to analyze the risk bounds of counterfactual prediction methods** with detailed proofs in Appendix A.3, showing that **CDSP has a tighter counterfactual prediction risk bound than traditional balancing approaches** in a new Section 4.4.
- [Reviewer PLDR, Reviewer 1NGR] We **re-write the related-work section with a more thorough and complete review** by adding over 15 new recent works in Section 2.
- [Reviewer FCGz] We **add experiments on extremely long sequences** using the TG simulator data with $L = 1000$ in Table 4, Page 10.
- [Reviewer PLDR and Reviewer FCGz] We **add sensitive analysis to study the choice of decorrelation hyperparameter $\alpha$** with varying time-steps $\tau \in \\{1, \ldots, 10\\}$ on the semi-synthetic MIMIC-III dataset in Table 3.
- [Reviewer 1NGR, Reviewer FCGz] We follow Huang et al. (ICML 2024) to **add experiments on a new real-world time-series prediction dataset,** i.e., the M5 dataset in Table 8.
- [Reviewer FCGz] We **add both theoretical analysis and empirical validations on computation complexity** to validate the efficiency of our proposed CDSP over traditional balancing methods in Section 4.3 and Appendix A.4.3.
- [Reviewer 1NGR, Reviewer PLDR] We **add synthetic experiments with increasing confounding bias** to validate our method outperforms causal transformer due to the confounding bias correction in Table 9.
- [Reviewer TFBs] We **add paired t-test throughout our revised paper** to show the superiority of our method.

These updates are temporarily highlighted in "$\textcolor{red}{red}$" for facilitating checking.

We hope our response and revision could address all the reviewers' concerns, and are more than eager to have further discussions with the reviewers in response to these revisions.

Many thanks,

Submission7314 Authors

---

### Meta-Review · Area_Chair_wVRF · 2024-12-24

**Metareview:**

This paper introduces Mamba-CDSP, an approach for time-varying counterfactual prediction (TCP) using state-space models. The key innovation lies in combining the Mamba architecture (a recent advance in state-space modeling) with a new Covariate-based Decorrelation towards Selective Parameters (CDSP) mechanism. The method effectively addresses two major challenges in TCP: computational efficiency and the over-balancing problem. The authors demonstrate that Mamba-CDSP outperforms existing methods like Causal Transformer and G-Net on both synthetic and real-world datasets, showing superior performance across multiple metrics.

In the rebuttal period, the authors have made substantial efforts to revise the manuscript based on the feedback provided. Most of the major concerns raised by the reviewers, such as improving theoretical justification, adding experiments on long sequences, sensitivity analysis, real-world dataset validation, and more thorough experimental setups, have been addressed. The revisions also include significant additions, including the inclusion of sensitivity studies, new datasets, and statistical tests, which strengthen the overall contribution of the paper.

Given the positive feedback from the reviewers and the improvements made, I recommend accepting the paper for publication.

**Additional Comments On Reviewer Discussion:**

The authors have made substantial revisions based on the reviewers' comments, addressing both theoretical and empirical concerns. The inclusion of new experiments, enhanced theoretical analysis, and a more thorough review of related work have significantly improved the manuscript. The added sensitivity analyses, real-world experiments, and statistical tests have strengthened the paper's claims and provide a clearer understanding of the method's strengths and limitations.

While some concerns were raised during the rebuttal period regarding the completeness of certain aspects, such as the theoretical analysis and real-world validation, the authors have made convincing changes to address these points. The majority of the issues raised by the reviewers have been resolved or adequately addressed, demonstrating the authors' efforts to enhance the manuscript.

---

### Decision · Program_Chairs · 2025-01-22

Accept (Poster)